# Recurrent connections facilitate occluded object recognition by explaining-away

Byungwoo Kang[1,2,3], Benjamin Midler[1,4], Feng Chen [1,5] &
Shaul Druckmann [1,6] ✉

Despite the ubiquity of recurrent connections in the brain, their role in visual processing is less understood than that of feedforward connections. Occluded object recognition, an important cognitive capacity, is thought to rely on recurrent processing of visual information, but it remains unclear whether and how recurrent processing improves recognition of occluded objects. Using convolutional models of the visual system, we demonstrate how a distinct form of computation arises in recurrent−but not feedforward−networks that leverages information about the occluder to "explain-away" the occlusion−i.e., recognition of the occluder provides an account for missing or altered features, potentially rescuing recognition of occluded objects. This occurs without any constraint placed on the computation and is observed both across a systematic architecture sweep of convolutional models and in a model explicitly constructed to approximate the primate visual system. In line with these results, we find evidence consistent with explaining-away in a human psychophysics experiment. Finally, we developed an experimentally inspired recurrent model that recovers fine-grained features of occluded stimuli by explaining-away. Recurrent connections' capability to explain-away may extend to more general cases where undoing context-dependent changes in representations benefits perception.

Vision is a key sensory modality for primates, and the study of visual processing has revealed diverse principles of neural computation, circuit organization and function[1,2]. Most classical descriptions of information processing in the visual cortex focus on its feedforward nature[3–5], yet a growing body of literature suggests that recurrent processes also play important computational roles based on computational[6–10] and experimental evidence[11,12]. In particular, the visual processing of occluded objects, i.e., objects in a scene that are partially obscured from view, has garnered much attention as one of the major tasks for which recurrent computation might be crucial. Recognizing occluded objects can be ethologically critical for animals, for instance, when identifying predators partially concealed in vegetation.

Evidence for the importance of recurrent computation in recognizing occluded objects has come both from the dynamics of neural responses and the effect of temporal context on behavior. First, studies have observed that object-selective responses emerge about 50–100 ms later for occluded stimuli than for unoccluded stimuli across multiple contexts[13–19]. Second, backward masking− when a high-contrast noise mask is presented soon after an image (typically 25 to 150 ms after the image onset)−was found to disrupt potential recurrent processing, and has been shown to impair recognition of occluded stimuli more severely than unoccluded ones[17,18,20]. While these observations do not prove that occluded object recognition involves recurrent processes, they do strongly

[1]Department of Neurobiology, Stanford University, Stanford, CA, USA. [2]Department of Physics, Stanford University, Stanford, CA, USA. [3]Department of Neurobiology, Harvard Medical School, Boston, MA, USA. [4]Princeton Neuroscience Institute, Princeton University, Princeton, NJ, USA. [5]Department of Applied Physics, Stanford University, Stanford, CA, USA. [6]Department of Electrical Engineering, Stanford University, Stanford, CA, USA. ✉e-mail: shauld@stanford.edu

suggest that it involves some form of temporally extended processing.

Several studies have proposed specific roles for recurrence in vision—including figure-ground segmentation, contour integration, and object recognition[21–24]—yet the precise nature and mechanistic role of recurrent computation remain only partially understood. A key experimental challenge is that most neurons possess both feedforward and recurrent connections, making it difficult to selectively perturb one or the other. Therefore, in this study, we tackled these questions by constructing feedforward and recurrent models of the visual system and analyzing their performance and internal representations. For our models, we use deep convolutional neural networks, which have been shown to not only achieve human-level performance but also predict cortical visual responses for object recognition[6,8,25–27].

To understand the impact of recurrence on processing occluded stimuli, we first created datasets of occluded images (due to the lack of established ones) and trained 23 architecturally different convolutional neural networks on occluded object recognition tasks. Through careful comparisons and controls across architectures and tasks, we analyzed whether recurrent computations offer any unique computational advantages over feedforward computations.

Unexpectedly, we found that "explaining-away"—the ability to incorporate the recognized occlusion into the classification of the occluded object[28–31]—arose naturally in recurrent but not feedforward networks. Our use of the term "explaining-away" is rooted in its typical formulation in probabilistic graphical models[32], where identifying one cause of an observed outcome can reduce (or "explain away") the need to posit other causes. Concretely, in the context of occlusion, the observed "outcome" is an unusual collection of visual features that deviates from those normally associated with a single visible object. One potential explanation for these unexpected features is that a genuinely unfamiliar object is present, whereas another explanation is that the scene includes an occluding object. Recognizing that an occluder is the cause of the unusual features thus makes it less necessary to posit alternative explanations for those features. While the exact usage of "explaining-away" varies in the literature, here we focus on this intuitive notion that identifying an occluder helps the system handle the mismatch in visual features, thereby facilitating recognition of the occluded object.

The fact that explaining-away emerged despite networks never directly being trained to do so suggests that this property is likely insensitive to the learning algorithm employed (of which we have very little knowledge), but rather is a robust consequence of the recurrent architecture together with the temporal structuring of the task where the network is required to recognize the occluder prior to the occluded object.

Additionally, our controlled architecture analyses found that recurrent computation is not intrinsically superior to feedforward computation on occluded object recognition, as sufficiently deep feedforward networks were able to outperform recurrent networks. Notably, control experiments showed that the high performance of deep feedforward networks was likely not achieved simply by converging on recurrent solutions (which are a subclass of feedforward solutions), but rather was composed of unique weight profiles. Performance correlated well not with overall parameter number but with computational depth, whether depth was achieved by feedforward layers or timesteps in a recurrent network. Our results challenge the conclusions of earlier studies, which reported considerable advantages for recurrent processing in comparison to feedforward processing on occluded object recognition[17,18,33]. This is consistent with previous studies showing that feedforward models can outperform recurrent models on object recognition without occlusion[34].

To test whether explaining-away computations are present both in more realistic models of the primate visual system and in humans, we conducted two additional experiments: one computational and one behavioral. First, we fine-tuned a network built to recapitulate key properties of the primate visual system and predict its neural activity, named CORNet[35], on occluded object recognition. To train the network, along with several control variants, which required higher resolution images, we created another dataset of occluded images in which realistically rendered 3D objects are set against a complex background and occlude one another. Second, we conducted a visual psychophysics experiment in human subjects designed to test behavioral signatures of explaining-away. We found evidence for the presence of explaining-away in both experiments.

We subsequently developed a more detailed, neurobiologically inspired model demonstrating that recurrent processing can morph the internal representation of an occluded object to a state more similar to that of an unoccluded one. This "reconstitution" of the unoccluded representation was effective enough for decoders trained on unoccluded objects to decode reconstituted occluded objects, while the same decoders performed much more poorly without reconstitution.

Finally, to understand how the need to process occluded stimuli affects circuit dynamics, we asked how the internal representations of the same visual stimulus differ between networks with the same architecture but trained either on unoccluded or occluded stimuli. As expected, we found that training on occluded stimuli substantially changes network connectivity. Consequently, responses to occluded stimuli were altered and enhanced. Surprisingly, we found that, despite the very different weights learned, representations of unoccluded stimuli were largely unchanged. This computational solution—enhancing occluded stimuli representations while keeping unoccluded stimuli representations intact—may facilitate downstream processing by limiting representational changes due to learning. In addition, our findings highlight the importance of appropriate probe stimuli when comparing circuits or when artificial networks are compared to real circuits.

In summary, our work elucidates the role of recurrent connectivity, a ubiquitous feature of biological circuits, in enabling distinctive computational properties and capabilities. In particular, it highlights the importance of "explaining-away" as a key computation for processing occluded visual stimuli and demonstrates a potential mechanism by which it could occur, with experimentally inspired models of visual circuits able to effectively reconstitute the representation of occluded objects. Such computational reconstitution can be tested experimentally by analyzing temporal response patterns to occluded and unoccluded objects. More generally, similar mechanisms may allow the brain to disentangle other types of complex sensory stimuli that naturally occur in ethological settings.

## Results

### Controlled comparisons across convolutional models of occluded object recognition reveal explaining-away by recurrent connections

We performed a large-scale computational experiment using 23 network architectures and a newly developed occluded image dataset (Fig. 1). It consisted of images of an occluded background object and a foreground object, with both objects sampled from FashionMNIST, a standard image classification dataset[36]. We first selected four classes (T-Shirt/Top, Pullover, Bag, and Ankle boot) out of the total ten classes in FashionMNIST. Classes were selected on account of their comparable areas, making it easier to control occlusion levels. To generate a two-object image with occlusion, we randomly sampled two FashionMNIST images from the selected classes, applied a random rotation and translation to each object, and placed them together on a $50 \times 50$-pixel square black background. We only kept images with occlusion levels between 50 and 75% and ensured the distribution of occlusion levels was uniform in that range (see "Methods"). In total, the dataset

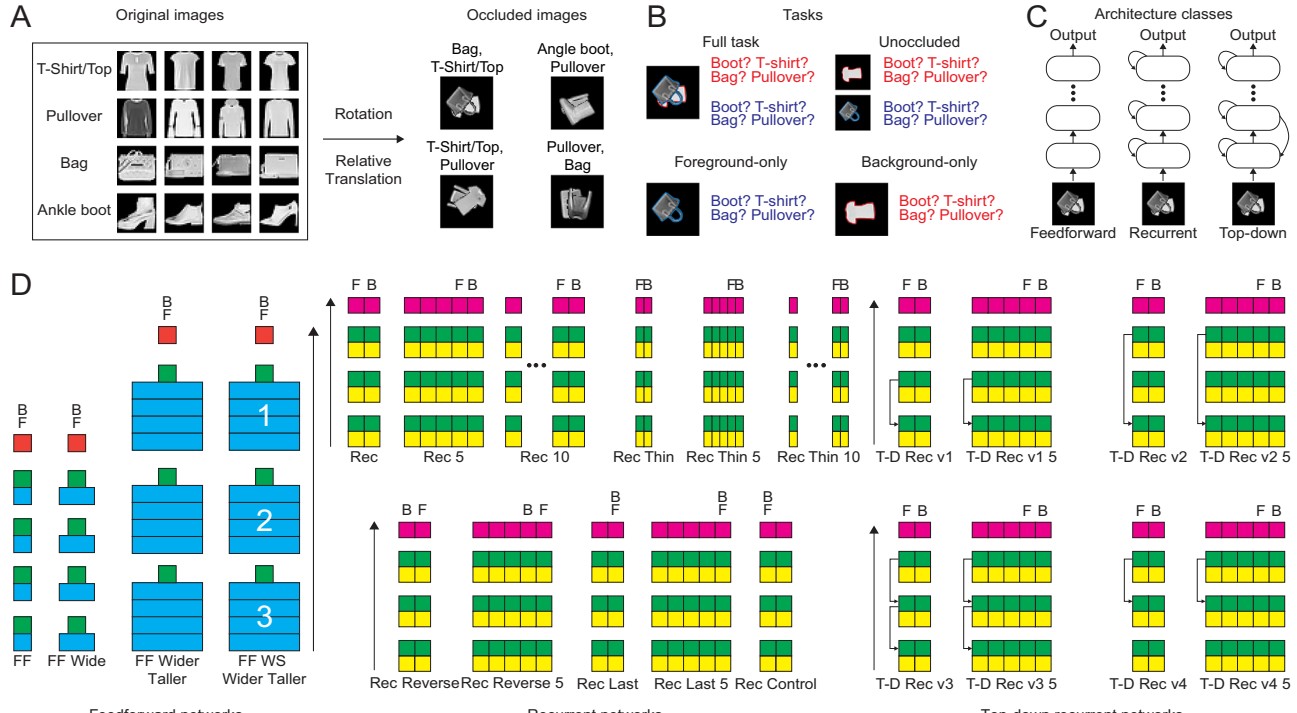

**Fig. 1 | Task description and network architectures. A** Example images from the fashion MNIST dataset and the dataset used for training. Left: original fashion MNIST images. Each row is a separate class and each column has a different image within the class. Right: images used in our study. **B** Schematics of the different tasks. In the full task, the category of both the foreground (occluding) and background (occluded) objects needs to be predicted. In the foreground- and background-only tasks, the class of only one object needs to be predicted. In the unoccluded task, the network receives an input containing each object separately on a black background. **C** Schematics of the three architecture classes. **D** Schematics of specific architectures. Width of the rectangle represents the number of units, with the smallest width corresponding to 64 units. Blue corresponds to convolutional layers, green to max-pool operations, red to fully connected layers, yellow to ConvLSTM layers and magenta to LSTM layers. All convolutional layers have a kernel size of 5, and max-pooling a kernel size of 2. Columns indicate the number of time steps networks are run (feedforward networks are run for one time step, recurrent and top-down for more). Letters indicate the time step at which a category classification is outputted for the foreground object (denoted by F) and background object (denoted by B). Letters stacked vertically indicate simultaneous output of both categories. Roman numerals indicate weight sharing in the weight-shared feedforward network. (*Left*) feedforward networks. (*Middle*) recurrent networks. (*Right*) top-down recurrent networks.

contained 16 classes, labeling both the foreground- and background-object classes (Fig. 1A).

We trained networks on four tasks (Fig. 1B). The first, which we refer to as the full task, requires networks to correctly determine the class of the foreground and background object. The rest of the tasks were designed as controls for different aspects of this task. The second, the foreground-only task, had the networks determine only the class of the foreground object. The third, the background-only task, had the networks determine the class of the background object. The fourth, the unoccluded task, had the networks determine the class of the foreground and background object, but with unoccluded inputs. Specifically, the networks' input was not the occluded image, but rather two images, one in which only the foreground object is present and another in which only the background object is present, in separate channels. In these images, the objects are translated and rotated exactly as in the occluded image. The foreground- and background-only task controls were used to more precisely tease apart temporal computations, as explained in more detail below. The unoccluded task kept the same two-object classification but controlled for the existence of occlusion.

We trained three general categories of network architectures: feedforward, recurrent, and top-down recurrent (Fig. 1C). The feedforward networks were composed of convolutional layers followed by fully connected readout layers. The recurrent models had ConvLSTM[37] layers, followed by an LSTM[38], and finally fully connected readout layers. The top-down recurrent models similarly had ConvLSTM and LSTMs followed by a fully connected readout layer, but, in addition, they had connections from higher layers to lower layers.

We trained four feedforward architectures on each task (Fig. 1D). The network FF had three convolutional and one fully connected hidden layers. The convolutional layers had 64 feature maps, and the fully connected layer had 1024 units. FF Wide had the same number of layers as FF but twice as many units per layer. FF Wider Taller had twelve convolutional layers and four times as many units per layer as FF. Finally, FF Weight Shared (WS) Wider Taller had the same number of units and layers as FF Wider Taller, but certain layers shared weights (Fig. 1D). The networks had the same architecture across tasks, except for the readout layer. In the full task and unoccluded task, in which the networks were required to predict the classes of both the foreground and background objects, the networks had two parallel fully connected readout layers following the last hidden layer. In the foreground- and background-only tasks, where only one class was to be predicted, the networks had only one fully connected readout layer.

We trained eleven recurrent network variants which differed in architecture, the time steps in which the class predictions were performed and the number of time steps their dynamics were run for (Fig. 1D). The network Rec had three layers of ConvLSTMs with 64 feature maps––the same number of feature maps as the feedforward architecture, each followed by a max-pool, then an LSTM layer with 512 units and finally a readout layer. We chose the LSTM layer to have half the number of units as the fully connected layer of FF because the recurrent networks output one prediction per time step, whereas the feedforward networks produce output predictions for both objects simultaneously. The network was run for two time steps. It outputs the identity of the foreground object in the first time step and the

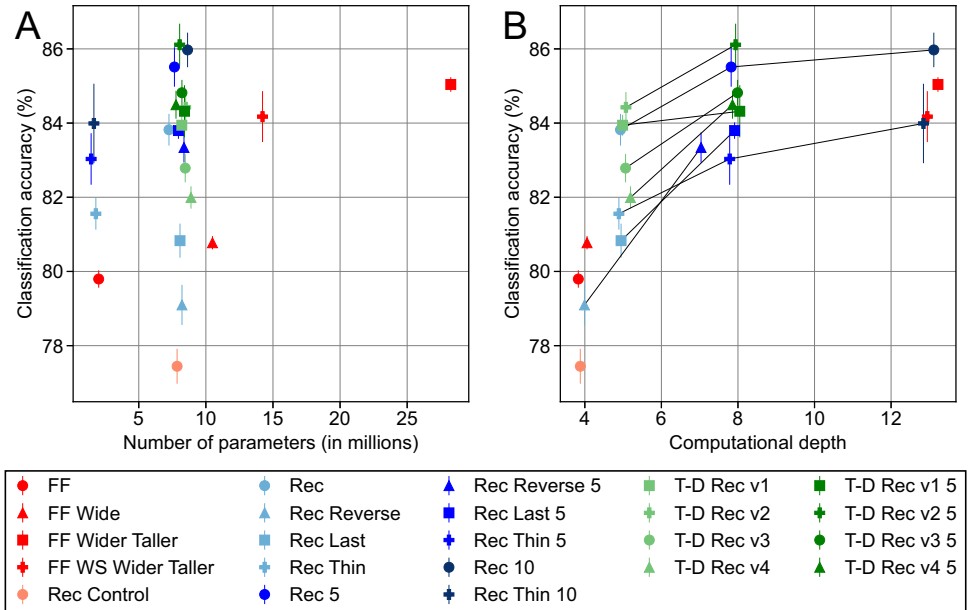

**Fig. 2 | Comparison of occluded image classification across architectures.**
**A** Task performance vs. number of parameters. Each symbol corresponds to a network architecture, with the average test performance of that architecture across five independent initializations on the $y$-axis and the number of parameters on the $x$-axis. The error bars indicate the standard deviations across the initializations ($n = 5$). A random Gaussian noise of zero mean and 0.5 standard deviation (in units of millions, consistent with the $x$-axis) was added to each dot's x-coordinate to separate overlapping dots. For Rec, instead of adding a random noise, we displaced its dot's x-coordinate by 1 toward the left to avoid overlap with other nearby dots. The symbol legend is given in the box below. **B** Task performance vs.

computational depth. Symbols indicate network architectures as in (**A**). The $x$-axis corresponds to the computational depth, defined as the maximum number of times an input variable is affine-transformed by the network's weights and then transformed by a nonlinearity until it reaches the output end of the network. The $y$-axis indicates the average test performance with the error bars indicating the standard deviations across the initialization ($n = 5$). Lines connect recurrent models of the same architecture that only differ in the number of time steps they are run for. We added a random Gaussian noise of zero mean and 0.15 standard deviation to each dot's x-coordinate to separate out overlapping dots.

background object in the second. The variants Rec 5 and Rec 10 were run for five and ten time steps, respectively, outputting the identity of the foreground object in the next-to-last time step and the identity of the background object in the last time step. The "Rec Thin" network contained half the number of units at each layer. Since ConvLSTM and LSTM layers have more parameters per unit than standard feedforward layers, cutting the number of units by half roughly equated the number of parameters between the Rec Thin architectures and the FF architecture above. The "Rec Reverse" network outputted the object classes in reverse order: the class of the background object in the next-to-last time step and that of the foreground object in the last time step, testing the utility of identifying the occluding object first. The "Rec Last" network outputs both object classes simultaneously at the last time step, testing the utility of sequential predictions. The "Rec Control" network outputs both classes on the very first time step, similar to feedforward networks.

For top-down recurrent networks, there are a large number of possible top-down connection patterns that can be added to a recurrent network to make it a top-down recurrent network. In other words, there exist many possible connection patterns between higher and lower layers. We tested a large number of possible combinations and found that, in our hands, many variants were difficult to train and performed poorly. We therefore present the four versions that were competitive with recurrent networks (Fig. 1D): T-D Rec v1 has connections from the second ConvLSTM layer to the first; T-D Rec v2 from the third ConvLSTM layer to the first; T-D Rec v3 from the third to the second and second to first; T-D Rec v4 from the third to the second (see "Methods").

We found networks from both the feedforward and recurrent network architectures that achieved high performance on the full task (Fig. 2A). When comparing networks with an equal number of layers, however, the recurrent models outperformed the feedforward

models. Specifically, FF was outperformed by all the standard recurrent models (i.e., recurrent models that first predicted the foreground then the background image, unlike some of the control recurrent architectures, for more details, see architecture descriptions above or in "Methods"). FF Wide, which has twice as many hidden activations as FF at each layer, performed somewhat better than FF but still worse than most of the recurrent models.

The recurrent models' superior performance was not due merely to an increased number of parameters, but rather due to their ability to perform recurrent computations. This was highlighted by the performance of Rec Control, which has the same architecture—and thus the same number of parameters as the recurrent models—but only runs for one time step. Rec Control was outperformed by all the recurrent models as well as the feedforward ones. Also consistent with this observation, the recurrent models performed significantly better with an increased number of iterations, even though these iterations did not increase the number of parameters. For all families of recurrent architecture (Rec, Rec Reverse, and Rec Last), the models that ran for five time steps outperformed those that ran for two time steps by a clear margin. The same tendency held for the top-down recurrent models as well.

Contrary to previous claims[16–18,33], however, recurrent computation was not intrinsically superior to feedforward computation in dealing with occlusion. In particular, a deep, purely feedforward model, FF Wider Taller, clearly outperformed most of the recurrent models and performed nearly as well as the best performing recurrent model, Rec 10.

We found only a weak correlation between performance and number of parameters (Fig. 2A). Instead, we observed that the "computational depth" of a neural network was a better predictor. Following the widely accepted intuition that recurrence enables networks to use time as a computational resource, we defined the computational depth

of a network as the maximum number of times an input variable is affine transformed by the network's weights and then transformed by a nonlinearity (see "Methods"). For an ordinary feedforward network, its computational depth is equal to the number of layers. For an ordinary recurrent network, its computational depth is equal to the sum of the number of layers and the number of recurrent iterations. We note that this measure is conceptually very similar to previously considered measures, such as the number of floating-point operations[34], and we only introduce this simplified measure as it can be more readily read out from the architecture of the network. Performance was highly correlated with computational depth, although it tended to saturate at a depth greater than eight (Fig. 2B). This nicely dovetails with our prior observations on recurrent versus feed-forward performance, as computational depth can be increased by adding recurrent iterations and/or having more layers.

To more cleanly delineate whether recurrent computations are advantageous at a given computational depth, we considered weight-shared feedforward networks. Feedforward networks typically have different weights across layers, but recurrent networks use the same weights across time. Similarly, weight-shared feedforward networks have the same structure as feedforward networks, but the weights at certain layers are forced to be identical to the weights at other layers, mimicking recurrent weight sharing. We found that the standard feedforward network, FF Wider Taller, outperformed its weight-shared counterpart, FF WS Wider Taller, but by a relatively small margin. This suggests that repeating an identical computation over time––a defining characteristic of recurrent networks––has no significant advantage in handling occlusion.

We find that the relation between recurrent architectures and top-down recurrent architectures was more complex (Fig. S1E). Similar observations were also recently made in ref. 6 in the context of single-object recognition.

While our discussion so far has focused on the multi-object classification task, we observed similar trends in the background and foreground object classification tasks as well (Fig. S1B, C). The ordering of relative performance across architectures was largely conserved for the foreground-only task. Notably, the performance difference across models in the unoccluded multi-object classification task was smaller than in the other tasks (Fig. S1D). This is in line with our expectation that easy tasks will not differentiate between architectures. More importantly, the change in relative performance for this easier task indicates the differences we find above are not just due to an intrinsic training difficulty for a subset of the architectures.

Finally, we tested one of the central intuitive notions behind the advantage of recurrent computation: "explaining-away"[28–31]. Namely, that recognition of a foreground object can feed back into the model's internal representation and account for the different appearances of occluded and unoccluded objects, facilitating classification (Fig. 3A). To study this phenomenon, we compared performance on background object recognition in two contexts: (i) when networks were tasked to recognize a background object after first recognizing a foreground object, and (ii) when networks identified only the background object. In line with the idea of explaining-away, we found that recurrent models–but not feedforward models–tended to recognize the background object better when they also had to recognize the foreground object (Fig. 3B). In more detail, we compared the performance of recognizing background objects between networks trained only to predict the background object and those performing the full task. In the background-only task, the recurrent networks predicted the background object in the last time step. In the full task, the recurrent networks also predicted the background object in the last time step, but first they predicted the foreground object in the next-to-last time step. In contrast, feedforward networks have no temporal activation and therefore perform simultaneous prediction both in the case of the background object only classification and multi-object

classification. We find that almost all the recurrent networks predicted the background object better when required to predict the foreground object at the previous time step in the full task (Fig. 3B). Feedforward networks, however, performed slightly worse on categorizing the background object in the full task (Fig. 3B). We note that the Rec Last and Rec Control architectures are recurrent but similar to feedforward models in that they also predict both objects simultaneously and thus are grouped with feedforward models for this comparison.

Importantly, we did not observe an analogous performance gain in foreground object recognition when models first predicted the background object (Fig. 3C). This is expected since the entire foreground object is fully visible; thus, there is nothing to explain away. This finding demonstrates that the performance gain in background object recognition when first predicting the foreground object cannot be attributed to having more training targets, as the same benefit did not extend to predicting the foreground object. Note that here we could only compare the subset of architectures that predict the foreground object in the same time step (the last time step) in the full task and foreground-only task. We did this to avoid the confound of having different numbers of time steps before prediction in the two tasks, which would likely skew the results given the aforementioned importance of computational depth.

As a further test of the "explaining-away" hypothesis, we compared our standard recurrent networks (Rec architecture), which predicted the foreground object first, with recurrent networks that were identical except they were required to predict the background object first (Rec Reverse architecture). The standard recurrent models outperformed the reverse ones, both in the two-time-step and five-time-step variants, consistent with the explaining-away hypothesis (Fig. 3D).

Having found evidence for explaining-away, we tested the phenomenon using a new dataset of realistically rendered images and a neural network model designed to capture key features of the primate visual system (Fig. 4). We created the 3D occlusion dataset from ThreeDWorld[39], a high-fidelity multi-modal platform for physical simulation. We curated a selection of 10 object classes (Fig. 4A) and constructed each image with an occluding object and an occluded object presented on a complex background of an artificial room (Fig. 4B, see "Methods").

We tested a model of the primate visual system, CORnet-R, which is among the top models on Brain-Score and was built specifically to model the neural mechanisms of object recognition[35]. As CORnet-R was not originally trained on occluded images, we fine-tuned it on a classification task based on two versions of the occlusion dataset: standard order and reverse order, as in our previous experiments. In the standard order condition, the model was trained to predict the occluding object first and then the occluded object. In the reverse order, the prediction order is reversed, and the occluded object identity is predicted first, followed by the occluder (Fig. 4C).

Testing on held-out data, and considering background object classification, we found that the standard order achieved higher accuracy, supporting the explaining-away hypothesis (Fig. 4D). Testing models run for an increasing number of timesteps, we found that the background object classification accuracy increased and the gap between the two orders decreased with processing time steps. Still, the performance of the standard order models was significantly higher than the reversed order models across all tested timesteps (Fig. 4D, E). In summary, we find the evidence consistent with explaining-away in rich images processed by a model directly built to model object recognition.

To further elucidate how recurrent networks "explain away" the presence of an occluded object, we performed targeted perturbation experiments inspired by prior work on causal manipulations of internal network states[23]. Concretely, we used our recurrent architecture (Rec) that predicts the foreground object at the first timestep and then

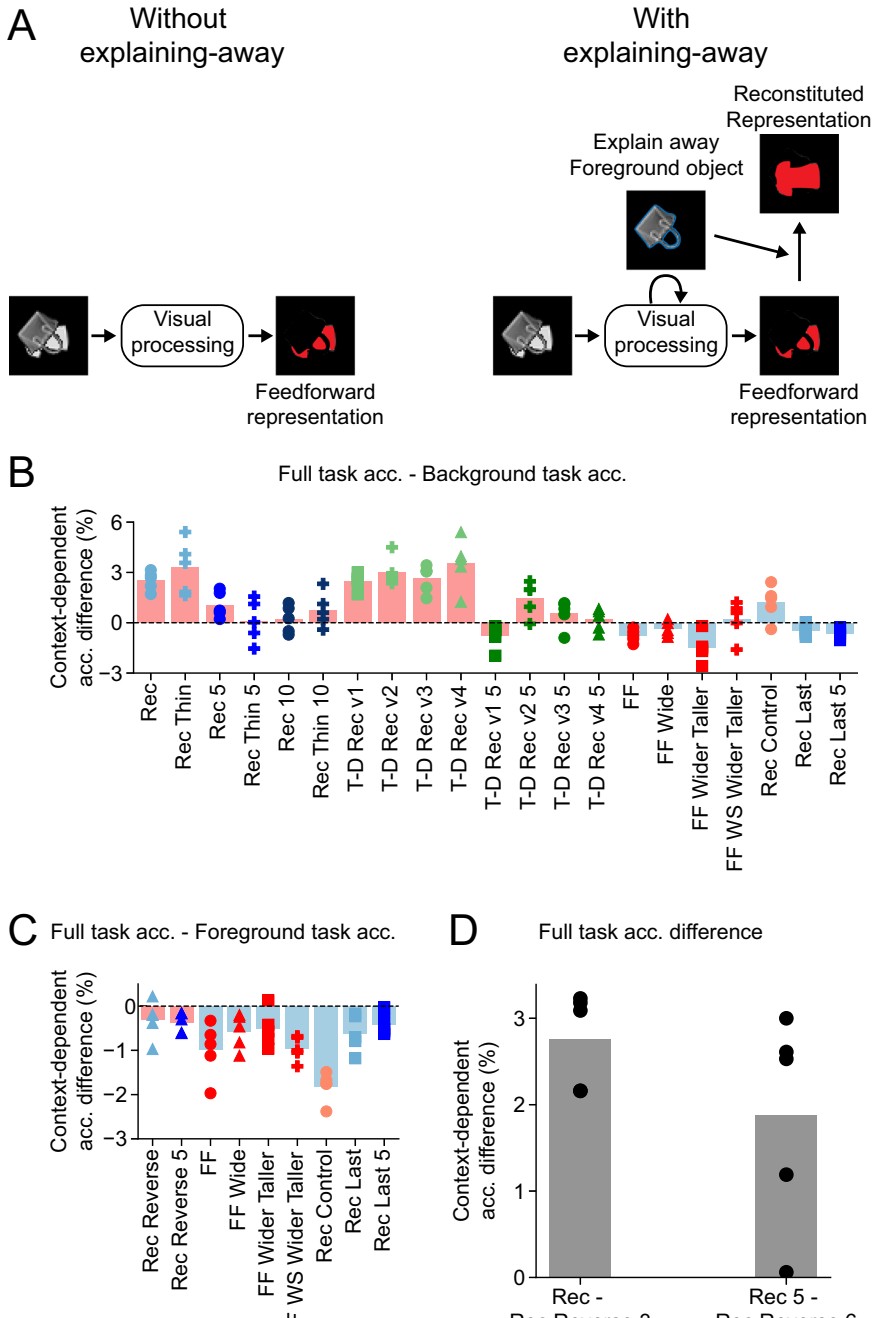

**Fig. 3 | Evidence for an "explaining-away" computation in occluded object recognition. A** A schematic describing the idea of explaining-away computation. Without explaining-away, visual processing has to rely on a (potentially complex or even fragmented) partial view of the occluded object to guess its identity. In contrast, with explaining-away, visual processing can leverage its knowledge of the foreground object to explain away the missing parts of the occluded object, thereby facilitating downstream cognition. **B** Comparison of classifying the background object in the context of full classification and background-only classification is consistent with explaining-away. For each network architecture, we plot the difference between the performance of classifying the background object when it is performed in the context of the full (background and foreground) task and in the background-only task. Positive values indicate better performance in the context of the full task. Bars indicate the average difference across multiple independent networks, symbols show differences between individual networks ($n = 5$). Light red bars indicate network architectures that first output the foreground class and then the background class (e.g., Rec and Rec 5). Light blue bars indicate networks that simultaneously output the foreground and background classes (e.g., feedforward networks). **C** Control analysis for "explaining-away". Same analysis and plotting conventions as in (**B**), but now comparing the performance of classifying the foreground object in the context of the full task and foreground-only task. Note that the smaller number of architectures is due to the fact that to match the number of processing steps we only take architectures that in the full task predict the foreground object in the last time step. **D** The difference in performance of the background object classification between full-task models with identical computational depth. Note that although the two models have identical computational depth, one of them predicts the foreground object first (e.g., Rec) while the other predicts the background object first (e.g., Rec Reverse 3). $n = 5$ independent networks.

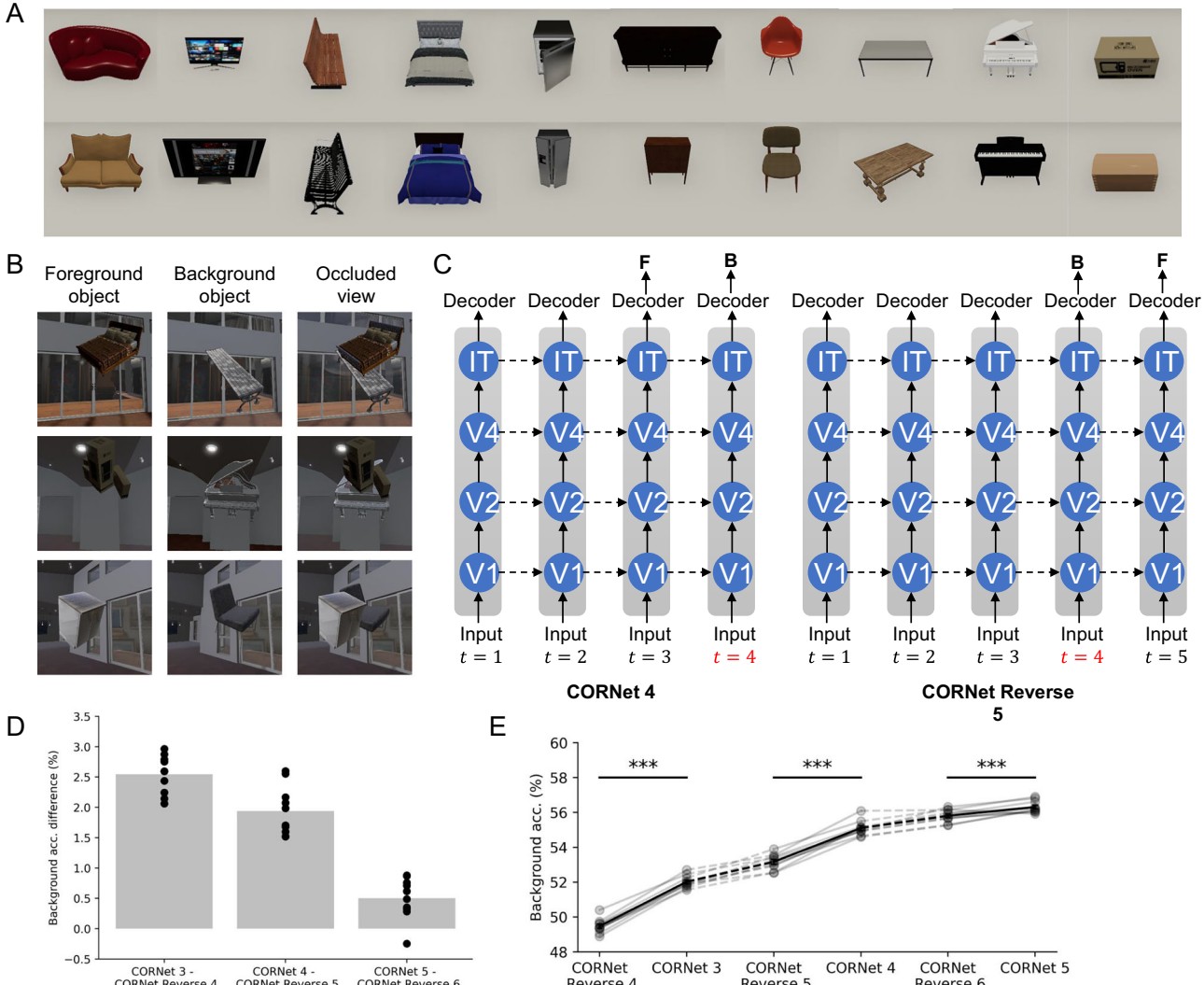

**Fig. 4 | "Explaining-away" computation in a richer dataset and more realistic models. A** The 10 object classes in the 3D occlusion dataset. **B** Example images from the 3D occlusion dataset. Each image involves one foreground object (occluding) and one background object (occluded). **C** The network architecture used in the experiments. Object class prediction was performed in the last two timesteps. In the normal order setting (left), the model is asked to predict the foreground class first and then background class. In the reversed order setting (right), the prediction order is swapped. The model predicts the background object class first. **D** Difference of model performance on background object recognition task between normal and reversed order settings. Each dot is an independently trained model ($n = 10$). Results are shown from left to right across different computational depths. **E** The performance on the background object recognition task under different model settings and computational depths ***$p < 0.001$, two-sided paired Student's $t$ test ($n = 10$).

predicts the background object at the second timestep. We presented the network with a multi-object image, and replaced its hidden state after the first time step with the internal state from an identical network, which was presented with a modified image in which the background object remained exactly the same, but the foreground object was modified. Crucially, the modified foreground object had either a different category, orientation, or both. If explaining-away arises from foreground-object information being fed back through recurrent connections to facilitate background recognition, then such a mismatch in internal states should hamper the network's ability to classify the background object correctly. Consistent with this idea, we found that perturbing the hidden state significantly impaired background-object recognition (Fig. 5A, B, left). In contrast, the same perturbation exerted little effect on foreground-object recognition (Fig. 5A, B, right), consistent with the intuition that a fully visible foreground object does not benefit from additional feedback about the background. This effect was most pronounced when we replaced the hidden state in lower layers of the network, suggesting that recurrent information about the occluder is integrated at lower-level feature

representations. Importantly, we note that there was no significant change in the distribution of occlusion levels due to the modified foreground objects (Fig. 5A bottom).

In addition, we investigated whether feedforward networks can similarly benefit from recognizing the foreground object before the background object. We introduced two new feedforward variants, FF Seq and FF Seq Reverse (Fig. 5C), which produce separate predictions for foreground and background objects at different layers. In FF Seq, the network predicts the foreground category at an intermediate layer and then continues to predict the background category in a subsequent layer. In FF Seq Reverse, the order is reversed: the network predicts the background category first and the foreground category afterward. Crucially, as in our recurrent experiments (Fig. 3D), to ensure a fair comparison, we kept the computational depth at which the background object is classified the same across these two variants.

We built FF Seq and FF Seq Reverse versions of the three feedforward architectures (FF, FF Wider Taller, and FF WS Wider Taller). For example, in the original FF model, both objects are classified after three convolutional and one fully connected layer. In comparison, in

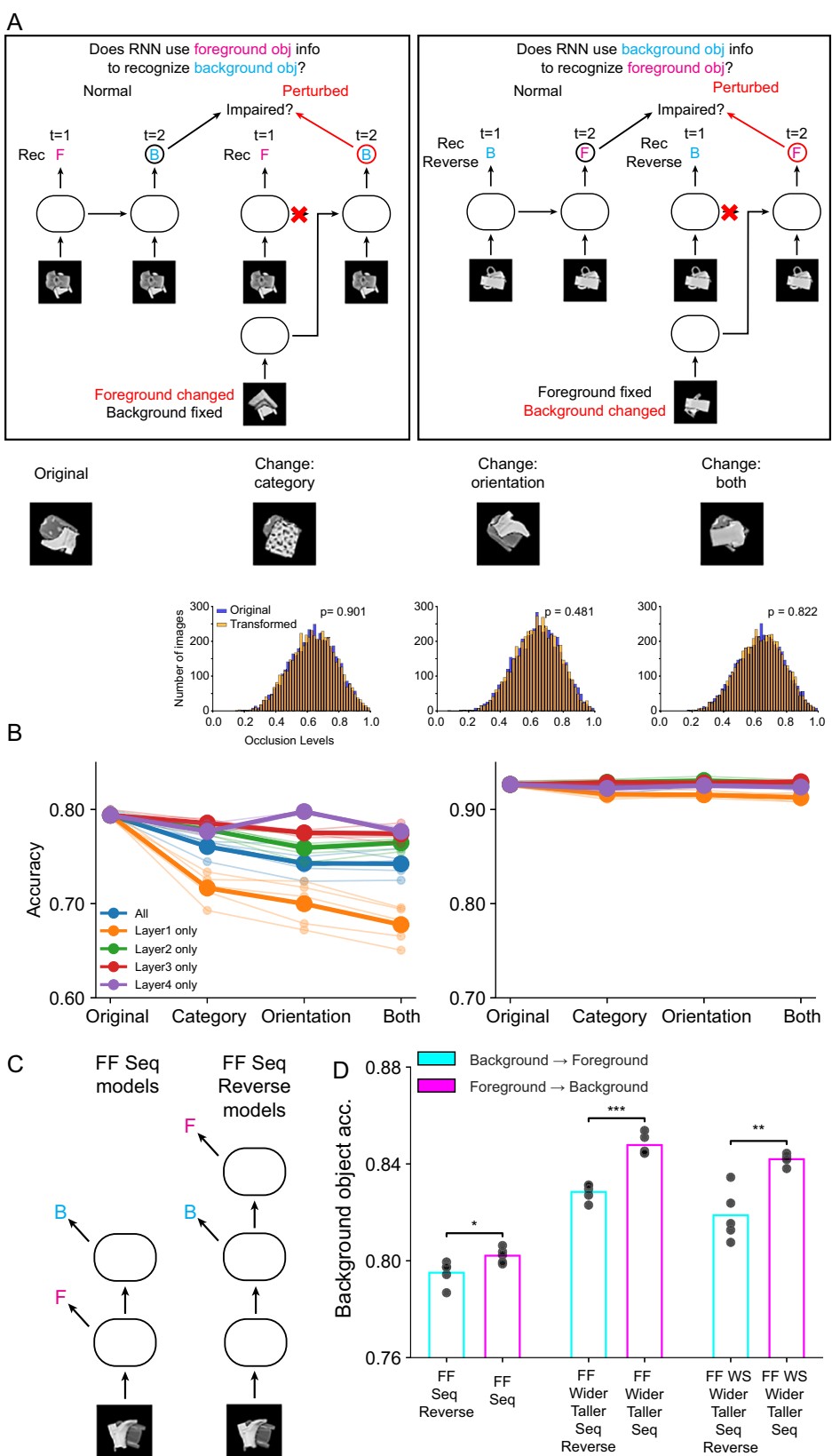

the FF Seq version, the foreground object is first classified after two convolutional layers and one fully connected layer branching out from the second convolutional layer, and then the background object after three convolutional layers and one fully connected layer. Conversely, in the FF Seq Reverse version, the background object is classified after three convolution layers and one fully connected layer branching out

from the third convolution layer, and then the foreground object after four convolution layers and one fully connected layer. Across all three base architectures, FF Seq outperformed FF Seq Reverse in recognizing the background object (Fig. 5D). This finding indicates that performing foreground-object recognition before background-object recognition in feedforward networks can recapitulate the explaining-

**Fig. 5 | Perturbation experiments and sequential versions of feedforward models demonstrate the role of sequential processing in explaining-away.** **A** Schematic of perturbation experiments in recurrent networks. Left: To test whether the network uses foreground-object information to recognize the background object, we replaced the hidden state at time step 1 with that from the same model that received an image in which only the foreground object was changed (category, orientation, or both), while the background object remained unchanged. Right: A corresponding experiment testing whether the network uses background information to recognize the foreground object. Bottom: Example stimuli for each transformation type and histogram of occlusion levels for original vs. transformed stimuli; occlusion level distributions did not differ significantly (two-sample Kolmogorov–Smirnov test without adjustments for multiple comparisons was used; change-category $p = 0.901$; change-orientation $p = 0.481$; change-both $p = 0.822$). **B** Accuracy of background (left) and foreground (right) object classification in the perturbation experiment when perturbation was applied to individual or all layers. Bold lines indicate average across model instances and lighter lines individual model instances ($n = 5$ networks per perturbation type). Background classification was impaired, especially when lower-layer hidden states were perturbed, whereas foreground classification was unaffected. **C** Schematics of feedforward sequential (FF Seq) and reverse-sequential (FF Seq Reverse) models, in which predictions for foreground and background objects are made at different layers. **D** Background-object accuracy across FF Seq and FF Seq Reverse models built on different base architectures. FF Seq models (foreground predicted before background) consistently outperformed FF Seq Reverse models (background predicted before foreground). Bars indicate mean accuracy across 5 networks; dots represent individual networks. *$p < 0.05$, **$p < 0.01$, ***$p < 0.001$, two-sided unpaired Student's $t$ test ($p = 0.029$, $p < 0.001$, $p = 0.0014$).

away effect observed in recurrent networks, suggesting that the advantage arises from leveraging information about the occluder rather than strictly requiring recurrence, although implementing this temporal ordering of foreground and background recognition is more natural in recurrent networks.

## Human psychophysics reveals behavior consistent with explaining-away

Having found evidence for explaining-away in both low-resolution models with basic architectures and high-resolution models designed to emulate the primate visual system, we next tested whether the same effect is found in human behavior. In principle, a direct analog of our computational experiments can be performed behaviorally. Participants would perform three task variants, the first in which they report both foreground and background object categories, the second in which they only report the background object's category, and the third in which they report the foreground object's category. However, this approach is problematic since, even when a subject is not instructed to report the foreground object's category, they may internally—even subconsciously—process the foreground object to facilitate recognition of the occluded object. This contrasts with artificial neural networks, for which we can specify the objective function. To avoid this issue, we performed an alternative experiment where we compared performance across two trial types in a background-object-only recognition task. Each trial consisted of three epochs: a single object image presentation epoch, followed by an occluded image (occluding and occluded object) presentation epoch, followed by a response epoch (Fig. 6A). In occluder object trials, the single object was identical to the foreground object in the occluded image. In random object trials, the single object was a randomly different object. If the explaining-away hypothesis is correct, in occluder trials, processing the single object in the single image epoch should allow it to be better explained away when the occluded image is shown, and therefore the background object to be more easily recognized than in random trials.

We performed the behavioral task online. In our hands, less than a fourth of participants (41/182) passed quality controls such as randomly appearing attention checks and minimal response time thresholds. As predicted by explaining-away, performance was significantly higher in the occluder object condition than the random object condition (mean 0.765 for occluder object trials and 0.725 for random object trials, $p < 1e-4$, paired Student's $t$ test, Fig. 6B). In addition, participants responded quicker in the occluder object trials than random object trials (mean 0.935 s compared to random object trials' 0.974 s, $p < 1e-11$, paired Student's $t$ test, Fig. 6C). Together, these results indicate that human subjects make significant use of explaining-away the occluder when tasked with identifying occluded objects.

The performance and reaction time effects were robust to the level of occlusion. We split trials into three equal-sized bins: low occlusion, medium, and high, and compared response time and performance across them. Performance was higher for occluder object trials than random item trials in all three occlusion levels (low occlusion $p < 0.0008$, medium occlusion $p < 0.004$, high occlusion $p < 0.02$, paired Student's $t$ test, Fig. 6E) and reaction times were shorter (low occlusion $p < 1e-5$, medium occlusion $p < 0.002$, high occlusion $p < 1e-4$, paired Student's $t$ test). We note that, in terms of the overall effect of occlusion level itself on behavior, for both response time and performance, there was no significant difference between low and medium occlusion bins, only between those bins and the high occlusion bin (response time: low and medium occlusion bins for occluder trials $p < 0.151$ and for random trials $p < 0.9$, low and high occlusion bins for occluder trials $p < 1e-5$ and for random trials $p < 1e-6$, medium and high occlusion bins for occluder trials $p < 0.0005$ and for random trials $p < 1e-4$; performance: low and medium occlusion bins for occluder trials $p < 0.774$ and for random trials $p < 0.834$, low and high occlusion bins for occluder trials $p < 1e-4$ and for random trials $p < 1e-5$, medium and high occlusion bins for occluder trials $p < 1e-4$ and for random trials $p < 0.002$, paired Student's $t$ test).

Taken together, these results demonstrate the importance of pre-processing the occluding item for identifying the occluded item. When human subjects were given greater opportunity to pre-process the occluding item, they were able to more easily explain it away, resulting in better performance in identifying occluded items. This behavioral effect supports our hypothesis that explaining-away occluding items is an important component of processing complex and layered visual scenes.

In conclusion, our modeling and human behavioral results suggest that recurrent processing is capable of explaining away missing pixels of the occluded object by first processing the occluding object. These findings pave the way for testing the explaining-away hypothesis in neural recordings.

## A biologically inspired model can perform an explicit form of explaining-away

Our model comparison revealed that, when training recurrent networks to solve occluded object recognition, explaining-away arises naturally and without being explicitly imposed. However, this by itself does not reveal the mechanisms or processes by which recurrent connections implement explaining-away. A recent study examining V4 representations[19] suggested more specific properties of recurrence that enhance the processing of occluded images. Namely, over time, the selectivity of V4 neurons to occluded shapes is enhanced, likely through feedback, from ventrolateral prefrontal cortex (vlPFC). We therefore hypothesized that the explaining-away suggested by the results of our previous sections may take the form of feedback inputs whose effect is to change—over time—the initial representation of an occluded object to make it more similar to its representation had it not been occluded, thus facilitating downstream recognition. We refer to this process as reconstituting the unoccluded representation. We sought to test the feasibility of reconstituting a high-dimensional

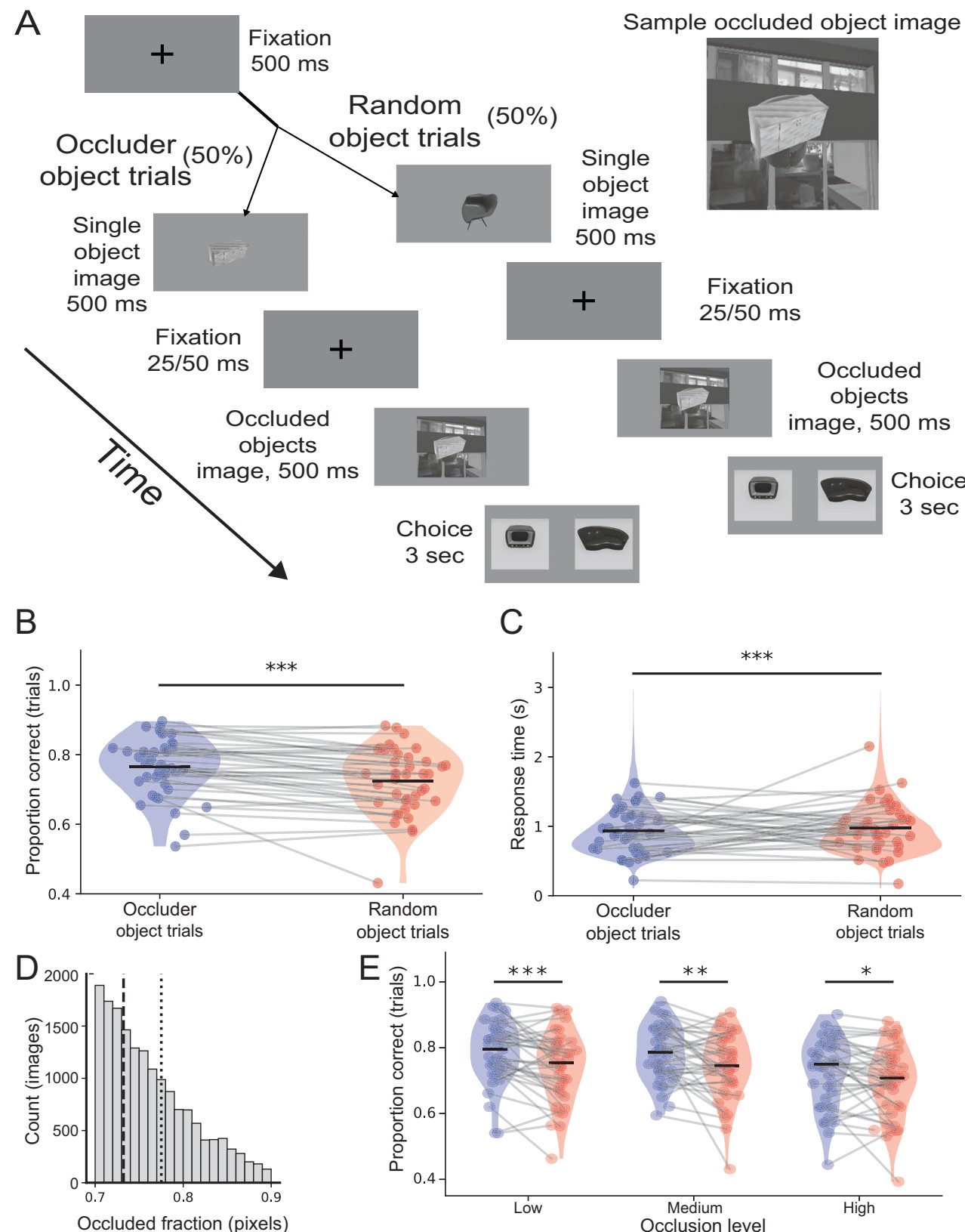

representation of an occluded object, such as those in V4, using a convolutional neural network.

We designed our model, which we refer to as Reconstitution Network (Recon-Net), to reflect some key qualitative features of the V4-vlPFC interactions suggested by Fyall et al.[19]. First, we assume that representation reconstitution is primarily carried out by a distinct neural circuit (vlPFC), although recurrent processes within the visual cortex may, to some extent, also be involved. This was motivated by the temporal relation between V4 and vlPFC selectivity. Selective V4 neurons had two distinct response peaks, and peak responses of many vlPFC neurons occurred between them, suggesting an early vlPFC-independent response in V4 followed by a later vlPFC-influenced

**Fig. 6 | Human psychophysics experiment supports explaining-away.**
**A** Schematic of a psychophysics task. Task trials were split between those in which the single object shown early in the trial was the same as the occluding item in the multi-object image (occluder object trials) or was a randomly chosen alternate item (random object trials). **B** Performance on occluder object trials was higher than random trials, consistent with explaining-away. Each dot corresponds to one participant, one dot per condition, occluder object trials (blue) and random object trials (red). Occluder mean = 0.765; random mean = 0.725; ***$p < 0.001$ two-sided paired Student's $t$ test ($n = 41$). **C** Response times from all subjects pooled into two conditions: occluder (blue) and random (red). Random 1% of trials shown as data points—each point is one trial. Occluder trials were slower than random trials. Occluder mean = 0.935 s; random mean = 0.974 s; ***$p < 0.001$ two-sided paired Student's $t$ test ($n = 41$). **D** Distribution of occlusion levels across all trials from all subjects. Calculated as the number of pixels of the occluded item obscured by the occluding item. Dashed lines split the distribution into low, medium, and high occlusion bins. Each bin has an equal number of trials. **E** Performance difference between occluded object and random object trials was present across all occlusion levels. Each dot corresponds to a subject, with one dot per occlusion level and trial type (i.e., 6 dots per participant–one per violin plot). Low occlusion occluder mean = 0.786, random mean = 0.742, $p < 0.001$; medium occlusion occluder mean=0.783, random mean=0.740, $p = 0.004$; high occlusion occluder mean=0.723, random mean=0.693, $p = 0.014$; ***$p < 0.001$, **$p < 0.01$, *$p < 0.05$ two-sided paired Student's $t$ test ($n = 41$).

response. In our simplified picture, vlPFC reconstitutes a V4 representation corrupted by occlusion and then sends the reconstituted representation back to V4. Second, we assume that vlPFC is given a representation of the foreground object that is well disentangled from the background object. This is suggested by the existence of vlPFC neurons that are highly selective to the occlusion level of the stimulus but do not exhibit much selectivity for the occluded object. The last feature we incorporated in our model is that vlPFC has object-selective neurons or object-selective population activity, implying that it has a neural representation of visual objects. It seems reasonable to suppose such a representation, given the shape-selective neurons observed in vlPFC.

Recon-Net takes as its input images of an occluded object and a foreground object and then predicts an occlusion-free (reconstituted) image of the background object (Fig. 7A). The occlusion-free output image represents the reconstituted V4 representation, and the foreground object image represents a disentangled representation of the occluder, which we assume is given to vlPFC as mentioned above. As discussed in the previous subsection, a representation of the occluder is an essential input needed to explain away occlusion. We note, however, that a qualitatively similar picture arises even if we train networks without access to an explicit disentangled occluder representation, possibly because they may extract a representation of the occluder from the occluded object image on their own (Fig. S2E). The model consists of two key parts: an inference module that predicts an occlusion-free latent representation of the background object from the two inputs and a generative module that decodes the latent representation into the corresponding raw-pixel image output (Fig. 7A).

The intuition for this dual architecture is that it will allow the inference module to improve its current estimate for the latent variable by synthesizing an occlusion-free image and evaluating its likelihood, given the input images. Such an operation is a concrete, explicit form of explaining-away and is closely related to the notion of "analysis by synthesis", whereby the brain possesses a generative model of the world, and perception is the search for the best explanation of incoming stimuli in terms of this generative model[40–46]. Phrased differently, one can view this dual architecture as performing a context-dependent computation with the occluder being the context in which the occluded image is interpreted. This is in line with the computational role typically associated with PFC in guiding flexible, context-dependent behavior[47]. Such a context-dependent computation is necessary to explain away occlusion.

The generative module of Recon-Net is a generative adversarial network (GAN), a state-of-the-art deep generative model[48] that was trained on original, unoccluded FashionMNIST images of the four classes that form the basis of our study (see "Methods" and Fig. S2A). The inference module iteratively predicts the latent representation based on the inputs and its previous prediction from the inputs (both the predicted latent representation and the image generated from it), refining its prediction over five time steps (see "Methods" and Fig. S2B).

Over the course of our experimentation with Recon-Net we discovered that a crucial factor for its performance was the degree to which the background object was spatially transformed. Though Recon-Net was able to deal with foreground object translation and rotation, its performance substantially deteriorated when the background object was randomly translated and rotated from its pose in the FashionMNIST dataset. This contrasted strongly with the object classification tasks we previously considered, where deep neural networks accurately identified highly transformed occluded objects. This confirms our intuition that reconstituting a high-dimensional representation of occluded objects may be a significantly more challenging task than just classifying them. To prevent this difficulty from becoming a confound, we report results when the background object was not spatially transformed, and only the foreground object was rotated and translated (a dataset we refer to as "original" pose, see "Methods"). In addition, we considered a more complex architecture that does work with the fully translated and rotated images. In this architecture, the spatial transformation is inferred and reversed by separately trained networks (see "Methods" and Fig. S2D).

If Recon-Net effectively reconstitutes images, then a decoder that has never encountered occluded representations should still be able to correctly classify a reconstituted representation of an occluded object. Thus, we quantified Recon-Net's performance by measuring the accuracy of a classifier that never encountered occluded images when it was given the output of Recon-Net. This way of quantifying representation reconstitution provides a more meaningful measure of how well semantic features of the occluded object are recovered than more naïve measures such as the pixel-space Euclidean distance, which would be strongly affected by uninformative differences such as pixel values in the black background of images.

We found that decoders that have never encountered occlusion are nevertheless able to classify occluded background objects when they are given images that have been reconstituted by Recon-Net, but not when given the original occluded images (Fig. 7B). This suggests that Recon-Net indeed effectively reconstituted the images. In fact, the level of accuracy achieved was almost as high as that of classifying occluded images by decoders trained directly on occluded images. This suggests that the reconstitution by Recon-Net was nearly as effective as an end-to-end training for occluded object recognition. This high level of accuracy was only slightly improved when we used a decoder trained directly on Recon-Net outputs (Fig. S2C). We found qualitatively similar results when the decoder had a recurrent architecture (Fig. S2C). We note that the absolute levels of accuracy presented here are different from those presented in the previous section due to the different nature of the background image (not rotated or translated).

We observed that most of the reconstitution of representation occurs in the first few time steps (Fig. 7C). However, in many cases, refinement continued to occur over time. Visual inspection suggests that the initial representation may be imperfectly reconstituted due to the occluder, but, over time, the global context assists in converging to the correct object (Fig. 7C). In some cases, Recon-Net first incorrectly inferred the category of the occluded object before rectifying its

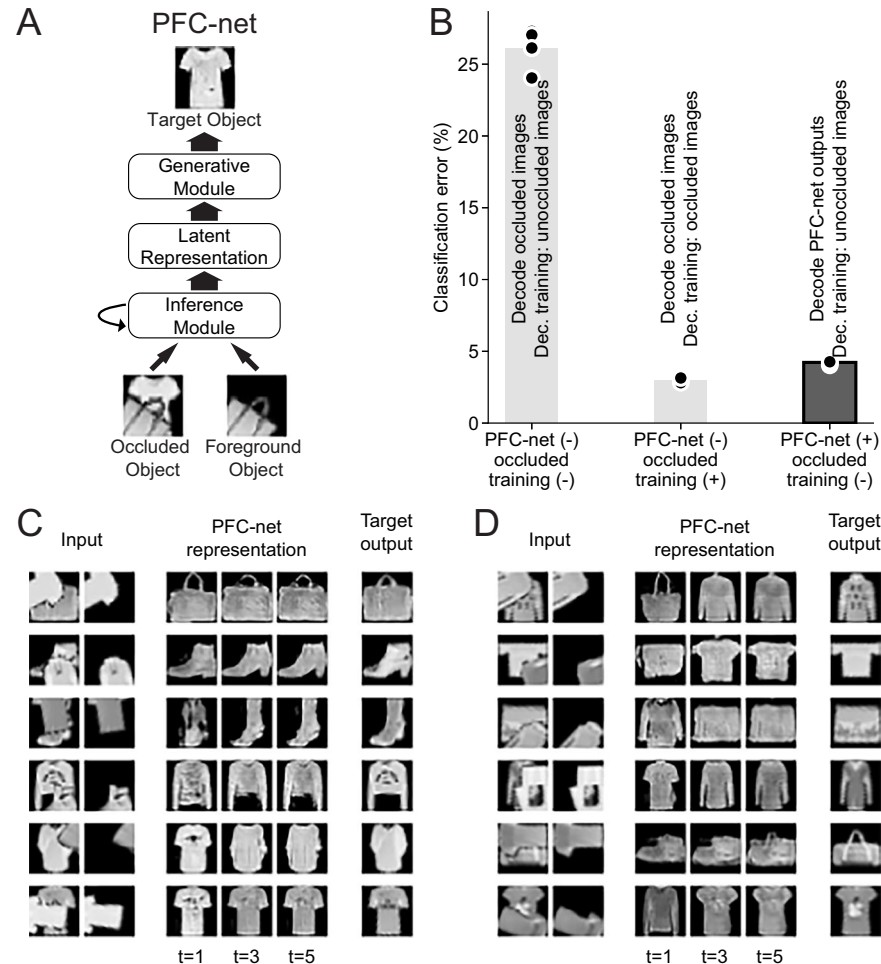

**Fig. 7 | Representation reconstitution by Recon-Net. A** A schematic of Recon-Net. Recon-Net consists of an inference module and generative module. The inference module takes as input images of an occluded object and a foreground object, and predicts an occlusion-free latent representation of the occluded object through recurrent iterations over five time steps. The generative module maps the predicted latent representation to the corresponding raw-pixel image. **B** Recon-Net allows successful classification of occluded images by decoders not trained with occlusion. Each bar corresponds to a different combination of network architecture and training. Height of bar indicates the average test classification error of predicting the identity of the background object, and black dots indicate individual independently initialized networks ($n = 5$). Left bar: performance of decoders trained on unoccluded images when classifying occluded images. Middle bar: performance of decoders trained on occluded images when classifying occluded images. Right bar: performance of decoders trained on unoccluded images when classifying occluded images first processed through Recon-Net. The performance of a decoder that has never seen occlusion is almost as good as a decoder fully trained with occlusion when given Recon-Net output. **C** Recon-Net tends to refine its representation over time. Left columns indicate the two inputs: original occluded object image and foreground object image. Middle columns show the images corresponding to Recon-Net's representations at three time points. Right column indicates the target output for Recon-Net, i.e., the image of the background object if unoccluded. **D** Similar to (**C**), but examples where Recon-Net initially gets the category of the occluded object incorrect and then rectifies it over time.

representation over time (Fig. 7D). Such time-dependent improvement of Recon-Net representations provides an interpretable explanation of how recurrent computations can facilitate the recognition of occluded objects. This is also reminiscent of delayed object-selective responses to occluded stimuli mentioned above[13–19]. We note that recurrent computations in Recon-Net can be viewed as being performed by top-down connections. We designed Recon-Net in this way because, in our opinion, this design, including the "top-down connections", made the explaining-away phenomenon more human-interpretable. This is not in contradiction with the previous results in Fig. S1E, which did not show that no top-down architectures are high-performing, just that their performances are more sensitive to details of top-down connections.

Having trained decoders on unoccluded images and tested them on occluded images, we performed the converse test, which yielded surprisingly asymmetric results. Across architectures, classifiers trained only on heavily occluded images (occlusion level between 50 and 75%) generalized unexpectedly well to unoccluded images, in fact

performing better in this generalization than on the images they were trained upon (Fig. S2F). This was surprising given the large shift in visual features caused by occlusion. Indeed, earlier in this subsection, we observed that neural networks do not generalize well from unoccluded to occluded images, resulting in about 20% drop in accuracy (Fig. 7B). The fact that neural networks generalize well in the opposite direction suggests that features they learn from heavily occluded images are not specialized to heavy occlusion regimes but generalizable to lighter occlusion levels, while the converse is not true.

### Effect of occlusion on internal representations of neural networks

To better understand the asymmetry of generalization across different occlusion levels, we compared the internal representations of networks trained on occluded and unoccluded images. We quantified the representational similarity between networks using a linear mapping approach. To be clear, we emphasize that it differs from Representational Similarity Analysis (RSA), another commonly used method to

investigate representations in deep neural networks[49]. Our method involves performing linear regression from activations of one layer in one architecture, the source model, to another layer of a second architecture, the target model[6,25] (Fig. 8A, see "Methods"). Specifically, we trained a set of five randomly initialized networks on occluded images and another set of five networks on unoccluded images. Then, for each pair of a source and target neural network type, we trained a linear regression from the source model's activations to those of the target model at each layer; we did this separately for occluded and unoccluded images. We then quantified representational similarity by the correlation between the predicted and target activations of the linear regression across held-out images, averaged over pairs of randomly initialized instances. Below, we used the non-occlusion-trained architecture as the source model and the occlusion-trained architecture as the target model, but the results are qualitatively the same if we reverse the source and target models.

Interpreting the representational similarity measure requires establishing a noise ceiling, which estimates the highest possible value for the measure. Previously[6,25], a noise ceiling for similarity was calculated by the correlation across input images between trial-average responses from two disjoint sets of trials. The noise (and its ceiling) originates from trial-to-trial variability of neural activity in the primate visual cortex. Since our neural networks are deterministic functions of the input, there is no such trial-to-trial variability. Instead, we used a noise ceiling that originates from the instance-to-instance variability of the random initialization of the network weights. Thus, our noise ceiling was defined to be the average representational similarity between distinct instances of a target neural network type. In a sense, it quantifies how stereotypical learned representations are given the randomness at initialization.

We compared the representations of occlusion-trained and non-occlusion-trained classifiers across our three classes of network architectures–feedforward, recurrent and top-down recurrent, the first two of which were identical to the decoders used in the previous subsection. They were also identical in form to the ones used in the first subsection, but they had more units in convolutional layers to achieve a sufficiently high performance (>70% classification accuracy) on the dataset used for Recon-Net. Specifically, the feedforward architecture had 64, 128, and 128 output feature maps in the three convolutional layers, and 512 output units in the fully connected layer. Similarly, the recurrent and top-down recurrent architecture had 64, 128, and 128 output feature maps in the three ConvLSTM layers and 512 hidden units in the LSTM layer. The top-down recurrent architecture's feedback connection was identical to that of T-D Rec v2 in the first subsection, the best-performing top-down recurrent variant. For fully connected and LSTM layers, we used all output unit activations as targets of the linear regression. For convolutional and ConvLSTM layers (which have a much larger number of output unit activations), we randomly sampled 256 units from each such layer of a target architecture to calculate representational similarity (see Methods). The images used had both foreground and background objects transformed (see "Methods"). Results in the original pose condition (only foreground image transformed, see "Methods") are provided in Fig. S6.

We found that the similarity of internal representations in the feedforward network strongly depended on the type of test images used (Fig. 8B). For unoccluded images, occlusion-trained networks had similar representations to non-occlusion-trained ones. In contrast, for occluded images, we find significant representation dissimilarity between networks trained with and without occlusion. While it is not surprising that networks trained with or without occlusion are dissimilar, as they were trained differently, we emphasize the non-trivial result that, despite the difference in training, they appear almost identical for unoccluded images. This highlights the importance of the test input one uses to evaluate the similarities of neural networks.

In other words, the fact that two neural networks share similar representations for one set of inputs does not necessarily imply they will continue to do so on another set of inputs. In addition, we found that the intermediate representations of networks trained with or without occlusion tended to be more distinguishable toward higher layers in a gradual manner (Fig. 8B). This tendency suggests that the presence of occlusion during neural network training primarily affects weights at higher layers. The results for the recurrent networks displayed similar features (Figs. 7C and S4). As in the feedforward case, the occlusion-trained and non-occlusion-trained classifiers had significantly more distinct representations for occluded images than for unoccluded images. Also, in parallel with the feedforward case, the representations became progressively more distinguishable toward higher layers. Counter to our expectation, however, the distinguishability of the representations did not tend to change as much over time.

The top-down recurrent architecture exhibited similar trends (Fig. S5). Our expectation was that top-down feedback might make the representations of the occlusion-trained and non-occlusion-trained classifiers more distinguishable at lower layers. We found a small yet noticeable enhancement in the distinguishability of the representations at lower layers, including the first layer that directly received top-down feedback. The fact that the distinguishability in the lower layers was not as high as higher layers from which top-down feedback is received can be explained by the top-down feedback mostly transmitting components of the representation that are more similar across occluded and non-occluded training.

In addition to occlusion, we also examined the effects of spatial transformations, such as translation and rotation, on representations learned by neural networks (Fig. S8). In parallel with the above results, representations of transformation-trained and non-transformation-trained classifiers tended to be more distinguishable on transformed images and in higher layers (Fig. S8A, C). Also mirroring the case of occlusion (Fig. S7), the generalization from transformed images to non-transformed images was easy, but the generalization in the reverse direction was not (Fig. S8B, D). In particular, we note that in our setup, the generalization from unrotated to rotated images seems significantly harder than that from unoccluded to occluded images. Though controlling for difficulty across distinct types of generalizations is challenging, systematically studying neural networks' ability to generalize across distinct types of variations in object appearance and how their representations are affected by such variations is an interesting future direction.

Overall, our results show that the choice of inputs is crucial in comparing computations performed by different neural networks. The inferred similarity of their computations depended on the inputs used to probe them, and therefore, differences in representations may be best dissected through judicious choice of inputs that reflect their training history. Moreover, the fact that training on occluded images only weakly changed representations of unoccluded images suggests that this type of solution may facilitate downstream processing by restricting representation shifts required by learning.

## Discussion

There has been significant progress towards understanding cortical activity in the context of recognizing single, unoccluded objects[6,8,25–27], but occluded object recognition has remained more elusive. One apparent tension between recent studies of single object recognition and occluded object recognition is the utility of recurrent computations: even though deep feedforward networks have achieved state-of-the-art performance in various visual tasks, including single object recognition, multiple studies claimed recurrent networks perform better at occluded object recognition. Resolving this tension and better understanding the significance of recurrent computations in the

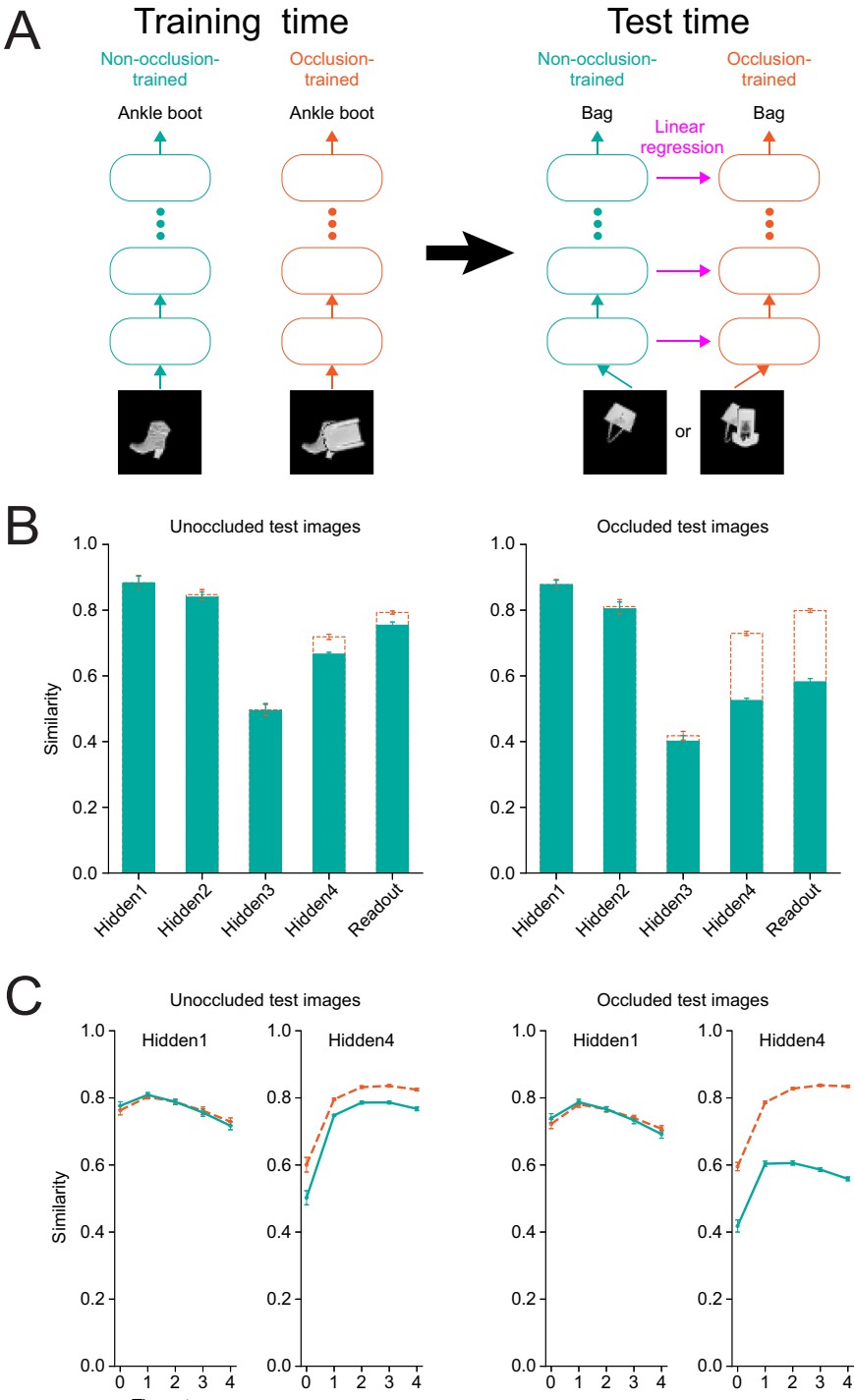

**Fig. 8 | Degree of representational similarity strongly depends on the type of images in the test dataset. A** A schematic for computation of representational similarity. We trained neural networks of the same architecture to classify either unoccluded images ("non-occlusion-trained" network) or occluded images ("occlusion-trained" network). We then presented identical unoccluded or occluded held-out test images to both networks and compared their intermediate representations at each layer. **B** Representational similarity of occlusion-trained feedforward networks across layers. Dotted orange bars show the average representational similarity of occlusion-trained networks to other occlusion-trained networks. Filled green bars show the average representational similarity of occlusion-trained networks to non-occlusion-trained networks. The error bars indicate the standard deviations across different pairs of randomly initialized instances of the networks ($n = 25$ pairs for the filled green bars, and $n = 20$ pairs for the dotted orange bars, excluding the pairs of the same instances) (*Left*) similarity when assessed on unoccluded images. (*Right*) similarity when assessed on occluded images. **C** Representational similarity of the first and fourth hidden layers of occlusion-trained recurrent neural networks. Dotted orange lines indicate the average representational similarity of occlusion-trained networks to other occlusion-trained networks across time steps. Solid green lines indicate the average representational similarity of occlusion-trained networks to non-occlusion-trained networks. The error bars indicate the standard deviations across different pairs of randomly initialized instances of the networks ($n = 25$ pairs for the solid green lines, and $n = 20$ pairs for the dotted orange lines, excluding pairs of the same instances). (*Left*) similarity when assessed on unoccluded images. (*Right*) similarity when assessed on occluded images.

context of occluded object recognition were primary motivations for our present work.

Only a few previous studies used recurrent and feedforward models with comparable architectures and were fully trained on occluded images, both essential for robust comparison[33,50]. However, the dataset used in one of them consisted of computer-generated digits that lacked intra-class variability in shape, which may not reflect the full complexity of occluded object recognition. In addition, the models considered in both studies have only two hidden layers (except for one feedforward model with four hidden layers, which does not have a recurrent counterpart), which may have been insufficiently deep for their results to generalize to deeper networks that are more comparable to visual cortex[25]. Other previous studies employed deeper networks and more realistic images and are thus less prone to the above concerns. However, their experimental design was not well-suited to comparing task performance across network architectures. For example, feedforward models were either not trained on occluded images[17] or the feedforward and recurrent models did not have comparable architectures[18]. Moreover, in these studies, occlusion was typically generated by simple geometric masks, which may cause their visual recognition tasks to be qualitatively different from those occurring in the real world, where occlusion is often caused by visually complex objects. By addressing these shortcomings, our approach facilitates fairer and more thorough comparisons across different architectures.

Our results demonstrate that feedforward architectures are not intrinsically worse at occluded object recognition, which is to be expected as recurrent networks are a subclass of feedforward networks (i.e., feedforward networks with shared weights). At the same time, we found that explaining-away--recurrent networks feeding the representation of an occluding object back to improve classification of the occluded item--arises naturally in our experiments and yields a modest increase in performance. This was true both for a low-resolution dataset based on fashionMNIST processed by a range of straightforward architectures as well as for a more photorealistic dataset processed by a model designed to emulate primate object recognition. Human behavioral experiments using the more realistic dataset found evidence suggesting that such explaining-away computations occur in human object recognition as well. Moreover, inspired by recent experimental results on the role of V4-vlPFC interactions in assisting occluded object recognition[19], we examined the feasibility of a more specific form of explaining-away: whether it is computationally viable to reconstitute a high-dimensional representation of occluded stimuli through explaining-away. This is a much more computationally demanding task than merely identifying the category of an occluded object, as the high-dimensional representation may encode richer information about the object, such as its texture and detailed shape. Nonetheless, our biologically inspired architecture, Recon-Net, was able to reconstitute occluded images in a manner that reveals recurrent feedback's potential role in bringing global context into consideration, thereby explaining-away occlusion. Lastly, we studied how occlusion may alter the internal representations of a neural network trained to perform an object recognition task. This was an important question to study, as recent work that found analogies between primate visual cortex and artificial neural networks relied on comparisons of such internal representations.

### Controlled comparisons across convolutional models of occluded object recognition reveal explaining-away by recurrent connections

Our results suggest a nuanced view of the significance of recurrent connections for the classification of occluded images. On one hand, unlike some previous studies[17,18,33], we do not find a striking difference in the level of performance achievable on occluded classification between recurrent and feedforward architectures, so long as the number of units and parameters was not controlled. On the other

hand, for the same number of parameters, recurrent architectures offered far better performance.

We believe that the utility of recurrent connections for occluded object recognition lies in their capacity to facilitate explaining-away. We found that this feature emerged in our results even when networks were trained end-to-end in a fully agnostic and unstructured way. Importantly, its emergence depends on the temporal ordering of task demands--namely, when the network is required to recognize the occluder prior to the occluded object. Thus, our findings do not suggest that recurrent networks universally perform explaining-away, but rather that they are capable of doing so under conditions where such a computation is facilitated by the task structure.

Targeted perturbation experiments in our recurrent network, in which we replaced the hidden state after recognizing the occluder with that from a network viewing a different foreground object, significantly impaired the ability to classify the background object. This result strongly suggests that recurrent feedback actively propagates information about the occluder to facilitate background-object recognition, particularly in lower-level feature representations. Nevertheless, we also show that feedforward networks can achieve a similar effect if their computations are explicitly organized in a sequential manner (e.g., recognizing the occluder first in an intermediate layer and the occluded object in a subsequent layer). Hence, while recurrence may offer a more natural means of implementing explaining-away, the key factor is leveraging information about one object to improve recognition of the other. We note that, the improved performance when the occluded object is reported after the occluder--or, more generally, when the more complex object recognition task is performed after the simpler one--is likely relevant for understanding the delayed response to occluded and other difficult stimuli in the visual cortex[13–18,34,51].

Although our analyses highlight how explaining-away emerges in both recurrent and suitably structured feedforward networks, the detailed mechanisms remain only partially understood. Perturbation-based approaches, like the hidden-state replacement used here, help reveal how information about the occluder feeds back or forward to facilitate background recognition, but a complete mechanistic account of these interactions is still lacking. As interpreting the computations performed by deep networks remains a formidable challenge[52], future work is needed to dissect how feedback or layer-to-layer gating supports explaining-away at the representational level. Understanding these processes more fully could ultimately clarify the broader principles by which biological and artificial systems integrate multiple objects and perform robust recognition under occlusion.

### Human psychophysics demonstrates explaining-away in the visual system

Our behavioral experiment in human subjects provides direct evidence that the visual system capitalizes on information about the occluder to facilitate recognition of occluded objects. Participants performed significantly better and faster when they were pre-exposed to the foreground object, even though the task required them to identify the occluded background object. This improvement is consistent with an explaining-away computation--having prior knowledge of the occluder can allow the brain to account for its effects and thereby more readily recognize the occluded object.

One question that arises is how these findings might relate to well-known object priming effects, wherein prior exposure to an object can bias or speed its subsequent processing[53]. In principle, priming can manifest in at least two ways: (i) a bias toward reporting the primed object and (ii) faster or more efficient processing of the primed object once it appears. However, neither of these straightforwardly explains our primary finding. First, a bias toward the primed (foreground) object would be counterproductive--participants needed to report the background object. Indeed, if classic priming were the dominant

factor, one would expect participants to be more likely to mistakenly choose the primed foreground in ambiguous situations rather than correctly identify the background. Such an effect would hurt performance instead of helping it. Second, even though priming the foreground object could speed its processing, it would not trivially confer an advantage on recognizing the different, unprimed background object, unless some use of the recognition of the foreground object is used to facilitate processing the background object, i.e., explaining away. Our observed benefit for identifying the background object thus suggests a process extending beyond conventional object priming.

### A biologically inspired model can perform an explicit form of explaining-away

Our model, Recon-Net, tested the feasibility of implementing explaining-away for complex objects in a high-dimensional representation space. Though the structure of the model was anchored in recordings from primates, our choice to have the model operate in raw pixel space was a compromise. Ideally, it would have been built upon V4-like representations of occluded stimuli rather than raw pixels. This compromise was prompted by how much is still unknown about the nature of the V4 representation, especially when visual stimuli are occluded; we contend that incorporating incomplete knowledge likely would have only introduced biases. We note, though, that our model is forward compatible: since the V4 representation enters as a given input, it would be straightforward to incorporate a more complete description and rebuild our model once such a description becomes available.

Using the raw-pixel representation had an important advantage as Recon-Net's intermediate representations were interpretable. Through visual inspection, we found that Recon-Net is capable of improving its representation of occluded objects over recurrent iterations. Although, as with other deep neural networks, it is hard to characterize the strategy Recon-Net employs to do so, we speculate that it might leverage its ability to compare its prediction and input: examining the difference between the predicted and actual appearance of the occluded object can help with inferring a more accurate representation at the next iteration.

While our modeling assumes that computations needed to reconstitute V4 representations of occluded stimuli occur in vlPFC, it is likely that this assumption holds only approximately. We expect neural computations that compensate for occlusion might also occur within the visual cortex itself (at least for low occlusion levels), given its ability to maintain invariance in neural responses (e.g., those of Inferior Temporal (IT) cortex) against variation in object appearance. More generally, it is possible that the reconstitution may happen outside vlPFC, or through interactions between vlPFC and other areas, and vlPFC activity may only (partially) reflect the result of the reconstitution. Thus, it will be interesting to study the interplay of occlusion-related computations in the visual cortex and in other areas, including vlPFC. Relatedly, a recent study showed, in the context of core object recognition, that a top-down feedback from vlPFC to the visual cortex is critical for recurrent processing of "challenging" images, for which IT takes additional time to develop selectivity[54]. This might hint at a broader role of recurrent computations in multi-regional circuits in solving challenging visual tasks[55].

The presence of occlusion renders the recognition of objects more difficult. Whether recognition is conceived of as a template-matching process or involves detecting the presence of a constellation of expected finer-scale features, occlusion leads to "atypical" values for these processes. We demonstrated one approach in Recon-Net, which reconstitutes a representation of the occluded object, potentially providing a more complete input to the recognition process. This is not the only option, and while we cannot enumerate all possibilities, a few stand out. First, the system may limit the scope of recognition to unoccluded portions of the image; such a process is reminiscent of the attention mechanisms used in transformer-based models and has been

explored in models supporting localized dynamics[56,57]. Alternatively, the system might maintain the same spatial extent but lower the recognition threshold to accommodate the mismatch caused by missing or corrupted features. Finally, if recognition involves competitive or winner-take-all dynamics, then none of the above mechanisms needs to perfectly "explain away" the missing information; they only need to resolve enough of the ambiguity so that the correct object emerges robustly from the recognition process.

### Effect of occlusion on internal representations of neural networks

Our comparison between representations of networks trained with occluded and non-occluded images revealed two issues. First, representations become more distinct as one progresses to higher layers. This suggests that good performance on occluded object recognition might be achieved by taking a network trained on non-occluded image datasets and re-tuning just the topmost weights of the network. This is potentially good news from the perspective of training a very deep network; training such networks requires large-scale datasets such as ImageNet[58], but no such large and diverse datasets with occlusion are easily available. Of course, the notion of only re-learning the topmost layers is not novel and is commonly performed in practice, but our results give support to why this is a sensible strategy, at least in the context of occluded object recognition.

The difference we observe in representation similarity across layers suggests that occlusion-induced modulations of neural representations are more likely to be found in higher layers of the visual cortex. We note, however, that it would be more interesting if this prediction were found to be false, that is, if one found significant occlusion-related modulations in lower layers of the visual cortex (contrary to our prediction); it might hint that learning in the visual cortex, or even its overall computational structure, has other fundamental mechanisms to process occlusion. Alternatively, it might also hint more generally that the visual cortex deviates from our current understanding of deep neural networks.

Second, our comparison revealed that networks with clearly dissimilar representations may nevertheless appear similar under some sets of stimuli. This is of particular importance for comparisons of artificial neural networks to real neural circuits, since we do not have access to the dataset used "to train" (so to speak) the real neural circuit. This finding, therefore, highlights the importance of not just the task and training set used in training an artificial neural network, but also of the dataset then used for comparison. More specifically, our results suggest that moving beyond using natural images of single objects will enrich comparisons of representations since features that may naturally arise when there are multiple objects, such as occlusion, may strongly affect the degree of similarity inferred. Finally, the ability of networks to change connectivity in a way that enhances responses to occluded images but does not change responses to unoccluded images raises the possibility of continual learning, where appropriate responses to one set of phenomena are learned after another without the latter learning erasing the former[59]. While it is not known whether animals develop the ability to recognize occluded objects later than non-occluded ones, similar processes may occur in expert learning of specialized categories[60–64] or perhaps even in learning before and after opening of eyes[65].

### Limitations of the present work

While our study provides a comprehensive evaluation of recurrent computations in occluded object recognition, it is important to acknowledge certain limitations. First, although we have demonstrated the utility of recurrent computations using both convolutional-LSTM-based models on the FashionMNIST dataset and more biologically realistic models (CORNet) on high-resolution images, it remains unclear how these results generalize to other neural network

architectures and more complex real-world scenarios. Additionally, the human psychophysics experiment provides initial evidence for the relevance of explaining-away computations in human object recognition, but further studies are necessary to confirm these findings across diverse tasks. Future work should explore the generalization of these findings to other network architectures and to more varied and challenging datasets.

In summary, we performed a comprehensive, realistic evaluation of the potential role of recurrent computation in identifying occluded objects. Our results resolve tensions between the purported computational capabilities of feedforward versus recurrent connections. They also suggest that recurrent networks are capable of leveraging information about the occluder to explain away occlusion thanks to their temporally structured computations, even when they were never explicitly trained to do so. Thus, we set a framework for further investigation of neural dynamics during visual processing of occlusion, how they are shaped by recurrent connections, and the underlying computation.

## Methods

### Controlled comparisons across convolutional models of occluded object recognition reveal explaining-away by recurrent connections

**Image dataset generation.** FashionMNIST[36] consists of 70,000 $28 \times 28$ grayscale images of fashion products on a black background. It contains 7000 images per each of its 10 classes and is split into a training set of 60,000 images and a test set of 10,000 images. For each set, we selected a subset of images that belong to 4 classes, T-Shirt/ Top, Pullover, Bag, and Ankle boot, yielding a reduced training set of 24,000 images and a test set of 4000 images. We generated 50,000 images of two overlapping FashionMNIST objects for a training set and 10,000 for a validation set from the reduced training set of FashionMNIST, and another 10,000 for a test set from the reduced test set. Labeling both the foreground and background objects, our dataset contains a total of 16 classes.

To generate an image of two overlapping FashionMNIST objects, we randomly sampled a pair of FashionMNIST images, and we zero-padded them symmetrically on all four sides to enlarge their size from $28 \times 28$ to $50 \times 50$ (0 corresponds to the black pixel and 1 to the white pixel). We then applied a random rotation and translation to each FashionMNIST image using the grid generator and bilinear sampler of Spatial Transformer Network[66]. Spatial Transformer Network regards both the source and target images as grids of pixels and uses normalized coordinates for the pixel locations such that those within the spatial bounds of the image are given x and y coordinates between −1 and 1. It parametrizes a 2-dimensional affine transformation from the source to target image by a $2 \times 3$ matrix acting on these normalized coordinates, which is a concatenation of a $2 \times 2$ matrix representing a linear transformation and a $2 \times 1$ matrix representing a 2-dimensional translation. To select a random rotation, we sampled an angle from the uniform distribution between −180 and 180°, from which the corresponding $2 \times 2$ rotation matrix was determined. To select a random translation, we first parametrized it by the radial distance and angle of the polar coordinates and sampled the radial distance from the uniform distribution between 0.1 and 0.15 and the angle from the uniform distribution between −180 and 180°, from which the corresponding $2 \times 1$ translation matrix was determined. While the rotation angle was sampled independently for each of the two objects, the radial distance and angle for the translation were sampled once for the pair, and the corresponding translation was applied to one of them, and the diametrically opposite translation was applied to the other. The lower and upper bounds for the distribution of the radial distance were chosen to make it easier to control the occlusion level of the resulting two-object image. The translations of the two objects were restricted to be diametrically opposite for the same reason.

To place the two spatially transformed objects in a single image, we then segmented them from their black backgrounds using an ad hoc algorithm. The algorithm first derives a rough segmentation mask by thresholding the input image at the pixel value of 0.01. The threshold value was chosen to ensure that the resulting mask contains the whole object while not including too many background pixels; if the threshold is higher than this value, the mask tends to miss dark regions of the object, and if it is lower, the mask tends to include too many background pixels that are not perfectly black. To get rid of a small number of background pixels included in the rough mask, the algorithm calculates the mean pixel value within the rough mask and thresholds the input image at 10% of this mean pixel value, producing a more accurate mask. The segmentation masks of the two objects from this algorithm were used to overlay one over the other, resulting in an image of the two objects overlapping with each other. Its occlusion level was defined as the ratio of the area of the overlap between the two masks to the area of the background-object mask.

Our dataset only includes such images whose occlusion levels range between 50 and 75%. We ensured that the distribution of the occlusion level in our dataset is uniform, such that it contains an equal number of images in each 5% bin in that range. We also ensured that, in each 5% bin, there exists an equal number of images for each of the 16 classes.

For the unoccluded task, where the input is two images, one in which only the foreground object is present and another in which only the background object is present, we concatenated these images in the channel dimension.

**Hyperparameter search and neural network training.** For each model, we performed 90 random searches to tune hyperparameters. Each random search began by randomly sampling a learning rate, an L2 weight decay coefficient, a batch size, and a dropout probability from the following distributions: the log-uniform distribution in [1e-4, 5e-3] (learning rate), 0 with 15% probability or a value sampled from the log-uniform distribution in [1e-12, 1e-3] with 85% probability (L2 weight decay coefficient), the discrete uniform distribution over 32, 64, 128 (batch size), and the uniform distribution in [0.2, 0.7] (dropout probability).

Once the hyperparameters were selected, the model was trained using the Adam optimizer, a standard variant of the mini-batch gradient descent algorithm, with its default hyperparameters[67] and its classification accuracy (the total classification accuracy, when there were two class labels to predict) on the validation set was recorded after each epoch of training. The learning rate was decayed by a factor of 2 if the number of "bad" epochs in which the validation set accuracy did not improve by more than 0.5% was greater than 4. The number of bad epochs was reset to 0 when a more than 0.5% improvement occurred, and once the learning rate was decayed, a cooldown period of 4 epochs had to be waited before it could be decayed again. If either the validation set accuracy did not reach 60% in the first 10 epochs or the number of bad epochs reached 10, the search stopped immediately or otherwise, the search continued for 100 epochs. Once the search was completed, the best validation set accuracy thus far was recorded, and the corresponding model was saved for the current set of hyperparameters. After 90 such random searches, we identified the hyperparameters with the highest validation set accuracy.

After identifying the optimal hyperparameters for each model, we trained and saved five instances of each model with its optimal hyperparameters exactly in the same way as above. The performances of the five saved instances on the test set were used to calculate the mean and standard deviation of each model's performance.

In all the feedforward models, all the weights were initialized by the Kaiming initialization[68] with the factor of the square root of 2 removed from the standard deviation of the Kaiming normal distribution when the weights were not followed by a ReLU. All the biases were initialized to 0.

In all the (top-down) recurrent models, all the weights were initialized by the normal distribution whose mean is 0 and standard deviation is the inverse of the square root of the number of input units. All the biases were initialized to 0, except for the forget biases of the (Conv)LSTMs, which were initialized to 1, following the common practice[69]. The initial hidden states of the (Conv)LSTMs were initialized to 0 and learned in the same way as the other parameters.

**Details on feedback connections in the top-down recurrent models.** In all feedback connections, we first performed upsampling, to match the difference in spatial scale between these layers (due to max-pooling), and applied a $1 \times 1$ convolution whose number of output feature maps equals the number of feedforward feature maps incoming to the target layer of the feedback connection. The output feature maps of the $1 \times 1$ convolution were then concatenated in the channel dimension with the incoming feedforward feature maps before being fed into the target layer.

**3D occlusion dataset generation.** We created the 3D occlusion dataset from ThreeDWorld (TDW)[39], a high-fidelity multi-modal platform for physical simulation. We selected 10 classes of objects from the full 3D model libraries in TDW. Each class contains at least 11 distinct objects. Each image consists of an occluding object and an occluded object, which gives the 100 categories from the 10 object classes. The dataset includes 163,963 RGB images (size $256 \times 256$) in total, and each category contains at least 1300 distinct images.

To generate an image with a given object pair, we first randomly sampled a position to place the camera. The camera position will be set as the origin for the purpose of describing the generation process. Two random points in the sphere centered at the origin are randomly sampled. The occluding object was placed at the point closer to the origin, and the occluded object was placed at the farther point. To increase the complexity, each object was allowed to rotate around the axis passing through itself and the origin, while only small jitters were added to the other two rotating directions. The camera was rotated to point at the occluded object, but we also applied small jitters to the pointing direction of the camera. To increase the diversity with a limited number of object samples, we randomly sampled materials for each substructure. The background environment is also randomly chosen and rotated for each image. We define the projection size of an object as the number of pixels it takes, normalized by the total number of pixels of the image. Each object underwent random scaling, but its projection size is restricted to the range from 10 to 20%. We define the occlusion level as the ratio between the number of pixels that both objects overlap and the number of pixels that the occluded object takes. We filter the images by the occlusion level. We only include images with occlusion levels from 0.3 to 0.5. To increase the possibility of generating an image with the desired occlusion level, we restrict that the two objects should be at most 15° apart. We iterated each object pair with objects from two chosen object classes to generate images of one category. The iteration will be repeated until we have enough images for each category. The occlusion level of the dataset has a distribution close to the uniform distribution. Each 0.01 range of occlusion level contains $8198 \pm 222$ images.

**Details of the experiments.** Model architecture: we use CORnet-R[35] as our base model to test the hypothesis. CORNet[35] is among the top models on Brain-Score. A CORnet-R block consists of 2 convolutional layers with a recurrent connection from the previous output to the input of the second convolutional layer, so the recurrence is only within each block. CORnet-R consists of 4 CORnet-R blocks (named by "V1", "V2", "V4" and "IT") and a top decoder (Avgpool-Flatten-FC). In our setup, the final output dimension is 10, corresponding to the 10 object classes in the 3D occlusion dataset. CORnet-R does not have a biologically plausible unrolling in time[35], which makes it easier for us to

connect to the previous models that do not have a biologically plausible unrolling in time as well.

Data preparation and training: We prepared 10 sets of train-test splits. For each category, we randomly chose 1000 images for training and 300 images for testing, which sums up to 100,000 images in the training set and 30,000 images in the test set. Each image is randomly cropped to a size of $224 \times 224$. For each model setup, we ran the training and reported the test performance 10 times (once for each train-test split). In the normal order setting, the model is trained to predict the occluding object in the second last step and the occluded object in the last step, while the prediction order is reversed for the reversed order setting. At the beginning of the training, each model is initialized from the pretrained weights. We use the model weights that are pretrained on the Imagenet[58] and randomly initialize the last FC layer. We used a fine-tuning learning rate of 0.01. All the other training parameters are the same as the ones used for pretraining[35]. The models were trained for 20 epochs and then evaluated on the test set.

No statistical method was used to predetermine sample size, and no data were excluded from the analyses.

## A biologically inspired model can perform an explicit form of explaining-away

**Image dataset generation.** The datasets used for Recon-Net were very analogous to the one described above. The only difference is in the implementation of spatial transformations applied to FashionMNIST images. Instead of first zero-padding $28 \times 28$ images and then transforming them, we directly transformed them to $50 \times 50$ images using Spatial Transformer Network[66]. Since we did not zero-pad source images, we needed to apply scale transformations to them as well as rotations and translations. For each source image, the scale factor was always fixed to 2.2 (this is the factor by which the object in the source image shrinks after the transformation), the rotation angle was sampled from the uniform distribution between −180 and 180 degrees, and the x- and y-translations were sampled from the uniform distribution between −0.55 and 0.55. Unlike in the above dataset, the translations for the foreground and background objects were not coupled. A $2 \times 2$ matrix was formed by multiplying the scale factor with the rotation matrix corresponding to the sampled angle, and concatenated with a $2 \times 1$ matrix representing the sampled x- and y-translations to yield a $2 \times 3$ affine transformation matrix for the Spatial Transformer Network. The spatially transformed individual-object images were segmented and composed in exactly the same way as above.

The core dataset used for the comparison of Recon-Net and end-to-end trained classifiers consists of images whose occlusion levels range between 10 and 60%, with an equal number of images in each 10% bin in that range (50,000 training and 10,000 test images). The occluded dataset used in the occluded-to-unoccluded generalization consists of images whose occlusion levels range between 50 and 75%, with an equal number of images in each 5% bin in that range (50,000 training and 10,000 test images). The unoccluded test set consists of unoccluded background-object images (i.e., the images of background objects before being occluded by the foreground objects) from test images in the core dataset.

**Hyperparameter search and neural network training.** For all models other than the generative module of Recon-Net, the learning rate of 1e-4 and batch size of 64 were used, the L2 weight decay coefficient was always set to zero, and the dropout probability was set to 0.5 if a model contained a dropout layer. Each model was trained for 100 epochs without a learning rate decay. The model parameters were initialized in exactly the same way as in the comparison study.

The generative module of Recon-Net was trained following a standard training practice proposed in ref. 70: the learning rates for both the generator and discriminator were 2e-4, the momentum

hyperparameter of the Adam optimizer was set to 0.5, the batch size was 64, and the number of training epochs was 25. Its weights were initialized by the normal distribution with mean 0 and standard deviation 0.02, and biases were initialized to zero, also as suggested in ref. 70.

We trained five instances of each model with randomly initialized parameters, and their performances on the test set were used to calculate the mean and standard deviation of each model's performance.

**Details of architectures.** We considered two versions of Recon-Net: one in which the background object is in its original pose (as in FashionMNIST dataset) and only the foreground object is randomly rotated and translated ("original pose" condition), and one in which both of them are randomly transformed but the transformation is inferred and reversed by separately trained networks ("predicted pose" condition).

In the original pose condition, the $50 \times 50$ images generated as described above were transformed back to the original pose of the background object and downsampled to $28 \times 28$ using Spatial Transformer Network[66]. The resulting images of an occluded background object were used as one of the inputs to Recon-Net, or "x" mentioned above. Also, for each of the $50 \times 50$ images, we applied the same transformation and downsampling to the corresponding foreground-object-only image, to obtain the other input to Recon-Net, "$x_f$".

In the predicted pose condition, a separately trained network, "Disentangler", first disentangled the images of a foreground and background object into foreground- and background-object-only images. Then, another network, "PoseNet", inferred the transformation applied to the background object, taking as input the background-object-only images predicted by Disentangler. The inferred transformation was applied to the original $50 \times 50$ images using the Spatial Transformer Network, producing x, and similarly to the inferred foreground-object-only images, producing $x_f$. As in the original pose condition, x and $x_f$ were also simultaneously downsampled to $28 \times 28$.

Details of the architectures of Recon-Net, Disentangler, PoseNet, and end-to-end trained decoders are described below.

Let FC[N] be a fully connected layer with N output units, Conv[N, k, s] a convolutional layer with N output feature maps, kernel size k, and stride s, TransConv[N, k, s] a transposed convolutional layer with N output feature maps, kernel size k, and stride s[71], BN a batch normalization layer[72], Dropout[p] a dropout layer with dropout probability p[73], ReLU a ReLU nonlinearity[74–76], LReLU[s] a Leaky ReLU nonlinearity with negative slope s[77], Sigmoid a sigmoid nonlinearity, and Tanh a tanh nonlinearity.

**Recon-Net.** The generative module of Recon-Net is the generator of a GAN[48] trained on original FashionMNIST images of the 4 selected classes (Fig. S2A). It takes as input a noise vector sampled from the uniform distribution over the 62-dimensional unit interval. Its architecture is FC[1024]-BN-ReLU-FC[128 7 7]-BN-ReLU-TransConv[64, 4, 2]-BN-ReLU-TransConv[1, 4, 2]-Sigmoid. The discriminator of the GAN has the following architecture: Conv[64, 4, 2]-LReLU[0.2]-Conv[128, 4, 2]-BN-LReLU[0.2]-FC[1024]-BN-LReLU[0.2]-FC[1]-Sigmoid. This GAN architecture is inspired by ref. 70.

The inference module of Recon-Net is a recursive feedforward network that takes images of an occluded background object and a foreground object, x and $x_f$, and iteratively predicts a 62-dimensional latent vector for the background object in the input image over 5 time steps (Fig. S2B). At each time step $t$, it is given its prediction of the latent vector from the previous time step, $z_{t-1}$, and the corresponding image generated by the generative module, $G(z_{t-1})$. $G(z_{t-1})$ is first channel-wise concatenated with x and $x_f$. The concatenated tensors are then fed into a block of convolutional and fully connected layers, which has the following architecture: Conv[256, 4, 2]-BN-LReLU[0.2]-Conv[256, 4, 2]-BN-LReLU[0.2]-Conv[128, 3, 1]-BN-LReLU[0.2]-FC[512]-BN-LReLU[0.2]. Its output is concatenated with $z_{t-1}$ and then fed into another black of fully connected layers with the following architecture:

FC[512]-BN-ReLU-FC[62]-Tanh. The output of the second block is a correction to $z_{t-1}$, which is added to $z_{t-1}$ and then clamped at 0 and 1 to yield $z_t$. The input latent vector at the beginning of the iteration, $z_0$, is the sigmoid of a learned parameter initialized to 0.

**Disentangler.** Disentangler consists of an encoder, decoder, and classifier. The encoder encodes the input image x into a 512-dimensional latent vector and has the following architecture: Conv[128, 4, 2]-LReLU[0.2]-Conv[128, 4, 2]-BN-LReLU[0.2]-Conv[128, 4, 2]-BN-LReLU[0.2]-FC[512]. The latent vector from the encoder is split in half, with the first half representing the foreground object in x and the second half the background object. The decoder reconstructs a single-object image of the foreground (background) object from the latent vector of the foreground (background) object and has the following architecture: FC[64 6 6]-BN-ReLU-TransConv[64, 4, 2]-BN-ReLU-TransConv[64, 4, 2]-BN-ReLU-TransConv[1, 4, 2]-Sigmoid. The classifier predicts the class label of the foreground (background) object from the latent vector of the foreground (background) object and has the following architecture: FC[128]-BN-ReLU-Dropout[0.5]-FC[64]-BN-ReLU-Dropout[0.5]-FC[4].

**PoseNet.** PoseNet consists of a component estimating the scale transformation and translation, and another component estimating the rotation applied to the FashionMNIST object in the input image. In our experiment, the input image is a background object image predicted by Disentangler. The first component estimates the three parameters specifying the scale transformation and translation from the input image and has the following architecture: Conv[128, 4, 2]-BN-ReLU-Conv[128, 4, 2]-BN-ReLU-Conv[128, 4, 2]-BN-ReLU-FC[256]-BN-ReLU-FC[128]-BN-ReLU-FC[3]. The input image is transformed by the inverse of the estimated scale transformation and translation using the Spatial Transformer Network (49), and the second component estimates the angle of the rotation from the transformed image. It has the following architecture: Conv[128, 4, 2]-BN-ReLU-Conv[128, 4, 2]-BN-ReLU-Conv[128, 4, 2]-BN-ReLU-FC[256]-BN-ReLU-FC[128]-BN-ReLU-FC[1].

**End-to-end trained decoders.** The feedforward decoders have the same architecture as FF in the previous subsection, with the following differences: the second and third convolutional layers have 128 output feature maps, and the first fully connected layer has 512 output units. Similarly, the recurrent decoders have the same architecture as Rec 5 in the previous subsection, with the following differences: the second and third ConvLSTM layers have 128 hidden feature maps. These decoders were either trained on unoccluded images, occluded images, or images predicted by Recon-Net. The same architectures were also used for the occluded-to-unoccluded generalization test.

No statistical method was used to predetermine sample size, and no data were excluded from the analyses.

## Effect of occlusion on internal representations of neural networks

**Image dataset generation.** For comparison of occlusion-trained and non-occlusion-trained classifiers, we used the same dataset as in the subsection "A biologically inspired model can perform an explicit form of explaining-away". In particular, images in the original pose condition were identical to those images used in the original pose condition in that subsection. Images in the random pose condition were the original $50 \times 50$ two-object images in that subsection.

The dataset used for comparison of translation-trained and non-translation-trained classifiers was generated in the same manner as the above dataset, except that we only had one object per image and did not randomly rotate it.

Similarly, the dataset used for comparison of translation-trained and non-translation-trained classifiers was generated in the same

manner as the above occlusion dataset, except that we only had one object per image and did not randomly translate it.

**Details of the calculation of representational similarity.** As explained in the main text, we quantified the representational similarity between occlusion-trained and non-occlusion-trained classifiers by performing linear regression from activations of one layer in one classifier (the source model) to those of the same layer in the other classifier (the target model). The representational similarity is defined as the correlation between the predicted and target activations of the linear regression across held-out images, averaged over pairs of randomly initialized instances of the source and target models. Specifically, the correlation between the predicted and target activations of the linear regression in each layer was calculated separately for each target unit. The median of this correlation over units (which was further averaged over pairs of instances, and for convolutional layers also averaged over different random samplings of target units) was taken to be the measure of the representational similarity in that layer.

No statistical method was used to predetermine sample size, and no data were excluded from the analyses.

## Human psychophysics demonstrates explaining-away in the human visual system

**Details of task construction and design.** The psychophysics task was custom-developed using PsychoPy[78] and hosted online through Pavlovia (Pavlovia.org). The task began with a guided run-through of a single trial. Participants could view what each phase of the experiment would look like, along with explanatory text. Next, participants performed at least one practice trial after which they received feedback on their performance. Participants could perform as many practice trials as they wanted and could continue to the experiment at any time.

Each trial begins with a white fixation + in the center of the screen for 500 ms. Next, the single-object image is presented for 500 ms. The object is either the occluding object or a randomly set different object. A second fixation of either 50 ms or 25 ms (set randomly with equal probability) separates the single object image from the multi-object image (400 ms). In the multi-object image, one item is occluding the other, and participants are tasked with identifying the background object. The degree of occlusion ranged from 70 to 90% coverage, with lower occlusion levels overrepresented.

Next, two items are shown on either side of the screen. One item is the occluded item in the multi-object image, and the other is randomly set. Both items appear in their canonical orientation—a contrast to how they may appear in the single and multi-object images, which are randomly rotated. Participants have three seconds to indicate which of the two items was the background object using their arrow keys. The trial then advances. All images are grayscale, so participants cannot rely on color.

In a random 10% of trials, participants were presented with an attention check. A white circle appears on either side of the screen, and participants have 3 s to use their arrow keys to indicate which. An unanswered attention check is considered a miss.

Each participant could perform up to 400 trials, split into groups of 100. At the end of each group, participants were given a 30 s break, with a 10 s countdown indicating the start of the next group.

**Subject recruitment and compensation.** All participants were recruited online via mTurk. Using mTurk's screening tools, participants were confirmed as English-speaking adults living in the U.S. Interested participants were directed to a Qualtrics survey that secured their informed consent prior to any study procedures, briefed them on the task, and collected self-reported demographic information. All protocols and procedures were approved by the Stanford University Institutional Review Board.

**Demographics and SAGER items.** Of the 182 individuals who initiated the task, 41 participants passed all pre-specified quality-control (QC) checks and were included in the analyses (see QC below). Demographic information could be confidently linked (via self-entered IDs) for 35/41 included participants. For these 35: age mean = 37.7 years (SD = 9.1, range 21–68); sex/gender (self-reported): 24 male (68.6%), 11 female (31.4%). Self-reported race/ethnicity: White/Caucasian 25/35 (71.4%), Hispanic 3/35 (8.6%), Black/African American 3/35 (8.6%), Indian 2/35 (5.7%), East Asian 1/35 (2.9%), Other 1/35 (2.9%); Southeast Asian, Native Hawaiian or Pacific Islander, Native American, and Middle Eastern/North African were each 0/35 (0%). Sex/gender was not considered in the study design, and no sex/gender-based analyses were performed because such analyses were orthogonal to our research questions and underpowered. Sex/gender was determined by self-report; no assignment was performed. Where consent and data linkage permit, disaggregated sex/gender counts are provided in the Reporting Summary and Source Data.

Participants received a base compensation of $2 for their participation in the study. Additionally, for each trial, subjects got right in excess of chance (50%), and they received an additional $0.04 for a maximum total compensation of $10.

**Psychophysics analysis.** Data from the psychophysics task was anonymized and analyzed using custom Python scripts. Since analyses were performed on different groups of trials performed by the same participant, paired Student's t-tests were used to assess the significance of any differences between conditions (e.g., the effect of explaining-away on performance or response time).

No statistical method was used to predetermine sample size. The experiments were not randomized (participants were not allocated to groups); however, within-participant trial order, fixation duration (25 vs. 50 ms), and stimulus identities were randomized by the task software. The investigators were not blinded to allocation during experiments and outcome assessment. Data were excluded only according to the a priori QC criteria detailed below; no other data were excluded.

Prior to analysis, rigorous quality control was performed. Participants with a mean response time below 100 ms were excluded to detect straightforward automated response systems, as were those who failed more than 20% of attention checks or whose accuracy was below 0.55. Trials with response times below 100 ms were excluded as well. Of the 182 participants, 41 passed all controls and were included in the analysis.

## Statistical tests
All statistical tests we used, except for the one used in Fig. 5A were two-sided Student's *t* tests. Data distributions were assumed to be approximately normal with comparable variances across groups. Data were visually inspected for normality and homogeneity of variance and found to be consistent with test assumptions. In Fig. 5A, we used a two-sample Kolmogorov–Smirnov (K–S) test. The assumptions of the two-sample K–S test were met (independent samples and one-dimensional continuous variables).

## Software
For all our data generation, collection and analysis, Python 3 was used, and for computational studies specifically, Pytorch v.0.3 was used.

## Reporting summary
Further information on research design is available in the Nature Portfolio Reporting Summary linked to this article.

# Data availability
The datasets used in this study (derived fashion MNIST, derived 3Dworld) are available in a GitHub repository (https://doi.org/10.5281/zenodo.17655370). Note that demographic information has been

removed from the psychophysics dataset. This information is available by email request from the corresponding author.

## Code availability

Code used in this study is available in a GitHub repository (https://doi.org/10.5281/zenodo.17655370).

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

## Acknowledgements

We thank Brett Larsen and Tyler Benster for discussions and Jon-Michael Knapp for comments on the manuscript. This work was supported by the National Institute of Health (NS113110, EB02871) and the Simons Collaboration on the Global Brain (SPI542969).

## Author contributions

B.K. and S.D. conceived and designed the study; B.K. and F.C. performed the computational experiments and analyses; B.M. implemented, ran and analyzed the psychophysics experiments. All authors discussed all results and contributed to the writing of the paper.

## Competing interests

The authors declare no competing interests.
