## [Transparent Peer Review File · Nature Communications]

Recurrent connections facilitate occluded object recognition by explaining-away

Corresponding Author: Professor Shaul Druckmann

A version of this paper was originally rejected for publication by Nature Communications, however that decision was reconsidered after appeal by the authors.

Version 0:

Reviewer comments:

Reviewer #1

(Remarks to the Author)

This manuscript tackles the critical problem of recognizing occluded objects. Many past publications have argued that recognition of occluded objects relies on recurrent connections, but we do not know why/how recurrent computations contribute to this process. This manuscript aims to clarify the role of recurrent computations in recognizing occluded objects. The questions addressed in this manuscript are important and the work is presented clearly. There are some insightful observations, but other claims are not entirely supported by the results and the biologically inspired model does not appear to consider alternative possibilities. I elaborate on each of these below. Overall, given the weaknesses/questions raised below, I believe that this manuscript will produce just a modest impact on the field.

Strengths:

The demonstration that the advantage of recurrent networks stems from computational complexity rather than number of weights or the nature of recurrent computation, is a novel and significant insight that's clearly laid out.

Questions/weaknesses:

While the demonstration that performance is improved for the full task when the foreground object is first identified followed by the background obj compared to background obj alone (Fig 3) its unclear why Fig 3C for full task – foreground alone skews negative for all models. Why does this make sense? This is not clarified.

It seems logical that the foreground and background objects would need to be segmented for accurate recognition of the background object and this is reflected in Fig 3B, perhaps. While “explaining away” the foreground object may seem like an intuitive explanation, nothing in Fig 3 suggests this is what is going on and I did not see any analysis of the weights to provide rigor for this interpretation.

It will be good to describe the rationale for the various network designs used here. Obviously, many different architectures could be used. Why did the authors test these chosen models?

Results based on the top-down recurrence models seem arbitrary. Some designs work well while others do not. Is this simply network idiosyncrasy or does it mean something?

The biological model in Fig 4 seems more consistent with the top-down rather than recurrent models? And the results from Fig 3 suggest that the top-down models may be idiosyncratic and not always perform well? How does the author justify this choice?

Is reconstitution at the PFC level necessary? What is the alternative? Would it not be more general to recognize the object based on the visible features? This model seems like one possibility and it would be important to consider alternatives.

The PFC reconstitution implies that PFC neurons will encode the background shape regardless of the occluder. Is there any physiological evidence for this? This model appears to be based on Fyall et al, but to the best of my understanding, in that paper PFC encodes a tangled representation of foreground and background objects.

Line 378: What does it mean to say that vIPFC has its own representation of visual objects independent of V4? vIPFC gets its visual inputs from V4 and IT cortex. So, unclear what an independent representation means. Greater clarity is needed.

Minor:

Line 153: typo

Line 176: typo

Reviewer #2

(Remarks to the Author)

This is an interesting read. In “Explaining away by recurrent connections arises naturally in models of occluded object recognition”, the authors perform a set of modelling experiments on DNNs in which they investigate the role of recurrent network connectivity in the processing of occluded stimuli. The paper is well written and accessible, and the experiments are thoughtful and well-executed. Nevertheless, I have detailed below a few major and minor comments that I hope will help the authors strengthen the overall paper.

Signed

Tim Kietzmann

Major:

(1) A larger claim of the paper is that explaining away occurs in recurrent networks and that due to the large set of recurrent neural network architectures tested, this provides clues for the generality of these findings. Yet, the authors predominantly test convolutional LSTM networks, whereas many other recurrent architectures exist and are commonly used. For example, recurrent convolutional networks (e.g. CorNet, BLT, etc) have previously been used to test for the role of recurrence. Would the results generalise to these architectures? Currently, it remains unclear to me (1) how well we can generalise from this to previous work given the core network units are substantially different (2) how fair the comparison to feedforward convolutional-networks is. Non-LSTM convolutional recurrent networks seem like a more conservative control here (which is why previous researchers have used them). It would be great if the authors could test such recurrent network structures alongside their LSTM models to allow for a match to previous work.

(2) I feel that statements of novelty and the implications of this work to highlight general principles are at times overstated. See below for a few examples that occur throughout the text:

- The authors claim that “Contrary to these previous studies, we arrive at a more nuanced answer. Our more extensive, controlled comparison analyses show that recurrent computation is not intrinsically superior to feedforward computation on occluded object recognition, as sufficiently deep feedforward networks were able to outperform recurrent networks.” Similar statements are made throughout the paper (e.g. again in line 230ff). Contrary to this statement, however, previous work has indeed demonstrated this very case, with feedforward models being able to outperform recurrent models, albeit at higher parametric costs (see e.g. Spoerer et al. 2020 Plos CB). Please adjust the wording accordingly.

- In a similar spirit, the authors introduce “computational depth” as a measure of network complexity. Related measures have been proposed before (e.g. Spoerer et al. 2020 use the number of floating-point operations). How does the new measure relate to existing ones, and, if the measures correlate strongly (as I expect), then could the authors indicate why the introduction of a new measure was required? The text follows up by claiming that recurrence allows networks to use time as a computational resource. This has been stated repeatedly across the literature on recurrent networks.

- See point 4 below.

- The claim of LSTMs trained on FashionMNIST to be a “realistic evaluation” of the role of recurrence is overblown. See (6) below.

(3) The transition from the extensive recurrent model tests to the authors’ new “PFC-net” seems a bit arbitrary to me. Can the authors re-state why PFC-net needed to be introduced in addition to the previous models? Were analyses performed that could not be done on the previous models? As far as I see, the decoding analyses (discussed on minor #4) could likely also be done on the recurrent networks if they indeed perform explaining away, no?

(4) The authors state that the networks exhibited “unmodified”, or “nearly unchanged” responses to unoccluded stimuli. This wording is too strong given the actual results. First of all, the responses are by no means unmodified, as the similarity is around 0.7 in a noise-normalised setting (i.e. similarity at 70% of the noise ceiling, but we do not know what the noise ceiling is, the raw correlations are therefore likely much lower). Second, the images have substantial areas with black pixels, thereby automatically leading to a larger agreement in a convolutional setting, irrespective of the learned features. Third, the

mapping is done by a linear mapping that involves substantial numbers of parameters. Currently, we do not know how much weight-lifting is done by this additional step. I suggest that the authors (1) dial down the claims of “unmodified” representations, (2) provide information on the raw, non-normalized similarity, and (3, perhaps optional) perform RSA to compare models (see Mehrer et al. 2020 Nat Comms for an example) as this does not require parameter fitting.

(5) Missing control: The authors compare recurrent models with task-order (e.g. unoccluded before occluded) to feedforward networks performing both tasks at once. This is an interesting comparison, but I would like to suggest an additional control: Why not train the feedforward networks with two readouts at different depths to mimic a similar sequence? This would distinguish between pure sequence effects and the impact of recurrence in this task.

(6) We clearly need a section on the limitations of this work, including the use of FashionMNIST only, and LSTM units to draw quite general conclusions about general recurrence.

Minor:

(1) Typo in l 154 (extra “be”): “The foreground- and background-only task controls were be used to more precisely...”.

(2) Terminology. Many researchers use RSA to investigate representations in deep neural networks. The authors’ choice of calling their analyses “representational similarity” is therefore highly unfortunate. A careful rewording would alleviate this issue.

(3) I quite like the suggestion of an additional (primate) experiment in which unknown foreground objects are used to test the explaining-away hypothesis. Why was this experiment not performed on the models in this work also?

(4) The control experiment in lines 427ff is fine, but I feel that the insights gained from it are somewhat overstated (“particularly surprising”). If the network is based on a GAN, trained to produce unoccluded images from the dataset, and the model is trained to recover these reconstructions from the occluded image, then a classifier should easily be able to work (as also shown by the authors). That is, I think the experiment provides an interesting measure of how well the reconstruction works, but it may not be as surprising as the paper makes it sound. Relatedly, the perhaps more standard way to establish how well the reconstruction works would be to use a perceptual loss (I agree with the authors in that MSE is not necessarily helpful).

(5) If I am not mistaken, then the claim that “occlusion profoundly modified network connectivity” is not tested. Rather, the authors test for network activation profiles, not connection profiles.

Reviewer #3

(Remarks to the Author)

This review is for the manuscript entitled “Explaining away by recurrent connections arises naturally in models of occluded object recognition”. As the title implies, the main contribution of this work appears to be in demonstrating that “explaining away” (or filling-in to be more accurate) arises naturally in an RNN trained for object recognition under occlusion.

In general, I found the manuscript to be clear and well written. The authors ran a large number of control networks with varying depth and width, etc. My main criticism, unfortunately, is that the novelty and associated intellectual contributions are relatively limited from either a machine or biological vision perspective.

There is currently a lot of interest in the field of biological vision to extend CNNs and other related feedforward networks with recurrent/top-down connections. This is, however, a crowded field. No new mechanism is described here and there is already a substantial body of work that has shown that RNNs of the type studied here fit better neural data. There is also already quite a bit of work on the role of recurrent connections for object recognition under occlusions (see missing reference from O’Reilly et al below but also work by Tang et al cited etc). Compared to that type of work, the present study does not propose any novel architecture or computational mechanism. The main contribution is simply to show that “filling in” arises naturally when training RNNs under occlusion. This feels somewhat limited.

Below are more detailed comments. While most of these comments could be addressed I do not believe the manuscript would pass the threshold for Nature Communications.

From the perspective of ML or computer vision, the study is subpar. It uses a subset of what is already a somewhat toy dataset. The bar would be higher from an ML / computer vision dataset where multiple and larger scale datasets would have to be used to validate the approach. The dataset used really is a toy dataset with 4 classes, simple stimuli consisting of objects pasted on a uniform background. Hardly representative of the problem of object recognition.

There are really two different models which were used to run different experiments. I am unclear what is biologically plausible about the biologically-inspired model which includes a GAN and a whole bunch of ML tricks. This seems also reminiscent of work done in computer vision including Huang et al (2020); see missing references below.

From the perspective of biological vision, a human baseline is missing. It is unclear whether human observers could solve the task without attention (as is the case with all the ANNs tested). Training computational models exhaustively on all pairs of classes seems completely unrealistic as a model of biology. As a result, it is unclear whether these ANNs have any validity as computational models. The use of very low resolution images further reduces the credibility of the models as models of biology.

The use of the term explaining away is problematic. The term “explaining away” has a precise meaning in the context of Bayesian/probabilistic networks. I am not sure why filling in here is equated with explaining away.

Please consider splitting up citations (6-10) in two with modeling work (6-8) and data analysis (9-10).

Missing references:

Randall C. O'Reilly^{1,2*}†, Dean Wyatte^{1*}†, Seth Herd¹, Brian Mingus¹ and David J. Jilk². Recurrent processing during object recognition. *Front. Psychol.*, 01 April 2013 | <https://doi.org/10.3389/fpsyg.2013.00124>

Michaelis, C., Bethge, M. & Ecker, A. One-shot segmentation in clutter. In *International Conference on Machine Learning*,

Linsley, D., Kim, J., Veerabadran, V., Windolf, C. & Serre, T. Learning long-433 range spatial dependencies with horizontal gated recurrent units. In Bengio, S. et al. 434 (eds.) *Advances in Neural Information Processing Systems*, vol. 31 (Curran Associates, 435 Inc., 2018).

Yujia Huang, James Gornet, Sihui Dai, Zhiding Yu, Tan Nguyen, Doris Y. Tsao, Anima Anandkumar. *Neural Networks with Recurrent Generative Feedback. Neural Information Processing Systems (NeurIPS) 2020*

Dileep George*, Wolfgang Lehrach, Ken Kansky, Miguel Lázaro-Gredilla*, Christopher Laan, Bhaskara Marthi, Xinghua Lou, Zhaoshi Meng, Yi Liu, Huayan Wang, Alex Lavin, D. Scott Phoenix. A generative vision model that trains with high data efficiency and breaks text-based CAPTCHAs. *Science* 08 Dec 2017: Vol. 358, Issue 6368, eaag2612. DOI: 10.1126/science.aag2612

Version 2:

Reviewer comments:

Reviewer #1

(Remarks to the Author)

I have no additional comments or questions to the authors.

(Remarks on code availability)

Reviewer #4

(Remarks to the Author)

In this paper, the authors study the role of recurrent computations in facilitating occluded object recognition. There are three central claims—I will detail them and list my concerns.

One, recurrent neural networks (RNNs) and humans use “explaining-away” to recognize occluded objects. The difference in recognition accuracy when the foreground object is expected to be classified before the occluded object and when only the occluded object is expected to be classified is taken as a signature of “explaining-away.” In humans, the occluded-object classification performance difference between a foreground object shown alone before the mixed display and another foreground object shown alone before the mixed display is taken as a signature of “explaining-away.” In both settings, the idea is that taking the foreground object into account makes the classification of the occluded object easier.

Is this actually evidence for “explaining-away”? “Explaining-away” seems to be a vaguely defined term here. Is the idea that the representation of the foreground object is “deleted” from the stimulus representation? I don't think the above results speak to this. If I think about what information the foreground object adds to the occluded object in terms of classifiability, it would be that it bounds the extent of the occluded object, possibly highlighting some full-object possibilities over others, thereby making classification better. This sounds more like conditional inference or, as the authors say sometimes, contextual inference. How exactly this computation would look in the RNN representations is unclear, although such “interpretability” steps can indeed be taken to understand RNN operations (for example, see Thorat et al., 2021). Another possibility, which is closer to “explaining away,” is the idea that the network/humans end up focusing too much on the foreground object (which is larger), thereby not allowing the readout to respond to the occluded object. Here, somehow, asking to temporally separate the recognition of the foreground and background might allow attention to the background object and not focus on the foreground object. However, it is unclear why asking for foreground classification first increases the classification accuracy of the background object compared to asking for background classification first. A mechanistic

understanding is essential and lacking here.

Additionally, the claim that RNNs “use” “explaining-away” to recognize occluded objects is a bit misleading given these results. To make that claim, you’d have to show that the RNN trained on background-classification-only internally performs “explaining-away.” Instead, here a very specific objective manipulation is made wherein the foreground object classification is required before the background. I’d say this possibly tells us that RNNs “can use” “explaining-away” to recognize occluded objects.

Two, RNNs use “explaining-away” but deep feedforward NNs (FNNs), which perform as well as the RNNs, do not. The FNNs are trained to recognize the foreground and background objects at the same computational depth, and joint classification hampers background object classification, if anything.

As mentioned above, RNNs “can use” “explaining-away”—it is unclear if they do this implicitly. Now the FNNs were never asked to “segregate” the classification of foreground and background objects in computational depth. Perhaps if the ANNs are trained, Inception-style, with foreground readout coming before background readout, possibly in a computational-depth-tied-with-RNNs manner, we could examine them for corresponding “explaining-away” effects in performance. Conversely, if the RNNs are trained to classify both foreground and background objects at the same time, would we also see the background-task performance deficits we see for FNNs? Apparently not—the Rec Control model shows a small advantage given joint classification versus background-only classification. How do we interpret this? This RNN-FNN comparison requires more nuance.

Three, PFC-net is a biologically inspired model for “explaining-away.” The authors assume that reconstitution of the occluded object’s representation happens in vIPFC. They model it by explicitly asking the encoder to take separate inputs of the foreground-only object and the mixed images and requiring an unoccluded representation as the output (as the latent space of the decoder/generator is tuned for intact objects). They show that this network uses the recurrence to refine the representation of the occluded object, and it seems (Fig. S5E) that the additional foreground-only input was essential. Isn’t the notion of asking for a “reconstruction” of the occluded image similar to Tang et al. (2018)’s approach using Hopfield networks? Of course, in your case, the additional foreground-only input is essential; however, that could also be given as an input to a Hopfield network. The reason I am asking this is that I’m unsure what the right computational motif is for vIPFC. Is it truly a generative model as it is being modeled here? If not, should we call this PFC-net?

In summary, I think this paper convincingly demonstrates an interesting effect—processing the foreground object temporally ahead (or perhaps with a lower computational depth) of the background object enhances the classifiability of the occluded, background object. How this interaction between the foreground and occluded objects works is left unclear, although the authors link it to the term “explaining-away.” Future work needs to understand how this interaction works inside RNNs and inside human brains.

Minor comments:

1. One way to check if the foreground object is used by a Rec model trained with a background-only condition is to train the Rec model with background-only cutouts. If the performance for the latter case is lower than the former, then the background-only task is already an interesting testbed for contextual inference effects.
2. In the Introduction, paragraph 3, you mention that the “nature and role of recurrent computations remain unknown.” On the contrary, there’s plenty of work showing how RNNs perform figure-ground segmentation, contour tracing, other Gestalt grouping phenomena, and also object recognition (e.g., Linsley et al., 2020; Thorat et al., 2021; Goetschalckx et al., 2022; Thorat et al., 2023). I wouldn’t say we know everything there is to know about recurrent computations, but there are a couple of neat ideas out there on what kind of functions RNNs can compute “naturally.”

Regards, Sushrut Thorat

References:

- Goetschalckx, L., Zolfaghar, M., Ashok, A. K., Govindarajan, L. N., Linsley, D., & Serre, T. (2022). Toward modeling visual routines of object segmentation with biologically inspired recurrent vision models. *Journal of Vision*, 22(14), 3773-3773.
- Linsley, D., Kim, J., Ashok, A., & Serre, T. (2020). Recurrent neural circuits for contour detection. In *International Conference on Learning Representations*.
- Thorat, S., Aldegheri, G., & Kietzmann, T. C. (2021). Category-orthogonal object features guide information processing in recurrent neural networks trained for object categorization. In *SVRHM 2021 Workshop@ NeurIPS*.
- Thorat, S., Doerig, A., & Kietzmann, T. C. (2023). Characterising representation dynamics in recurrent neural networks for object recognition. *Conference on Cognitive Computational Neuroscience*.

(Remarks on code availability)

Reviewer #5

(Remarks to the Author)

The manuscript by B. Kang et al. titled “Recurrent connections facilitate occluded object recognition by explaining-away” presents very interesting work, but I am not really certain that the quality of the work is suitable for a high impact journal as *Nature Communications*. I can see many important problems which question the suitability of this paper.

I have structured the major problems into five areas:

Training dataset

I very much doubt the ecological validity of the training datasets. I don't think that flying pianos are really very common in our natural environment. I think the authors need to work really hard to convince me and potential readers that these are suitable test sets for the scientific problem in question.

Feedforward networks

I don't think that the structure of the feedforward networks with two outputs is a suitable structure. It seems very counter-intuitive to me that we would have different representations in the brain one for background objects and one for foreground objects. Perhaps an architecture with a "task node" (e.g. "look for background object") would be more suitable.

Behavioural experiment

Like with the training dataset I doubt that the pictures in the behavioural experiment are ecologically valid. Perhaps this would also explain the surprising high error rate. I don't think the objects as such are that difficult to recognize to warrant such a high error rate. Perhaps the context and the spatial arrangement put participants off? In general the high error rate questions the validity of the outcome. I would suggest to re-run that study with better images.

I am also wondering whether this effect is related to well-known object priming effects. I suggest to include a discussion of priming effects in the manuscript e.g., how their findings compare to these well-established effects.

Also I should note that what the authors termed "epochs" are probably experimental blocks.

Explaining away

At some point in the manuscript the authors defined "explaining away" as a process as "... reconstituting the unoccluded representation". Given centrality of this concept, it is important to relate their definition to other definitions in the literature. Is it unique? Is there agreement in what it means? Is it actually implicitly based on an iterative mechanism? If this is the case, it would be impossible in principle for a feedforward architecture to realized "explaining away"? I guess the definition does not include iterative processing. This makes me wonder whether the forward architectures also reconstitute representations or perhaps other types of "explaining away". This seems an important question in the context of the paper. So the authors should explore this using their RSA method.

Finally, when the authors analysed the recurrent networks, did they actually re-train them again? If so, why? Or where these the same networks with the same weights as in the previous sections?

Attention

The authors completely ignore an important alternative mechanism on how the brain may deal with occlusion, visual attention. This mechanism is particularly pertinent given the recent success of Transformers (or see below for papers from my own work with a transformer-like attentional mechanism). It is also interesting to note that my work explicitly combines recurrent connections with visual attention. I would like the authors to include a discussion of the relevance of attention in the paper and this concept is related to their findings, possibly as part of a future studies discussion.

Abadi, A. K., Yahya, K., Amini, M., Friston, K., & Heinke, D. (2019). Excitatory versus inhibitory feedback in Bayesian formulations of scene construction. *Journal of The Royal Society Interface*, 16(154), <https://doi.org/10.1098/rsif.2018.0344>
Heinke, D., & Humphreys, G. W. (2003). Attention, spatial representation and visual neglect: Simulating emergent attention and spatial memory in the Selective Attention for Identification Model (SAIM). *Psychological Review*, 110(1):29-87.

(Remarks on code availability)

Version 3:

Reviewer comments:

Reviewer #4

(Remarks to the Author)

Thank you for a very thorough response to my questions/concerns. I am convinced that asking networks (feedforward or recurrent) to first identify the occluder helps with subsequently identifying the occluded object. How exactly the identification of the occluder helps with the subsequent identification of the occluded object, in the RNNs, is left unclear.

Reconstruction of the occluded object is put forth as a hypothesis, exemplified through ReconNet, however, it is unclear if the previously discussed RNNs are indeed in the business of reconstruction. This requires going beyond the manipulations presented in Fig. 5, which is non-trivial and could possibly be for future work. However, a simple modification to the Rec networks - feedback to input (similar to the trick employed by Thorat et al. 2021) - might help us "see" what the Rec network chooses to do with the initial identification of the occluded object - does it 1) remove those pixels, 2) fill-in those pixels (similar to ReconNet), or 3) does something else entirely? Of course, this modified Rec network would only provide another hint at the underlying mechanism. Whether this is within the scope of this work I leave to the authors and the editor - it definitely is a non-trivial addition. A related next step would be the analysis of the background_only Rec networks. Does classifying just the occluded object lead to similar mechanisms seen in the F+B classification networks? If yes, then we are looking at a general mechanism which is boosted by explicitly asking the networks to classify the foreground first. If not, then indeed, asking for prior foreground classification would be an important task prior for a performant visual processor. How the brain acquires and executes such a prior is the next interesting question.

I think this work opens an interesting direction in understanding the processing of occluded objects and needs to be read by

a wide audience interested in this fundamental aspect of visual processing. I'm happy to support its publication.

Thank you,
Sushrut Thorat

(Remarks on code availability)
Code not provided.

Reviewer #5

(Remarks to the Author)

I very much appreciate the way the authors have dealt with my comments. They really responded to most of my points in a convincing way.

However, I must admit that I still have some gremlins with the lack of the ecological validity of their stimuli. But I can see that that there are good arguments for using them. I guess it comes down on how much out-of-distribution these images can be before they become no longer a valid experimental test of human information processing. Anyway, I will leave this to future empirical studies and I would like to congratulate the authors to their excellent manuscript.

(Remarks on code availability)

We thank the reviewers for their constructive comments. In addition to addressing specific points, in response to the reviews we performed two major sets of experiments, one computational and one human psychophysics. In brief, the psychophysics experiment was based on the hypothesis that prior exposure to a foreground object would allow it to be better 'explained away' when it reappeared in the occluded image, thereby improving recognition of the background object. This is exactly what we found. The computational experiment consisted of generating a new set of high-resolution 3D complex objects on rich room-like backgrounds and then testing a network built to model the primate visual system (CORNet) in a way analogous to our previous experiments with more generic architectures. This more advanced model on a more realistic dataset recapitulated our main findings. We feel these substantially strengthen our main claims and thank the reviewers for suggesting them.

Below, we attach reviewer comments in standard font, responses in green font and added here new text included in the manuscript to address specific points in blue font.

Reviewer #1 (Remarks to the Author):

This manuscript tackles the critical problem of recognizing occluded objects. Many past publications have argued that recognition of occluded objects relies on recurrent connections, but we do not know why/how recurrent computations contribute to this process. This manuscript aims to clarify the role of recurrent computations in recognizing occluded objects. The questions addressed in this manuscript are important and the work is presented clearly. There are some insightful observations, but other claims are not entirely supported by the results and the biologically inspired model does not appear to consider alternative possibilities. I elaborate on each of these below. Overall, given the weaknesses/questions raised below, I believe that this manuscript will produce just a modest impact on the field.

Strengths:

The demonstration that the advantage of recurrent networks stems from computational complexity rather than number of weights or the nature of recurrent computation, is a novel and significant insight that's clearly laid out.

Questions/weaknesses:

While the demonstration that performance is improved for the full task when the foreground object is first identified followed by the background obj compared to background obj alone (Fig 3) its unclear why Fig 3C for full task – foreground alone skews negative for all models. Why does this make sense? This is not clarified.

We thank the reviewer for this question and should have indeed clarified this point. In our experiments, the same neural network architectures with the same number of parameters (except for readout layers) are used for comparison between the two tasks. Therefore, as there are more

objects to recognize in the full task than the foreground-only task, meaning that it is a harder task, the performance in the former is expected to be no better than in the latter, unless somehow learning the additional task (i.e. background object recognition) also benefits the original task (i.e. foreground object recognition). Intuitively, there is no reason to think that the background object recognition will help the foreground object recognition, as the foreground object is fully visible. Thus, the drop in performance in the full task compared to the foreground-only task makes sense.

It seems logical that the foreground and background objects would need to be segmented for accurate recognition of the background object and this is reflected in Fig 3B, perhaps. While “explaining away” the foreground object may seem like an intuitive explanation, nothing in Fig 3 suggests this is what is going on and I did not see any analysis of the weights to provide rigor for this interpretation.

First, we completely agree that intuitively the most straightforward way to implement “explaining away” is to explicitly segment the foreground and the background object. We would have loved to show, based on the weights, that this is precisely what is going on. However, in general, producing a human-interpretable explanation of the algorithm a neural network has learned is a notoriously difficult problem in machine learning, and for any complex operation there are many patterns of weights that could implement it. We therefore do not expect that this could be revealed by an analysis of weights in a straightforward manner. Instead, we believe the best one can do is to establish a pattern of performance across architectures and controls, then associate it with the most parsimonious explanation. This is what we have done, and, in our opinion, the best possible interpretation of our results is by explaining away. In this new study we now also detail additional experiments, including psychophysics in humans (Fig. 5) and computational experiments involving a deep neural network class built to be predictive of the neural and behavioral responses of non-human primates (CORNet; Fig 4), to further support our conclusion. Both sets of experiments provide results supporting explaining away. We completely agree that it would be interesting to investigate the specific algorithms used by these neural networks in a future study, but this is beyond the scope of our current study, and it is not clear to us whether such a study could be conclusively performed. In addition, we note that examining the weights of a trained network might not yield a conclusive human-interpretable algorithm for complex computations, and we took a related but conceptually distinct approach: we developed and tested a model named PFC-net (Fig 6), which is designed to have a more human-interpretable algorithm for explaining away, along the line of the intuition mentioned above.

It will be good to describe the rationale for the various network designs used here. Obviously, many different architectures could be used. Why did the authors test these chosen models?

The main rationale for our choice of various network architectures in Fig 1-3 was that our neural networks should only involve generic, widely used architectural elements, such as convolutional layers, to minimize the risk of our results being specific to a particular architecture. We then sampled broadly across these classes of architectures (which yielded the very large number of models we trained). The scale of our neural networks was then chosen so that they perform the tasks reasonably well while still being able to be trained and tested with a reasonable amount of compute and time.

Results based on the top-down recurrence models seem arbitrary. Some designs work well while others do not. Is this simply network idiosyncrasy or does it mean something?

We completely agree with the reviewer that there is no consistent trend in the top-down recurrent models' results. Indeed, we believe that is actually the main results of these experiments: the effect of top-down connections on occluded object recognition is not consistent and not necessarily positive, unlike for the recurrent connections. Top-down models are highly sensitive to details of their connections, such as the layers they connect, and the recurrent dynamics, such as the number of recurrent time steps. We thought that this is a finding worth mentioning, which exposes an unexpected difference between the role of top-down and recurrent connections in the context of occluded object recognition. We tried to mention it in a cautious way, but we agree with the reviewer that this result could just confuse the reader in understanding the main finding of this manuscript: the explaining-away phenomenon. We therefore moved these figures to the supplementary material.

The biological model in Fig 4 seems more consistent with the top-down rather than recurrent models? And the results from Fig 3 suggest that the top-down models may be idiosyncratic and not always perform well? How does the author justify this choice?

Indeed, the model we introduced in Fig 4 of the old manuscript (Fig 6 of the current manuscript), PFC-net, can be thought of as having top-down connections. The reason why we designed PFC-net this way is because its design including the “top-down connections” made, in our opinion, the explaining-away phenomenon more human interpretable compared to the generic recurrent models considered in Fig 1-3. We do not at all argue that this design is optimized for performance or has the best performance possible. Instead, it was meant to: (i) have more interpretable recurrent dynamics that implement explaining away. (ii) perform occluded object recognition on high-dimensional visual inputs unlike the parsimonious model considered in Fyall et al. (2017), which takes essentially one-dimensional inputs. (iii). consist of interpretable components that may have functional counterparts in the brain as suggested by Fyall et al. (2017). We also note that there is no contradiction with the previous results in Fig. 3, as Fig. 3 did not show that no top-down architectures are high-performing, just that they are more sensitive to the specific details of their connections.

We now clarify this point and added the following text: *“We note that recurrent computations in PFC-net can be viewed as being performed by top-down connections. We designed PFC-net in this way because, in our opinion, this design including the “top-down connections” made the explaining-away phenomenon more human-interpretable. This is not in contradiction with the previous results in Fig. S1E, which did not show that no top-down architectures are high-performing, just that their performances are more sensitive to details of top-down connections.”*

Is reconstitution at the PFC level necessary? What is the alternative? Would it not be more general to recognize the object based on the visible features? This model seems like one possibility and it would be important to consider alternatives.

We agree with the reviewer that there are multiple possibilities for how occluded object recognition is performed in the brain. In general, though, we expect that neural networks recognize objects

not just by an individual feature, which may also be present in objects from different categories, but rather on the constellation of features present or absent. Therefore, the absence of a feature can be evidence against an object belonging to a particular class, as can the existence of unexpected features. This absence can be ameliorated by explaining away and correctly recognizing and dealing with an occluder can at least partially address ambiguity regarding which visual features belong to the occluded object as opposed to the occluder. With regard to the question of where explaining away happens in the brain, we are definitely open to the possibility that it occurs in areas different from PFC. But, in our reading, the results of Fyall et al. (2017) suggest that PFC is a candidate area where it may occur.

The PFC reconstitution implies that PFC neurons will encode the background shape regardless of the occluder. Is there any physiological evidence for this? This model appears to be based on Fyall et al, but to the best of my understanding, in that paper PFC encodes a tangled representation of foreground and background objects.

In Fig 3 of Fyall et al. (2017), they show that about 70% of PFC neurons have stronger object selectivity at higher occlusion levels and 30% at lower occlusion levels. This suggests that, at high occlusion levels where the representation of the background object is expected to be degraded compared to low occlusion levels, PFC generates a reconstituted representation of the background object. It is possible that, at low occlusion levels where explaining away is not necessary, PFC is less engaged, which could explain why it seems to exhibit weaker object selectivity. Also, as the reviewer mentioned, at the level of individual neurons, we expect the representations of the foreground and background objects are tangled. This question could be more directly addressed perhaps by investigating the representation of the foreground and background object by training population decoders, but this was not performed in Fyall et al. (2017). Such analyses would be helpful in gaining more insights into object representations in PFC.

Also, even though, in motivating our model, we made a simplifying assumption that the reconstitution happens in PFC, we acknowledge in the discussion section that the reconstitution may happen outside PFC, or through interactions between PFC and other areas, and PFC activity may only (partially) reflect the result of the reconstitution.

We have now made this into a more explicit statement in the discussion section: “While our modeling assumes that computations needed to reconstitute V4 representations of occluded stimuli occur in vIPFC, it is likely that this assumption holds only approximately. We expect neural computations that compensate for occlusion might also occur within the visual cortex itself (at least for low occlusion levels), given its ability to maintain invariance in neural responses (e.g. those of Inferior Temporal (IT) cortex) against variation in object appearance. More generally, it is possible that the reconstitution may happen outside vIPFC, or through interactions between vIPFC and other areas, and vIPFC activity may only (partially) reflect the result of the reconstitution. Thus, it will be interesting to study the interplay of occlusion-related computations in the visual cortex and in other areas, including vIPFC. Relatedly, a recent study showed, in the context of core object recognition, that a top-down feedback from vIPFC to the visual cortex is critical for recurrent processing of “challenging” images, for which IT takes additional time to develop selectivity (48). This might hint at a broader role of recurrent computations in multi-regional circuits in solving challenging visual tasks (49).”

Line 378: What does it mean to say that vIPFC has its own representation of visual objects independent of V4? vIPFC gets its visual inputs from V4 and IT cortex. So, unclear what an independent representation means. Greater clarity is needed.

What we meant is simply that there exist object-selective neurons or object-selective population activity in vIPFC. We are agnostic as to how such object-selective activity arises in PFC. It could arise from bottom-up visual inputs from V4 and IT cortex, or it may have some intrinsic object representations that may guide visual perception through its top-down influence.

We corrected the use of “its own representation” here which can indeed be confusing. The text now reads: “The last feature we incorporate in our model is that vIPFC has object-selective neurons or object-selective population activity, implying that it has a neural representation of visual objects. It seems reasonable to suppose such a representation given the shape-selective neurons observed in vIPFC”.

Minor:

Line 153: typo

Line 176: typo

Reviewer #2 (Remarks to the Author):

This is an interesting read. In “Explaining away by recurrent connections arises naturally in models of occluded object recognition”, the authors perform a set of modelling experiments on DNNs in which they investigate the role of recurrent network connectivity in the processing of occluded stimuli. The paper is well written and accessible, and the experiments are thoughtful and well-executed. Nevertheless, I have detailed below a few major and minor comments that I hope will help the authors strengthen the overall paper.

Signed

Tim Kietzmann

Major:

(1) A larger claim of the paper is that explaining away occurs in recurrent networks and that due to the large set of recurrent neural network architectures tested, this provides clues for the generality of these findings. Yet, the authors predominantly test convolutional LSTM networks, whereas many other recurrent architectures exist and are commonly used. For example, recurrent convolutional networks (e.g. CorNet, BLT, etc) have previously been used to test for the role of recurrence. Would the results generalise to these architectures? Currently, it remains unclear to me (1) how well we can generalise from this to previous work given the core network units are substantially different (2) how fair the comparison to feedforward convolutional-networks is. Non-LSTM convolutional recurrent networks seem like a more conservative control here (which is why previous researchers have used them). It would be great if the authors could test such recurrent network structures alongside their LSTM models to allow for a match to previous work.

We thank the author for this valuable comment. We note however that our main results regarding the explaining-away computation in Fig 3 do not depend on the comparison between feedforward and recurrent networks, but on comparison across different tasks and different ordering of tasks within the same architecture class. Therefore, a vanilla recurrent version of convolutional neural networks (vanilla recurrent CNN) would not necessarily be a better control in these experiments. That being said, our rationale for choosing the convolutional LSTM over a vanilla recurrent CNN is that convolutional LSTMs have been widely tested and used by the machine learning community for large-scale computer vision problems, such as video predictions, generations, and action recognition from videos, making it a safer choice for performance-based comparisons, which to us was the most relevant criterion to our approach. To the best of our knowledge, vanilla recurrent CNNs have been less extensively tested for their performance on a variety of standard computer vision benchmarks, and we were concerned that their performance may be limited by their specific architectural choices. However, we agree with the reviewer that replicating our core results for recurrent architectures other than the convolutional LSTM is important. Therefore, as the reviewer suggested, we performed additional experiments using CORNet, which is likely a better model of the primate visual system and showed that our key results regarding the explaining-away computation still hold.

(2) I feel that statements of novelty and the implications of this work to highlight general principles are at times overstated. See below for a few examples that occur throughout the text:

We apologize for that and have changed the text to address this.

- The authors claim that “Contrary to these previous studies, we arrive at a more nuanced answer. Our more extensive, controlled comparison analyses show that recurrent computation is not intrinsically superior to feedforward computation on occluded object recognition, as sufficiently deep feedforward networks were able to outperform recurrent networks.” Similar statements are made throughout the paper (e.g. again in line 230ff). Contrary to this statement, however, previous work has indeed demonstrated this very case, with feedforward models being able to outperform recurrent models, albeit at higher parametric costs (see e.g. Spoerer et al. 2020 Plos CB). Please adjust the wording accordingly.

We thank the reviewer for pointing out Spoerer et al. (2020). Indeed, the paper shows that feedforward models can outperform recurrent models on core object recognition without occlusion. However, at the time of writing the manuscript, to the best of our knowledge, we were not aware of previous studies that explicitly showed that recurrent computation is not intrinsically superior to feedforward computation on occluded object recognition. Given the findings on core object recognition, it is not surprising that the same is true for occluded object recognition, but we wanted to emphasize this point because, in our opinion, most previous studies gave the impression that recurrent computation is somehow intrinsically superior specifically for occluded object recognition.

We have now adjusted the tone of our statement as the reviewer suggests. The text now reads: “Additionally, our controlled architecture analyses found that recurrent computation is not intrinsically superior to feedforward computation on occluded object recognition, as sufficiently deep feedforward networks were able to outperform recurrent networks. Notably, control experiments showed that the high performance of deep feedforward networks was likely not achieved simply by converging on recurrent solutions (which are a subclass of feedforward solutions), but rather were composed of unique weight profiles. Performance correlated well not with overall parameter number but with computational depth, whether depth was achieved by feedforward layers or timesteps in a recurrent network. Our results challenge the conclusions of earlier studies, which reported considerable advantages for recurrent processing in comparison to feedforward processing on occluded object recognition (15, 16, 26). This is consistent with previous studies showing that feedforward models can outperform recurrent models on object recognition without occlusion (27).”

- In a similar spirit, the authors introduce “computational depth” as a measure of network complexity. Related measures have been proposed before (e.g. Spoerer et al. 2020 use the number of floating-point operations). How does the new measure relate to existing ones, and, if the measures correlate strongly (as I expect), then could the authors indicate why the introduction of a new measure was required? The text follows up by claiming that recurrence allows networks to use time as a computational resource. This has been stated repeatedly across the literature on recurrent networks.

We did not intend to claim any novelty regarding our statement that recurrence allows networks to use time as a computational resource and apologize that it came out that way. We introduced “computational depth” simply as a conceptually and operationally straightforward measure that captures this intuition to explain our finding in Fig 2A, B. We now rewrote the section, mentioning the relationship of computation depth to similar measures and appropriately adjusted the tone of our statements regarding computational depth.

The text now reads: “We found only a weak correlation between performance and number of parameters (Fig. 2A). Instead, we observed that the “computational depth” of a neural network was a better predictor. Following the widely accepted intuition that recurrence enables networks to use time as a computation resource, we defined the computational depth of a network as the maximum number of times an input variable is affine transformed by the network’s weights and then transformed by a nonlinearity (see methods). For an ordinary feedforward network, its computational depth is equal to the number of layers. For an ordinary recurrent network, its computational depth is equal to the sum of the number of layers and the number of recurrent iterations. We note that this measure is conceptually very similar to previously considered measures such as the number of floating-point operations (27), and we only introduce this simplified measure as it can be more readily read out from the architecture of the network. Performance was highly correlated with computational depth, although it tended to saturate at a depth greater than eight (Fig. 2B). This nicely dovetails with our prior observations on recurrent versus feed-forward performance, as computational depth can be increased by adding recurrent iterations and/or having more layers.”

- See point 4 below.

Please see our response below.

- The claim of LSTMs trained on FashionMNIST to be a “realistic evaluation” of the role of recurrence is overblown. See (6) below.

Please see our response below.

(3) The transition from the extensive recurrent model tests to the authors’ new “PFC-net” seems a bit arbitrary to me. Can the authors re-state why PFC-net needed to be introduced in addition to the previous models? Were analyses performed that could not be done on the previous models? As far as I see, the decoding analyses (discussed on minor #4) could likely also be done on the recurrent networks if they indeed perform explaining away, no?

The main reasons we introduced PFC-net are: 1. We wanted to make a more concrete connection with the brain by building a model that incorporates existing neurobiological findings. 2. Even though we presented evidence showing that generic neural networks perform the explaining-away computation in the previous section of the manuscript, the way such generic neural networks operate is often not human-interpretable. We wanted to build a deep neural network model that performs the explaining-away computation in a more explicit and intuitive way. 3. The neural networks previously considered were trained to identify an object’s category, but more fine-grained visual features, such as its texture and detailed shape, should also be represented in the brain. We wondered to what extent such fine-grained visual features can be reconstituted by an explicit version of the explaining-away mechanism.

As for the decoding analysis, if we can identify the layer(s) in which the reconstitution happens, then we can consider the following analog of the decoding analysis for the neural networks previously considered. We train a neural network named A incapable of reconstitution (e.g. by training it on only unoccluded images) and another named B which is capable of reconstitution (e.g. by training it on both unoccluded and occluded images). In network A, the intermediate representations of occluded objects will be very different from those of unoccluded objects as they would form very different visual stimuli. Therefore, if we train a decoder on the representations of unoccluded objects, it will not perform well on those of occluded objects. On the other hand, in network B, the representations of occluded and unoccluded objects will be similar because of reconstitution. Thus, if we train a decoder on the representations of unoccluded objects, it will perform reasonably well on those of occluded objects. Although this analysis can be done, there are a few technical difficulties with it. First, it is unclear a priori in which layer(s) the reconstitution happens. We will have to first identify such layer(s), which can be computationally expensive and time-consuming. Second, it is possible that reconstitution happens across multiple layers in a gradual manner. In this case, the above analysis may not reveal a very clear difference between the network A and B, except at the last hidden layer. But the decoding analysis results at the last hidden layer would not give much further insights beyond what can be already inferred from the network’s performance. We may do this decoding analysis for all layers and show that in the network B, the performance gap of the decoder gradually decreases across layers, which would suggest that reconstitution happens gradually. This will be computationally expensive, too. Although such analyses may still be doable in principle, we strongly expect the effects to be dispersed across layers and thus these analyses would be unlikely to provide much additional insight or support for our conclusions.

(4) The authors state that the networks exhibited “unmodified”, or “nearly unchanged” responses to unoccluded stimuli. This wording is too strong given the actual results. First of all, the responses are by no means unmodified, as the similarity is around 0.7 in a noise-normalised setting (i.e. similarity at 70% of the noise ceiling, but we do not know what the noise ceiling is, the raw correlations are therefore likely much lower). Second, the images have substantial areas with black pixels, thereby automatically leading to a larger agreement in a convolutional setting, irrespective of the learned features. Third, the mapping is done by a linear mapping that involves substantial numbers of parameters. Currently, we do not know how much weight-lifting is done by this additional step. I suggest that the authors (1) dial down the claims of “unmodified” representations, (2) provide information on the raw, non-normalized similarity, and (3, perhaps optional) perform RSA to compare models (see Mehrer et al. 2020 Nat Comms for an example) as this does not require parameter fitting.

We apologize for the manuscript not making this clearer, the similarity measure shown in Fig 7 is actually an unnormalized correlation. The noise ceiling is given by the orange-colored graphs, which represent the similarity of occlusion-trained networks to other occlusion-trained networks. Their similarity is limited by the instance-to-instance variability in random initialization of the network and thus represents the noise ceiling. When normalized to this noise ceiling, the similarity between non-occlusion trained and occlusion-trained networks for unoccluded stimuli is indeed very close to one for lower layers and > 0.9 for higher layers (e.g. see Fig7B’s left panel). Thus, we reasoned that the term “nearly unchanged” was appropriate. However, we agree with the reviewer that this wording was too strong and have changed it to “only weakly changed”.

The section now reads: “Moreover, the fact that training on occluded images only weakly changed representations of unoccluded images suggests that this type of solution may facilitate downstream processing by restricting representation shifts required by learning.”

As for the second point, since our claim is based on the normalized similarity compared to the noise ceiling, as explained above, it is not sensitive to the areas with black pixels.

As for the third point, we agree, of course, with the reviewer’s point that, though being the standard procedure in many studies, the linear mapping may involve a substantial number of parameters. However we note that, since our claim is based on the normalized similarity relative to the noise ceiling, this concern is mitigated because the linear mapping plays a role also in similarity of occlusion-trained networks to other occlusion-trained networks (the normalizing factor) and, therefore, the effects of the linear mapping should be largely normalized out.

(5) Missing control: The authors compare recurrent models with task-order (e.g. unoccluded before occluded) to feedforward networks performing both tasks at once. This is an interesting comparison, but I would like to suggest an additional control: Why not train the feedforward networks with two readouts at different depths to mimic a similar sequence? This would distinguish between pure sequence effects and the impact of recurrence in this task.

We thank the reviewer for this suggestion and agree that to see whether explaining-away can happen in feedforward networks when the readouts are arranged appropriately could be an interesting experiment. As the reviewer points out, it could distinguish between the impact of task ordering and recurrence. But we are not sure that such an arrangement of readouts is biologically realistic, and we feel that, though these suggested experiments are indeed elegant, the paper is

already complex, especially with the newly added sections and we therefore leave the reviewer's suggestion for a future effort.

(6) We clearly need a section on the limitations of this work, including the use of FashionMNIST only, and LSTM units to draw quite general conclusions about general recurrence.

We believe that the additional experiments involving the CORNet and a high-resolution dataset, and the human psychophysics experiment significantly strengthen our conclusions. However, we agree with the reviewer that a section on the limitations of this work is useful and have added one. It now reads:

Limitations of present work

While our study provides a comprehensive evaluation of recurrent computations in occluded object recognition, it is important to acknowledge certain limitations. First, although we have demonstrated the utility of recurrent computations using both convolutional LSTM-based models on the FashionMNIST dataset and more biologically realistic models (CORNet) on high-resolution images, it remains unclear how these results generalize to other neural network architectures and more complex real-world scenarios. Additionally, the human psychophysics experiment provides initial evidence for the relevance of explaining-away computations in human object recognition, but further studies are necessary to confirm these findings across diverse tasks. Future work should explore the generalization of these findings to other network architectures and to more varied and challenging datasets.

Minor:

(1) Typo in l 154 (extra "be"): "The foreground- and background-only task controls were be used to more precisely..."

We thank the reviewer for pointing this out. We corrected it.

(2) Terminology. Many researchers use RSA to investigate representations in deep neural networks. The authors' choice of calling their analyses "representational similarity" is therefore highly unfortunate. A careful rewording would alleviate this issue.

We understand that our terminology could be confusing to some readers familiar with RSA. We have now revised the text to distinguish our method from RSA.

The text now reads: "To better understand the asymmetry of generalization across different occlusion levels, we compared the internal representations of networks trained on occluded and unoccluded images. We quantified the similarity in representation between networks using a linear mapping approach. To be clear, we emphasize that this method differs from Representational Similarity Analysis (RSA), another commonly used method to investigate representations in deep neural networks (39). Our method involves performing linear regression from activations of one layer in one architecture, the source model, to another layer of a second architecture, the target model (6, 19) (Fig. 7A, see Methods)."

(3) I quite like the suggestion of an additional (primate) experiment in which unknown foreground objects are used to test the explaining-away hypothesis. Why was this experiment not performed on the models in this work also?

While we did not perform this precise experiment on the models, we performed essentially an analogous psychophysics experiment on humans. In brief, participants engaged in a background-object recognition task with two trial types. Each trial consisted of three epochs: a single object image presentation epoch, followed by an occluded image (foreground and background object) presentation epoch, followed by a response epoch. In foreground object trials, the single object was identical to the foreground object in the occluded image; in random object trials, the single object was a random different object. Our hypothesis predicted that prior exposure to the foreground object would allow it to be better explained away when it reappeared in the occluded image, thereby improving recognition of the background object. Consistent with this hypothesis, we found that the performance of the participants was significantly higher and the reaction time significantly shorter in the foreground object trials than random object trials.

(4) The control experiment in lines 427ff is fine, but I feel that the insights gained from it are somewhat overstated (“particularly surprising”). If the network is based on a GAN, trained to produce unoccluded images from the dataset, and the model is trained to recover these reconstructions from the occluded image, then a classifier should easily be able to work (as also shown by the authors). That is, I think the experiment provides an interesting measure of how well the reconstruction works, but it may not be as surprising as the paper makes it sound. Relatedly, the perhaps more standard way to establish how well the reconstruction works would be to use a perceptual loss (I agree with the authors in that MSE is not necessarily helpful).

We agree with the reviewer’s comment and have adjusted the tone of the statement. The text now reads: “We found that decoders that have never encountered occlusion are nevertheless able to classify occluded background objects when they are given images that have been reconstituted by PFC-net, but not when given the original occluded images (Fig. 6B). This suggests that PFC-net indeed effectively reconstituted the images. In fact, the level of accuracy achieved was almost as high as that of classifying occluded images by decoders trained directly on occluded images. This suggests that the reconstitution by PFC-net was nearly as effective as an end-to-end training for occluded object recognition.”

(5) If I am not mistaken, then the claim that “occlusion profoundly modified network connectivity” is not tested. Rather, the authors test for network activation profiles, not connection profiles.

We removed the sentence from the abstract of the current manuscript. As the reviewer said, we did not directly examine the network connectivity. Our original intent was that the substantially different activations in occlusion-trained networks compared to the non-occlusion-trained ones is most likely the result of changes in the network connectivity, but since we have not shown it we removed the statement.

Reviewer #3 (Remarks to the Author):

This review is for the manuscript entitled “Explaining away by recurrent connections arises naturally in models of occluded object recognition”. As the title implies, the main contribution of this work appears to be in demonstrating that “explaining away” (or filling-in to be more accurate) arises naturally in an RNN trained for object recognition under occlusion.

In general, I found the manuscript to be clear and well written. The authors ran a large number of control networks with varying depth and width, etc. My main criticism, unfortunately, is that the novelty and associated intellectual contributions are relatively limited from either a machine or biological vision perspective.

There is currently a lot of interest in the field of biological vision to extend CNNs and other related feedforward networks with recurrent/top-down connections. This is, however, a crowded field. No new mechanism is described here and there is already a substantial body of work that has shown that RNNs of the type studied here fit better neural data. There is also already quite a bit of work on the role of recurrent connections for object recognition under occlusions (see missing reference from O’Reilly et al below but also work by Tang et al cited etc). Compared to that type of work, the present study does not propose any novel architecture or computational mechanism. The main contribution is simply to show that “filling in” arises naturally when training RNNs under occlusion. This feels somewhat limited.

We believe that the main strength of our work is to show that the explaining-away computation occurs in generic neural networks that are not specifically designed to perform that type of computation. This suggests that the emergence of the explaining-away computation is probably generic given appropriate task orderings and recurrent structures and does not require any novel architecture or computational mechanism. This is important as we have relatively limited knowledge about the details of the architecture and computational mechanisms of most biological visual systems.

There have been previous studies focusing on object recognition under occlusion, but in our opinion, many, if not most, of them did not perform sufficiently extensive comparisons across feedforward and recurrent architectures as we mention in the second paragraph of the discussion. Moreover, in our opinion they gave a potentially inaccurate impression that the recurrent architecture is intrinsically superior to the feedforward architecture specifically for object recognition under occlusion, in contrast to object recognition without occlusion. Given the success of feedforward networks on object recognition without occlusion, perhaps it is not very surprising that they are not intrinsically inferior to the recurrent networks on object recognition under occlusion. But, given the ecological relevance of object recognition under occlusion, we feel that it is important to show this conclusively.

The more important contribution of our study is to find evidence showing that the explaining-away computation emerges in generic neural networks with natural task orderings and recurrent architectures. In this new manuscript, **we have strengthened our claim by showing that: 1. The explaining-away computation also emerges in a more realistic model of the primate visual system, CORNet, which predicts behavioral and neural responses to visual stimuli in primates and has a more biologically similar architecture, on a high-resolution dataset.**

2. Human subjects exhibit behavioral patterns strongly suggesting that the human visual system performs the explaining-away computation.

Below are more detailed comments. While most of these comments could be addressed I do not believe the manuscript would pass the threshold for Nature Communications.

From the perspective of ML or computer vision, the study is subpar. It uses a subset of what is already a somewhat toy dataset. The bar would be higher from an ML / computer vision dataset where multiple and larger scale datasets would have to be used to validate the approach. The dataset used really is a toy dataset with 4 classes, simple stimuli consisting of objects pasted on a uniform background. Hardly representative of the problem of object recognition.

We appreciate this comment as it drove us to perform more extensive experiments. As we mentioned above, we addressed this point by performing additional experiments using the CORNet model and a new high-resolution dataset with complex and realistic backgrounds and diverse orientations in 3D.

There are really two different models which were used to run different experiments. I am unclear what is biologically plausible about the biologically-inspired model which includes a GAN and a whole bunch of ML tricks. This seems also reminiscent of work done in computer vision including Huang et al (2020); see missing references below.

We agree with the reviewer that this model is not particularly biologically plausible in its specific architecture (we meant 'biologically inspired' to be taken literally, not as a synonym for 'biologically plausible'). We note that creating a fully biologically plausible model was not our goal. The main reasons for us to introduce PFC-net are: 1. To create a more concrete connection with the brain by incorporating existing neurobiological findings into the model. 2. Although we previously demonstrated that generic neural networks perform the explaining-away computation, their operations are often not human-interpretable. We aimed to develop a deep neural network model that executes this computation in a more explicit and intuitive manner. 3. The previous neural networks were trained to identify object categories, but the brain also represents finer visual features, such as texture and detailed shape. We sought to explore the extent to which these fine-grained visual features can be reconstructed using an explicit version of the explaining-away mechanism.

We have now added the missing references.

From the perspective of biological vision, a human baseline is missing. It is unclear whether human observers could solve the task without attention (as is the case with all the ANNs tested). Training computational models exhaustively on all pairs of classes seems completely unrealistic as a model of biology. As a result, it is unclear whether these ANNs have any validity as computational models. The use of very low resolution images further reduces the credibility of the models as models of biology.

We completely agree with the reviewer and have now performed both human psychophysics experiments and additional computational experiments involving a high-resolution dataset and the CORNet model. We believe these address the concerns.

The use of the term explaining away is problematic. The term “explaining away” has a precise meaning in the context of Bayesian/probabilistic networks. I am not sure why filling in here is equated with explaining away.

Our understanding is that 'explaining away' in the context of Bayesian/probabilistic networks refers to a reasoning pattern, where confirming one cause of an observed effect reduces the probability of other potential causes for the same effect. In fact, we believe that the explaining-away computation we described in the manuscript is an instance of this reasoning pattern: a neural network sees only parts of the background object and thus without first recognizing the presence of the foreground object, it may not easily recognize the background object. But once it understands why only parts of the background object are visible (i.e. due to the foreground object), it will deem other possibilities, such as the background object being some other novel object, less likely, and will be more likely to correctly classify the background object.

Based on the reviewer's suggestion, we considered using the term “filling in” but we found that it does not adequately describe the phenomenon we observed; the way the networks explain away occlusion may not involve explicitly filling in the occluded part of the background object, but potentially less human-interpretable computations that nevertheless reduce the estimated probability of other potential causes for the unusual shape of the background object. If the reviewer has a different suggestion, we would be happy to consider it.

Please consider splitting up citations (6-10) in two with modeling work (6-8) and data analysis (9-10).

We have now split up the citations.

Missing references:

Randall C. O'Reilly^{1,2*†}, Dean Wyatte^{1*†}, Seth Herd¹, Brian Mingus¹ and David J. Jilk². Recurrent processing during object recognition. *Front. Psychol.*, 01 April 2013 | <https://doi.org/10.3389/fpsyg.2013.00124>

Michaelis, C., Bethge, M. & Ecker, A. One-shot segmentation in clutter. In *International Conference on Machine Learning*,

Linsley, D., Kim, J., Veerabadran, V., Windolf, C. & Serre, T. Learning long- 433 range spatial dependencies with horizontal gated recurrent units. In Bengio, S. et al. 434 (eds.) *Advances in Neural Information Processing Systems*, vol. 31 (Curran Associates, 435 Inc., 2018).

Yujia Huang, James Gornet, Sihui Dai, Zhiding Yu, Tan Nguyen, Doris Y. Tsao, Anima Anandkumar. *Neural Networks with Recurrent Generative Feedback*. *Neural Information*

Dileep George*, Wolfgang Leirach, Ken Kinsky, Miguel Lázaro-Gredilla*, Christopher Laan, Bhaskara Marthi, Xinghua Lou, Zhaoshi Meng, Yi Liu, Huayan Wang, Alex Lavin, D. Scott Phoenix. A generative vision model that trains with high data efficiency and breaks text-based CAPTCHAs. *Science* 08 Dec 2017: Vol. 358, Issue 6368, eaag2612. DOI: 10.1126/science.aag2612

Reviewer #1 (Remarks to the Author):

I have no additional comments or questions to the authors.

Reviewer #4 (Remarks to the Author):

In this paper, the authors study the role of recurrent computations in facilitating occluded object recognition. There are three central claims—I will detail them and list my concerns.

One, recurrent neural networks (RNNs) and humans use “explaining-away” to recognize occluded objects. The difference in recognition accuracy when the foreground object is expected to be classified before the occluded object and when only the occluded object is expected to be classified is taken as a signature of “explaining-away.” In humans, the occluded-object classification performance difference between a foreground object shown alone before the mixed display and another foreground object shown alone before the mixed display is taken as a signature of “explaining-away.” In both settings, the idea is that taking the foreground object into account makes the classification of the occluded object easier.

Is this actually evidence for “explaining-away”? “Explaining-away” seems to be a vaguely defined term here. Is the idea that the representation of the foreground object is “deleted” from the stimulus representation? I don’t think the above results speak to this. If I think about what information the foreground object adds to the occluded object in terms of classifiability, it would be that it bounds the extent of the occluded object, possibly highlighting some full-object possibilities over others, thereby making classification better. This sounds more like conditional inference or, as the authors say sometimes, contextual inference.

We thank the reviewer for pointing out the potential confusion about the term explaining-away. Explaining-away is indeed a form of conditional or contextual inference where the recognition of the occluded object is improved by being conditioned on the fact that there exists a foreground object or by incorporating the context of the foreground object. We note that it doesn’t have to be by “deleting” the foreground object or bounding its extent, though those are certainly possible. For instance, it could be through reconstitution (Fig. 7).

How exactly this computation would look in the RNN representations is unclear, although such “interpretability” steps can indeed be taken to understand RNN operations (for example, see Thorat et al., 2021). Another possibility, which is closer to “explaining away,” is the idea that the network/humans end up focusing too much on the foreground object (which is larger), thereby not allowing the readout to respond to the occluded object. Here, somehow, asking to temporally separate the recognition of the foreground and background might allow attention to the background object and not focus on the foreground object. However, it is unclear why asking for foreground classification first increases the classification accuracy of the background object compared to asking for background classification first. A mechanistic understanding is essential and lacking here.

We thank the reviewer for suggesting a very interesting perturbation experiment to gain further insights into the inner workings of RNNs performing explaining-away. In the spirit of Thorat et al. (2021), we conducted an experiment where we feed a multi-object image to an RNN and replace its hidden state with that of an identical RNN, which is instead fed with a modified image in which the background object remains exactly the same but the foreground object is modified (Fig. 5A top left in the revised manuscript). This RNN is the Rec model considered in Fig. 1-3, which predicts the foreground object category in the first timestep, and then the background object category in the second timestep. If it feeds back information about the foreground object through its recurrent connections to improve the recognition of the background object, then such a perturbation will result in decrease in accuracy of the background object recognition. We found that this is indeed the case across different types of modifications of the foreground object, where we changed its category, its orientation, or both (Fig. 5B left). This was not due to the change in the distribution of occlusion levels (Fig. 5A bottom).

In addition to perturbing the hidden states at all layers at once, we perturbed the hidden state of each layer separately to understand which layer's recurrent information flow is most important. We found that the performance deficit was most significant when the first layer was perturbed, and the effect of the perturbation tended to decrease for higher layers. This suggests that the recurrent information flow most important for explaining-away happens in lower layers. Interestingly, the performance deficit was greater when only the first layer was perturbed than when all layers were perturbed. We speculate that when the recurrent information flow is inconsistent between the first and other layers, that could cause some kind of "confusion" in the network, leading to greater performance deficit. On the other hand, when we performed the same kind of a perturbation experiment for the recognition of the foreground object (Fig. 5A top right), we found that there was almost no performance deficit (Fig. 5B right). This implies that, in contrast to the case of background object recognition, information about the background object fed back through recurrent connections did not improve the recognition of the foreground object. This makes sense, as the foreground object is already fully visible and thus the network would not benefit much from the information about the background object. Therefore, these experiments provide direct evidence that information about the foreground object improves the recognition of the background object through recurrent connections, especially in the lower layers of the network.

We also tried the control perturbations from Thorat et al. (2021), where we randomly permuted elements of the original perturbations. However, in both of our experiments above, the control perturbations led to a much more significant decrease in performance than the original perturbations (Fig. R1). Although we do not fully understand why this is so, the hidden states in the control perturbation condition are highly out-of-distribution for the network and therefore we believe they can lead to such a significant drop in performance.

Fig. R1. Performance change under the control perturbation conditions for the background (left) and foreground (right) object recognition. Same convention as in Fig. 5B.

Additionally, the claim that RNNs “use” “explaining-away” to recognize occluded objects is a bit misleading given these results. To make that claim, you’d have to show that the RNN trained on background-classification-only internally performs “explaining-away.” Instead, here a very specific objective manipulation is made wherein the foreground object classification is required before the background. I’d say this possibly tells us that RNNs “can use” “explaining-away” to recognize occluded objects.

We completely agree with the reviewer that RNNs do not always perform explaining away to recognize occluded objects. As the reviewer pointed out, one of our main findings is that the temporal ordering of foreground and background object recognition is crucial for explaining-away effect to arise. We clarified this point in the revised manuscript with the following paragraphs (the added parts in bold):

“The fact that explaining-away emerged despite networks never directly being trained to do so suggests that this property is likely insensitive to the learning algorithm employed (of which we have very little knowledge), but rather is a robust consequence of the recurrent architecture **together with the temporal structuring of the task where the network is required to recognize the occluder prior to the occluded object.**” (in the introduction, on page 3 of the revised manuscript)

“We believe that the utility of recurrent connections for occluded object recognition lies in their capacity to facilitate explaining-away. We found this feature emerged in our results even when networks were trained end-to-end in a fully agnostic and unstructured way. **Importantly, its emergence depends on the temporal ordering of task demands —namely when the network is required to recognize the occluder prior to the occluded object. Thus, our findings do not suggest that recurrent networks universally perform explaining-away, but rather that they are capable of doing so under conditions where such a computation is facilitated by the task structure.**” (in the discussion, on page 21 of the revised manuscript)

Two, RNNs use “explaining-away” but deep feedforward NNs (FNNs), which perform as well as the RNNs, do not. The FNNs are trained to recognize the foreground and background objects at the same computational depth, and joint classification hampers background object classification, if anything.

As mentioned above, RNNs “can use” “explaining-away”—it is unclear if they do this implicitly. Now the FNNs were never asked to “segregate” the classification of foreground and background objects in computational depth. Perhaps if the ANNs are trained, Inception-style, with foreground

readout coming before background readout, possibly in a computational-depth-tied-with-RNNs manner, we could examine them for corresponding “explaining-away” effects in performance.

We thank the reviewer for suggesting this interesting experiment. We performed a new experiment where we compared the performance of two new classes of feedforward models, FF Seq models, and FF Seq Reverse models (Fig. 5C). FF Seq first predicts the foreground object category and then predicts the background object category in the subsequent layer, whereas FF Seq Reverse first predicts the background object category and then the foreground object category. Importantly, we controlled for the computational depth of the background object recognition between the two types of models for fair comparison, as we have done for recurrent networks (Fig. 3D). Specifically, we built and trained FF Seq and FF Seq Reverse versions of the feedforward models we considered in the previous comparison experiment, FF, FF Wider Taller, and FF WS Wider Taller (the weight-shared version of FF Wider Taller) (Fig. 1D). They are designed such that the computational depth of the background object recognition matches that of the original feedforward models. For example, FF predicts categories of both objects after three convolutional layers and one fully connected layer. In comparison, FF Seq first predicts the foreground object category after two convolutional layers and one fully connected layer branching out from the second convolutional layer, and then the background object category after three convolutional layers and one fully connected layer. Conversely, FF Seq Reverse first predicts the background object category after three convolutional layers and one fully connected layer branching out from the third convolutional layer, and then the foreground object category after four convolutional layers and one fully connected layer. We found that for all three cases, the FF Seq version outperformed the FF Seq Reverse version (Fig. 5D), suggesting that the temporal structuring of the readouts gives rise to explaining-away computation not just in recurrent networks, but also in feedforward networks.

Finally, we note that in the process of implementing these temporally structured feedforward models, we found that training some variants, including the FF Seq version of FF Wide, was less stable and yielded more variable outcomes. This suggests that while structuring readouts to emulate explaining-away computations can yield performance benefits, it may also make these models more difficult to train reliably than their standard counterparts.

Conversely, if the RNNs are trained to classify both foreground and background objects at the same time, would we also see the background-task performance deficits we see for FNNs? Apparently not—the Rec Control model shows a small advantage given joint classification versus background-only classification. How do we interpret this? This RNN-FNN comparison requires more nuance.

We indeed considered RNNs trained to classify both foreground and background objects at the same time, named Rec Last and Rec Last 5 (Fig. 1D). Both models exhibited the background-task performance deficits, similarly to the feedforward models (Fig. 3B).

As for the Rec Control model, we found that training of the Rec Control model was substantially more unstable than that of recurrent models that are actually trained to perform recurrent

computation, such as the Rec model (Fig. R2). We suspect that this is because certain RNN architectures, such as convolutional LSTMs, are not designed to perform purely feedforward computation. Such difficulty in training might have contributed to the Rec Control model showing a small advantage given joint classification, as joint classification would provide more labeled training samples to learn from.

Fig. R2. Total classification accuracies of Rec and Rec Control models. n=20 model instances for each model.

Three, PFC-net is a biologically inspired model for “explaining-away.” The authors assume that reconstitution of the occluded object’s representation happens in vIPFC. They model it by explicitly asking the encoder to take separate inputs of the foreground-only object and the mixed images and requiring an unoccluded representation as the output (as the latent space of the decoder/generator is tuned for intact objects). They show that this network uses the recurrence to refine the representation of the occluded object, and it seems (Fig. S5E) that the additional foreground-only input was essential.

Isn’t the notion of asking for a “reconstruction” of the occluded image similar to Tang et al. (2018)’s approach using Hopfield networks? Of course, in your case, the additional foreground-only input is essential; however, that could also be given as an input to a Hopfield network. The reason I am asking this is that I’m unsure what the right computational motif is for vIPFC. Is it truly a generative model as it is being modeled here? If not, should we call this PFC-net?

As the reviewer pointed out, PFC-net is indeed similar to the Hopfield network approach in Tang et al. (2018) in terms of the objective. The motivation for devising it was to build a neural network model that incorporates qualitative features observed by Fyall et al. (2017) and reconstitutes semi-realistic occluded visual stimuli, unlike the more abstract model considered there. It also serves as an example of a neural network that performs a more explicit form of explaining-away. However, although we believe that vIPFC might perform a version of explaining-away computation, we do not claim that PFC-net is a true mechanistic model of how vIPFC improves occluded object recognition. To avoid potential misunderstanding, we now call it Reconstitution Network (Recon-Net) in the revised manuscript.

In summary, I think this paper convincingly demonstrates an interesting effect—processing the foreground object temporally ahead (or perhaps with a lower computational depth) of the background object enhances the classifiability of the occluded, background object. How this interaction between the foreground and occluded objects works is left unclear, although the

authors link it to the term “explaining-away.” Future work needs to understand how this interaction works inside RNNs and inside human brains.

Minor comments:

1. One way to check if the foreground object is used by a Rec model trained with a background-only condition is to train the Rec model with background-only cutouts. If the performance for the latter case is lower than the former, then the background-only task is already an interesting testbed for contextual inference effects.

We thank the reviewer for this thoughtful suggestion. In response, we compared the Rec model’s performance in the standard and the background-only-cutout condition (Fig. R3). We found that the model actually performed substantially better in the background-only-cutout condition. We interpret this as follows: in the background-only-cutout condition, the model only sees the partially visible background object, allowing the model to focus exclusively on its features for recognition. In contrast, in the standard condition, the background object is partially occluded by the fully visible foreground object, requiring the model to disentangle competing signals from both objects. This increased complexity likely accounts for the model’s lower accuracy in the standard condition.

Fig. R3. Background-only classification accuracies of Rec models in the standard and background-only-cutout condition. n=5 model instances for each condition.

2. In the Introduction, paragraph 3, you mention that the “nature and role of recurrent computations remain unknown.” On the contrary, there’s plenty of work showing how RNNs perform figure-ground segmentation, contour tracing, other Gestalt grouping phenomena, and also object recognition (e.g., Linsley et al., 2020; Thorat et al., 2021; Goetschalckx et al., 2022; Thorat et al., 2023). I wouldn’t say we know everything there is to know about recurrent computations, but there are a couple of neat ideas out there on what kind of functions RNNs can compute “naturally.”

We thank the reviewer for pointing out this sloppy phrasing and these references. We only intended to say the role is not fully known. In the introduction of the revised manuscript (page 2), we clarify and also refer to these previous studies on potential functions of recurrent computation in visual recognition (the added references indicated in bold):

“Several studies have proposed specific roles for recurrence in vision—including figure-ground segmentation, contour integration, and object recognition **(21-24)**—yet the precise nature and

mechanistic role of recurrent computation remain only partially understood. A key experimental challenge is that most neurons possess both feedforward and recurrent connections, making it extremely difficult to selectively perturb one or the other. Therefore, in this study, we tackled these questions by constructing feedforward and recurrent models of the visual system and analyzing their performance and internal representations. For our models, we use deep convolutional neural networks, which have been shown to not only achieve human-level performance but also predict cortical visual responses for object recognition (6, 8, 25-27). ”

Regards, Sushrut Thorat

References:

Goetschalckx, L., Zolfaghar, M., Ashok, A. K., Govindarajan, L. N., Linsley, D., & Serre, T. (2022). Toward modeling visual routines of object segmentation with biologically inspired recurrent vision models. *Journal of Vision*, 22(14), 3773-3773.

Linsley, D., Kim, J., Ashok, A., & Serre, T. (2020). Recurrent neural circuits for contour detection. In *International Conference on Learning Representations*.

Thorat, S., Aldegheri, G., & Kietzmann, T. C. (2021). Category-orthogonal object features guide information processing in recurrent neural networks trained for object categorization. In *SVRHM 2021 Workshop@ NeurIPS*.

Thorat, S., Doerig, A., & Kietzmann, T. C. (2023). Characterising representation dynamics in recurrent neural networks for object recognition. *Conference on Cognitive Computational Neuroscience*.

Reviewer #5 (Remarks to the Author):

The manuscript by B. Kang et al. titled “Recurrent connections facilitate occluded object recognition by explaining-away” presents very interesting work, but I am not really certain that the quality of the work is suitable for a high impact journal as *Nature Communications*. I can see many important problems which question the suitability of this paper.

I have structured the major problems into five areas:

Training dataset

I very much doubt the ecological validity of the training datasets. I don't think that flying pianos are really very common in our natural environment. I think the authors need to work really hard to convince me and potential readers that these are suitable test sets for the scientific problem in question.

We of course completely agree that our image sets—featuring objects in unusual orientations or in atypical environments—are not ecologically valid in the sense that they are not representative of specific real world environments. However, this design choice was motivated by the need to rigorously test our model's ability to recognize occluded objects without relying on contextual shortcuts.

Specifically, in typical real-world images, certain objects (e.g., chairs) tend to co-occur with specific backgrounds (e.g., living rooms). If we used only naturalistic photos, a human or machine observer could often guess the identity of an occluded object merely by leveraging common co-occurrence patterns (e.g., seeing part of a piano on a concert stage and inferring “piano” from the stage). By placing objects in unconventional settings—such as a piano “flying” against a random backdrop—we deliberately disrupt these common contextual correlations. This compels the observer (model or human participant) to rely on the actual features of the occluded object rather than on typical scene cues.

Similarly, we use objects in unusual orientations or locations to mitigate the possibility that indirect cues (e.g., typical position in a scene, standard viewpoint) will drive recognition. By presenting objects in orientations that deviate from everyday viewpoints, we ensure that recognition must be achieved primarily through the fundamental shape and texture properties of the object, rather than from familiarity with a canonical pose or environment.

These design choices help us isolate the specific mechanism under investigation—namely, the recurrent “explaining-away” of occlusion—by removing confounding factors such as scene context or canonical object poses. While the resulting stimuli may look unusual, they provide a stricter test of the network’s (or participant’s) capacity to integrate incomplete visual information over time.

In addition, we note that many prior studies on visual object recognition have deliberately used carefully contrived images in order to isolate and examine specific visual processes. For instance:

1. Pinto et al. (2008) explicitly showed that using uncontrolled “natural” images could be misleading for benchmarking object recognition models, as even a simple “V1-like” toy model outperformed more advanced models on these images. Furthermore, they showed that more controlled images similar to ours exposed the inadequacy of the V1-like model.
2. Yamins et al. (2014) employed controlled synthetic stimuli to systematically test hierarchical models against neural responses in higher visual cortex.
3. Rajalingham et al. (2018) used controlled image sets to conduct large-scale comparisons of visual object recognition behavior in humans, monkeys, and deep neural networks.

These works demonstrate how synthetic or atypical stimuli can be valuable in disentangling the fundamental computational principles underlying vision, free from the confounds often present in fully naturalistic environments.

In summary, our use of synthetic, unconventional images is a commonly used, strategic choice to strip away contextual biases and thoroughly assess the principle of recurrent inference under occlusion.

Feedforward networks

I don’t think that the structure of the feedforward networks with two outputs is a suitable structure. It seems very counter-intuitive to me that we would have different representations in the brain one for background objects and one for foreground objects. Perhaps an architecture with a “task node” (e.g. “look for background object”) would be more suitable.

We appreciate the reviewer's insightful comments regarding our feedforward network structure. We would like to clarify that our model is not intended to imply that the brain necessarily has dedicated, separate readout neurons for foreground and background objects. In fact, our current understanding of how different objects in a multi-object scene are represented—especially under occlusion—is far from complete. It is conceivable that, rather than relying on anatomically distinct readouts, the brain may encode foreground and background objects through distinct dimensions in the same population response or, alternatively, via temporally separated processing.

Our recurrent network models capture this latter possibility by predicting the categories of the foreground and background objects at different time steps using a single, shared set of neurons and parameters. This finding aligns with the recent work by Kar et al. (2019), which demonstrated that challenging images—particularly those involving occlusion—require additional processing time for robust object category selectivity to emerge in the ventral stream.

While the idea of incorporating a task node is indeed interesting, it presupposes that the observer is performing a specific, task-driven discrimination (e.g., explicitly deciding whether to recognize the foreground or background object). In contrast, object-selective responses in IT cortex are observed even during passive viewing, without any explicit task instructions (e.g. Hung et al. (2005), Kar et al. (2019), Kar et al. (2021)). Thus, we believe that a temporal separation in the representation of occluded objects, as implemented in our recurrent model, offers an equally parsimonious and neurobiologically plausible account as architectures requiring explicit task-driven gating.

In summary, while our feedforward models with separate readouts serve as a useful approximation for probing the effects of occlusion, our main conclusion—that recurrent connections facilitate occluded object recognition by explaining-away—remains robust irrespective of this specific architectural choice.

Behavioural experiment

Like with the training dataset I doubt that the pictures in the behavioural experiment are ecologically valid.

We acknowledge that the images in our behavioral experiment are not fully ecologically valid. However, as we explained above, our goal was to remove confounding contextual cues—commonly present in natural scenes—that could aid object recognition. By using controlled stimuli, we ensure that participants rely on intrinsic object features rather than on contextual correlations. Moreover, the lack of available datasets that both control occlusion and maintain full ecological validity necessitates this compromise for a rigorous test of our hypothesis.

Perhaps this would also explain the surprising high error rate. I don't think the objects as such are that difficult to recognize to warren such a high error rate. Perhaps the context and the spatial arrangement put participants off?

We agree of course that the spatial arrangement and context are unusual. However, we would like to note that prior to the main task, participants underwent a preparatory phase in which they were shown the types of images used in the experiment and given an opportunity to practice with them. This was intended to familiarize them with the non-naturalistic format of the stimuli, so the visual presentation was not entirely novel or unexpected by the time the actual task began.

In general the high error rate questions the validity of the outcome.

We respectfully disagree with the concern regarding high error rates. Difficult tasks naturally yield lower performance, and our findings are consistent with prior work. For instance, in Tang et al. (2018) (see Fig. 1F), participants performing an occluded object recognition task—with occlusion levels overlapping with our study (70–90% occlusion)—achieved 60–90% accuracy for image presentation times greater than 75 ms, aligning well with our results. In fact, for presentation times less than 50 ms, their accuracy dropped to 40–70%, which is lower than our observed performance. Similarly, Ollikka et al. (2024) (see Fig. 3) reported accuracies ranging from 70% (at 40 ms viewing time) to 90% (at 150 ms) for non-occluded stimuli in unusual object poses, and Kassaw et al. (2024) (see Table 3) observed approximately 80% accuracy for greater than 50% occlusion. These examples demonstrate that our error rates fall within the expected range for challenging visual tasks, thereby supporting the validity of our outcomes.

Regardless, we performed a new psychophysics experiment to test whether the images used in the study impede subjects' ability to perform the task. This new experiment was identical to the original except subjects were asked to identify the foreground object as opposed to the occluded object. 20 out of 55 subjects passed the same quality controls as in the original experiment. Subjects performed this task with mean accuracy > 90%, indicating that the images used for the psychophysics experiment do not preclude high performance (Fig. R4). The relatively lower performance on the original occluded object recognition task is thus most likely explained by that task being more challenging due to occlusion.

Fig. R4. Performance in the new psychophysics experiment. n=20 subjects.

I would suggest to re-run that study with better images.

We respectfully maintain that our choice of synthetic images is deliberate and essential for isolating the mechanisms under investigation. Our stimuli were specifically designed to eliminate contextual correlations that might otherwise facilitate recognition through scene cues. As we discussed, the atypical presentations force both human observers and models to rely solely on intrinsic object features, thereby rigorously testing the role of recurrent processing in occluded object recognition.

Moreover, the observed error rates are consistent with previous studies employing similarly challenging conditions, and they reflect the increased difficulty imposed by occlusion rather than any deficiency in the images per se. Re-running the study with more “naturalistic” images risks reintroducing confounding contextual cues. In light of these considerations, we believe that our current approach effectively addresses the scientific question at hand without necessitating additional experiments with alternative image sets.

I am also wondering whether this effect is related to well-known object priming effects. I suggest to include a discussion of priming effects in the manuscript e.g., how their findings compare to these well-established effects.

We thank the reviewer for pointing out the issue of priming. We had multiple discussions regarding how to recapitulate the computational experiments in a psychophysics experiment before settling on comparing trials in which the specific foreground object is shown before the multi-object images to those in which a random object was shown. Priming effects were the reason we did not consider alternatives in which the background object, which is the object to be reported, were shown in any way. In a way, our experiment is designed to tap into priming effects, for example *“in occluder trials, processing the single object in the single image epoch should allow it to be better explained-away when the occluded image is shown, and therefore the background object to be more easily recognized than in random trials”*. However, the priming should apply to the foreground image and we believe that having that effect transfer to the background image is non-trivial. While priming effects are complex, they (in our reading) have two broad effects: (i) bias towards selecting a primed object and (ii) reaction time. A bias towards the primed object would actually be *counterproductive* here since the background object needs to be reported. Because of reaction time effects we focused on the improvement in accuracy (Fig. 6B,E) and didn’t over-interpret the improvement in reaction time.

Therefore, it is unlikely that priming effects explain our findings. While showing the foreground object could arguably lead to faster processing of the foreground object, it could equally well lead to biases in picking the foreground object instead of the background object in ambiguous cases, which would actually reduce performance. We now include a discussion of this issue (page 22 in the discussion section):

“One question that arises is how these findings might relate to well-known object priming effects, wherein prior exposure to an object can bias or speed its subsequent processing (53). In principle, priming can manifest in at least two ways: (i) a bias toward reporting the primed object and (ii) faster or more efficient processing of the primed object once it appears. However, neither of these straightforwardly explains our primary finding. First, a bias toward the primed (foreground) object

would be counterproductive—participants needed to report the background object. Indeed, if classic priming were the dominant factor, one would expect participants to be more likely to mistakenly choose the primed foreground in ambiguous situations rather than correctly identify the background. Such an effect would hurt performance instead of helping it. Second, even though priming the foreground object could speed its processing, it would not trivially confer an advantage on recognizing the different, unprimed background object, unless some use of the recognition of the foreground object is used to facilitate processing the background object, i.e., explaining away. Our observed benefit for identifying the background object thus suggests a process extending beyond conventional object priming.”

Also I should note that what the authors termed “epochs” are probably experimental blocks. We appreciate the reviewer’s careful reading and thoughtful feedback. We chose “epochs” to denote the distinct, time-bounded periods within each trial of our experimental design (i.e., the single-object presentation, the occluded-object presentation, and the response period). In our view, “blocks” typically refer to larger segments of an experiment, each of which might contain multiple trials, whereas we wanted to emphasize the sequential and functionally distinct nature of these sub-trial phases. Hence, we felt that “epochs” would most accurately capture these discrete time windows. However, we appreciate the reviewer’s point regarding potential differences in terminology across contexts, and we hope this clarification addresses any confusion.

Explaining away

At some point in the manuscript the authors defined “explaining away” as a process as “... reconstituting the unoccluded representation”. Given centrality of this concept, it is important to relate their definition to other definitions in the literature. Is it unique? Is there agreement in what it means? Is it actually implicitly based on an iterative mechanism? If this is the case, it would be impossible in principle for a feedforward architecture to realized “explaining away”? I guess the definition does not include iterative processing. This makes me wonder whether the forward architectures also reconstitute representations or perhaps other types of “explaining away”. This seems an important question in the context of the paper. So the authors should explore this using their RSA method.

We thank the reviewer for these insightful suggestions. Before we address feedforward architectures below, just to clarify, our definition of explaining-away does not inherently assume an iterative mechanism or reconstitution. Rather, we conceptualize explaining-away as a form of conditional inference in which recognizing the foreground object facilitates the recognition of the background object by providing a context to interpret the unusual appearance imposed by occlusion. Possible versions of that could be, as Reviewer #4 suggests, “zeroing out” occluding pixels, or establishing a bounding box for the unoccluded parts. Another version, which we develop in more detail since it is more involved than others is a process by which the conditioning does not happen at the readout, which can be kept the same, but rather an iterative modification of the internal representation to make it more similar to the one which would have been generated by the object if it were not occluded. We don’t define explaining away this way, we just present it as a possible implementation (which in our minds is both interesting and related at a high level to certain recent physiological findings such as Fyall et al. (2017)).

To address the reviewer's suggestion, we examined whether feedforward architectures can exhibit explaining-away by constructing models that sequentially predict the foreground and background objects in both the standard (foreground \rightarrow background) and reverse (background \rightarrow foreground) order. Across all feedforward architectures tested (Fig. 5D), networks trained with the standard ordering consistently outperformed those trained with the reverse ordering. Notably, we controlled for computational depth for processing the background object, isolating the effect of explaining-away. These results suggest that the explaining-away effect arises due to the temporal ordering of readouts, even without recurrent connections or iterative mechanisms.

Additionally, we analyzed the representational similarity between feedforward and recurrent models trained to predict the foreground and background objects in either the standard or reverse order. For recurrent models, we selected the Rec and Rec Reverse 3 architectures (corresponding to standard and reverse ordering, respectively). For feedforward models, we used FF Wider Taller Seq and FF Wider Taller Seq Reverse, as they best matched the performance of the recurrent models. We conducted representational similarity analysis in both directions—using recurrent models as source models and feedforward models as target models (Fig. R5A), and vice versa (Fig. R5B). In both cases, results were qualitatively similar.

To ensure a meaningful comparison, we performed the analysis on activations before and after the readout weights at the layer or time step where the network predicts the background object's category. These were the only activations that could be meaningfully mapped between feedforward and recurrent architectures. Our findings revealed that models trained with the standard ordering—whether feedforward or recurrent—exhibited greater representational similarity to their recurrent or feedforward counterparts than models trained with the reverse ordering did, although the gap narrowed when the target model had reverse ordering. Moreover, across all comparisons, representational similarity was lower when the target model had reverse ordering than when it had standard ordering.

In addition, we compared the degree of agreement between predictions of different models (Fig. R5C). For this analysis, we only used held-out test examples where both models made incorrect predictions to exclude the possibility that agreement arises due to both models making the correct predictions. We found that the degree of agreement was highest when both models had standard ordering and lowest when both models had reverse ordering.

This suggests that networks capable of explaining-away (i.e., those with standard ordering) tend to converge on similar computational processes and representations, whereas those without this mechanism (i.e., those with reverse ordering) exhibit greater variability in both.

These findings support the notion that explaining-away can emerge in both feedforward and recurrent architectures, provided the temporal ordering of object recognition aligns with conditional inference principles.

Fig. R5. (A) Representational similarity when Rec and Rec Reverse 3 are the source models for activations before (left) and after (right) readout weights. $n=25$ pairs for the solid magenta and cyan lines, and $n=20$ pairs for the dotted black lines, excluding the pairs of the same instances. **(B)** Same as (A) but when FF Wider Taller Seq and FF Wider Taller Seq Reverse were the source models. **(C)** Degree of agreement, defined as the fraction of examples where predictions of both models were the same. $n=25$ pairs.

Finally, when the authors analysed the recurrent networks, did they actually re-train them again? If so, why? Or were these the same networks with the same weights as in the previous sections?

For our representational similarity analyses, the models' architectures were identical in form to the ones used in the first subsection, but they had more units in convolutional layers to achieve a sufficiently high performance (>70% classification accuracy) on the dataset used for PFC-net (which we renamed "Reconstitution Network" in the revised manuscript) as we explained in the main text (page 17 of the revised manuscript).

Attention

The authors completely ignore an important alternative mechanism on how the brain may deal with occlusion, visual attention. This mechanism is particularly pertinent given the recent success of Transformers (or see below for papers from my own work with a transformer-like attentional mechanism). It is also interesting to note that my work explicitly combines recurrent connections with visual attention. I would like the authors to include a discussion of the relevance of attention in the paper and this concept is related to their findings, possibly as part of a future studies discussion.

We thank the reviewer for pointing out the importance of discussing attention. Our intention was to be clear that we do not claim to have found *the* mechanism for dealing with occlusion, rather a mechanism for dealing with occlusion (e.g., from the abstract "Finally, to demonstrate a specific mechanism for explaining-away..."). We completely agree however with the reviewer that the focus of the figures (and therefore writing) on reconstitution may imply that it is the only mechanism. This is certainly not the case and we did not intend to imply otherwise. Explaining away could occur without reconstitution, for example, as Reviewer #4 suggests, by deleting the foreground object or transforming the "bounding box" for the occluded object leading to only the

unoccluded parts being used for classification, and is closely related to the machine learning version of attention referred to above. We thank the reviewer for the opportunity to correct this oversight and have now included the following paragraph in the discussion (page 23 of the revised manuscript) with the added references indicated in bold:

“The presence of occlusion renders the recognition of objects more difficult. Whether recognition is conceived of as a template-matching process or involves detecting the presence of a constellation of expected finer-scale features, occlusion leads to “atypical” values for these processes. We demonstrated one approach in Recon-Net, which reconstitutes a representation of the occluded object, potentially providing a more complete input to the recognition process. This is not the only option, and while we cannot enumerate all possibilities, a few stand out. First, the system may limit the scope of recognition to unoccluded portions of the image; such a process is reminiscent of the attention mechanisms used in transformer-based models and has been explored in models supporting localized dynamics (**56, 57**). Alternatively, the system might maintain the same spatial extent but lower the recognition threshold to accommodate the mismatch caused by missing or corrupted features. Finally, if recognition involves competitive or winner-take-all dynamics, then none of the above mechanisms needs to perfectly ‘explain away’ the missing information; they only need to resolve enough of the ambiguity so that the correct object emerges robustly from the recognition process.”

Abadi, A. K., Yahya, K., Amini, M., Friston, K., & Heinke, D. (2019). Excitatory versus inhibitory feedback in Bayesian formulations of scene construction. *Journal of The Royal Society Interface*, 16(154), <https://doi.org/10.1098/rsif.2018.0344>

Heinke, D., & Humphreys, G. W. (2003). Attention, spatial representation and visual neglect: Simulating emergent attention and spatial memory in the Selective Attention for Identification Model (SAIM). *Psychological Review*, 110(1):29-87.

Regarding “explaining away”:

While explaining away is a term that is used in many different ways in different contexts, we feel that the way it is used in our paper is consistent with both the colloquial use of the term and in core aspects of the more precise usage of the term. Specifically, we take the more precise usage of the term to be: “...information about one of the causes tends to make the other more or less likely, given that the consequence has occurred. This pattern is known as selection bias... and as the explaining away effect” (Pearl (2000), *Causality* pp. 17) or “Mr. Holmes perceives two episodes which may be potential causes for the alarm sound, an attempted burglary and an earthquake. Even though these two events are *a priori* independent and so, not mutually exclusive, still the radio announcement reduces the likelihood of a burglary, as it “explains away” the alarm sound” (Kim and Pearl 1983). There are different ways to set up the causes in our case (occlusion) but loosely speaking, the event is an unexpected collection of visual features present in the occluded image (unexpected in the sense of the usual collection of features associated with single objects), one possible cause is a truly new object being present or the object recognition system not being rich enough and the other cause is an occluding object. Finding an occluding object (one cause) makes the other cause (mismatch) less likely. That being said, we are definitely aware that there are other subtler aspects to the use of explaining away in probabilistic graphical models that may end up being confusing. We feel that completely removing the term wouldn’t be

the optimal solution and therefore added the explanation below to the introduction (page 3 of the revised manuscript). That being said, if the reviewers believe strongly that using alternative terminology, such as “occlusion-recognition conditional inference”, would enhance clarity, we are open to making this adjustment:

“Our use of the term “explaining-away” is rooted in its typical formulation in probabilistic graphical models (32), where identifying one cause of an observed outcome can reduce (or “explain away”) the need to posit other causes. Concretely, in the context of occlusion, the observed “outcome” is an unusual collection of visual features that deviate from those normally associated with a single visible object. One potential explanation for these unexpected features is that a genuinely unfamiliar object is present, whereas another explanation is that the scene includes an occluding object. Recognizing that an occluder is the cause of the unusual features thus makes it less necessary to posit alternative explanations for those features. While the exact usage of “explaining-away” varies in the literature, here we focus on this intuitive notion that identifying an occluder helps the system handle the mismatch in visual features, thereby facilitating recognition of the occluded object.”

Reviewer #4 (Remarks to the Author):

Thank you for a very thorough response to my questions/concerns. I am convinced that asking networks (feedforward or recurrent) to first identify the occluder helps with subsequently identifying the occluded object. How exactly the identification of the occluder helps with the subsequent identification of the occluded object, in the RNNs, is left unclear.

Reconstruction of the occluded object is put forth as a hypothesis, exemplified through ReconNet, however, it is unclear if the previously discussed RNNs are indeed in the business of reconstruction. This requires going beyond the manipulations presented in Fig. 5, which is non-trivial and could possibly be for future work. However, a simple modification to the Rec networks - feedback to input (similar to the trick employed by Thorat et al. 2021) - might help us "see" what the Rec network chooses to do with the initial identification of the occluded object - does it 1) remove those pixels, 2) fill-in those pixels (similar to ReconNet), or 3) does something else entirely? Of course, this modified Rec network would only provide another hint at the underlying mechanism. Whether this is within the scope of this work I leave to the authors and the editor - it definitely is a non-trivial addition. A related next step would be the analysis of the background_only Rec networks. Does classifying just the occluded object lead to similar mechanisms seen in the F+B classification networks? If yes, then we are looking at a general mechanism which is boosted by explicitly asking the networks to classify the foreground first. If not, then indeed, asking for prior foreground classification would be an important task prior for a performant visual processor. How the brain acquires and executes such a prior is the next interesting question.

We thank the reviewer for this thoughtful suggestion. Indeed, that could be an interesting approach to further investigating how exactly the RNNs achieve improved recognition of the occluded object using the information about the occluder. However, the suggested modification to the Rec network (and similarly to the background-only Rec network) involves a highly significant and non-trivial re-design of the architecture, and getting it to properly to train, if possible at all, would require a new extensive hyperparameter search, which costs significant computational resources and time. Moreover, as the reviewer noted himself, visual inspection of the feedback input would only provide yet another hint, but no conclusive evidence as to the exact mechanism by which the RNNs leverage the occluder information; even in the best-case scenario where the feedback input looks as if the network removes or fill in the missing pixels, there is no guarantee that the network actually leverages the feedback input in a human-interpretable way. We therefore believe that this is beyond the scope of this work.

I think this work opens an interesting direction in understanding the processing of occluded objects and needs to be read by a wide audience interested in this fundamental aspect of visual processing. I'm happy to support its publication.

Thank you,
Sushrut Thorat

Reviewer #4 (Remarks on code availability):

Code not provided.

We have updated our manuscript to include a link to our GitHub repository containing the codes used in this study.

Reviewer #5 (Remarks to the Author):

I very much appreciate the way the authors have dealt with my comments. They really responded to most of my points in a convincing way.

However, I must admit that I still have some gremlins with the lack of the ecological validity of their stimuli. But I can see that that there are good arguments for using them. I guess it comes down on how much out-of-distribution these images can be before they become no longer a valid experimental test of human information processing. Anyway, I will leave this to future empirical studies and I would like to congratulate the authors to their excellent manuscript.

We thank the reviewer for this comment. As the reviewer noted, even though there are good reasons to use our visual stimuli, our stimuli are indeed ecologically atypical. We believe future studies could shed more light on this matter by using a wider variety of naturalistic objects and more ecologically realistic arrangements of occlusion.